# Pessimism in the Face of Confounders: Provably Efficient Offline Reinforcement Learning in Partially Observable Markov Decision Processes

**Miao Lu[1], Yifei Min[2], Zhaoran Wang[3], Zhuoran Yang[2]**
[1]University of Science and Technology of China, [2]Yale University, [3]Northwestern University
`lumiao@mail.ustc.edu.cn, zhaoranwang@gmail.com`
`{yifei.min,zhuoran.yang}@yale.edu`

## Abstract

We study offline reinforcement learning (RL) in partially observable Markov decision processes. In particular, we aim to learn an optimal policy from a dataset collected by a behavior policy which possibly depends on the latent state. Such a dataset is confounded in the sense that the latent state simultaneously affects the action and the observation, which is prohibitive for existing offline RL algorithms. To this end, we propose the Proxy variable Pessimistic Policy Optimization (`P3O`) algorithm, which addresses the confounding bias and the distributional shift between the optimal and behavior policies in the context of general function approximation. At the core of `P3O` is a coupled sequence of pessimistic confidence regions constructed via proximal causal inference, which is formulated as minimax estimation. Under a partial coverage assumption on the confounded dataset, we prove that `P3O` achieves a $n^{-1/2}$-suboptimality, where $n$ is the number of trajectories in the dataset. To our best knowledge, `P3O` is the first provably efficient offline RL algorithm for POMDPs with a confounded dataset.

## 1 Introduction

Offline reinforcement learning (RL) (Sutton and Barto, 2018) aims to learn an optimal policy of a sequential decision making problem purely from an offline dataset collected a priori, without any further interactions with the environment. Offline RL is particularly pertinent to applications in critical domains such as precision medicine (Gottesman et al., 2019) and autonomous driving (Shalev-Shwartz et al., 2016). In particular, in these scenarios, interacting with the environment via online experiments might be risky, slow, or even possibly unethical. But oftentimes offline datasets consisting of past interactions, e.g., treatment records for precision medicine (Chakraborty and Moodie, 2013; Chakraborty and Murphy, 2014) and human driving data for autonomous driving (Sun et al., 2020), are adequately available. As a result, offline RL has attracted substantial research interest recently (Levine et al., 2020).

Most of the existing works on offline RL develop algorithms and theory on the model of Markov decision processes (MDPs). However, in many real-world applications, due to certain privacy concerns or limitations of the sensor apparatus, the states of the environment cannot be directly stored in the offline datasets. Instead, only partial observations generated from the states of the environments are stored (Dulac-Arnold et al., 2021). For example, in precision medicine, a physician's treatment might consciously or subconsciously depend on the patient's mood and socioeconomic status (Zhang and Bareinboim, 2016), which are not recorded in the data due to privacy concerns. As another example, in autonomous driving, a human driver makes decisions based on multimodal information of the environment that is not limited to visual and auditory inputs, but only observations captured by LIDARs and cameras are stored in the datasets (Sun et al., 2020). In light of the partial observations in the datasets, these situations are better modeled as partially observable Markov decision processes (POMDPs) (Lovejoy, 1991). Existing offline RL methods for MDPs, which fail to handle partial observations, are thus not applicable.

In this work, we make the initial step towards studying offline RL in POMDPs where the datasets only contain partial observations of the states. In particular, motivated from the aforementioned real-world applications, we consider the case where the behavior policy takes actions based on the states of the environment, which are not part of the dataset and thus are latent variables. Instead, the trajectories in datasets consist of partial observations emitted from the latent states, as well as the actions and rewards. For such a dataset, our goal is to learn an optimal policy in the context of general function approximation.

Furthermore, offline RL in POMDP suffers from several challenges. First of all, it is known that both planning and estimation in POMDPs are intractable in the worst case (Papadimitriou and Tsitsiklis, 1987; Burago et al., 1996; Goldsmith and Mundhenk, 1998; Mundhenk et al., 2000; Vlassis et al., 2012). Thus, we have to identify a set of sufficient conditions that warrants efficient offline RL. More importantly, our problem faces the unique challenge of the confounding issue caused by the latent states, which does not appear in either online and offline MDPs or online POMDPs. In particular, both the actions and observations in the offline dataset depend on the unobserved latent states, and thus are confounded (Pearl, 2009).

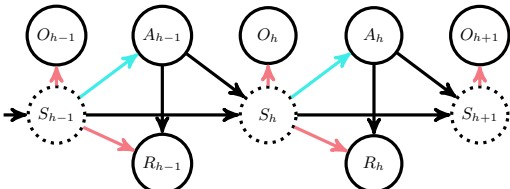

Figure 1: Causal graph of the data generating process for offline RL in POMDP. Here $S_h$ is the state at step $h$. Besides, $A_h$, $R_h$, and $O_h$ are the action, immediate reward, and observation, respectively. The dotted nodes indicate that the variables are not stored in the offline dataset. Solid arrows indicate the dependency among the variables. In specific, the action $A_h$ is specified by the behavior policy which is a function of $S_h$ (**blue** arrows). Moreover, both the observation $O_h$ and reward $R_h$ depend on the state $S_h$ (**red** arrows). We remark that $S_h$ affects both $A_h$ and $(O_h, R_h)$ and thus serves as an **unobserved confounder**.

Such a confounding issue is illustrated by a causal graph in Figure 1. As a result, directly applying offline RL methods for MDPs will nevertheless incur a considerable confounding bias. Besides, since the latent states evolve according to the Markov transition kernel, the causal structure is thus dynamic, which makes the confounding issue more challenging to handle than that in static causal problems. Furthermore, apart from the confounding issue, since we aim to learn the optimal policy, our algorithm also needs to handle the distributional shift between the trajectories induced by the behavior policy and the family of target policies. Finally, to handle large observation spaces, we need to employ powerful function approximators. As a result, the coupled challenges due to (i) the confounding bias, (ii) distributional shift, and (iii) large observation spaces that are distinctive in our problem necessitates new algorithm design and theory.

To this end, by leveraging tools from proximal causal inference (Lipsitch et al., 2010; Tchetgen et al., 2020; Miao et al., 2018b;a), we propose the Proxy variable Pessimistic Policy Optimization (`P3O`) algorithm, which provably addresses the challenge of the confounding bias and the distributional shift in the context of general function approximation. Specifically, we focus on a benign class of POMDPs where the causal structure involving latent states can be captured by the past and current observations, which serves as the negative control action and outcome respectively (Miao et al., 2018a;b; Cui et al., 2020; Singh, 2020; Kallus et al., 2021; Bennett and Kallus, 2021; Shi et al., 2021). Then the value of each policy can be identified by a set of confounding bridge functions corresponding to that policy, which satisfy a sequence of backward moment equations that are similar to the celebrated Bellman equations in classical RL (Bellman and Kalaba, 1965). Thus, by estimating these confounding bridge functions from offline data, we can estimate the value of each policy without incurring the confounding bias.

More concretely, `P3O` involves two components — policy evaluation via minimax estimation and policy optimization via pessimism. Specifically, to tackle the distributional shift, `P3O` returns the policy that maximizes pessimistic estimates of the values obtained by policy evaluation. Meanwhile, in policy evaluation, to ensure pessimism, we construct a coupled sequence of confidence regions for the confounding bridge functions via minimax estimation, using function approximators. Furthermore, under a partial coverage assumption on the confounded dataset, we prove that `P3O` achieves a $\widetilde{\mathcal{O}}(H\sqrt{\log(\mathcal{N}_{\mathrm{fun}})/n})$ suboptimality, where $n$ is the number of trajectories, $H$ is the length of each trajectory, $\mathcal{N}_{\mathrm{fun}}$ stands for the complexity of the employed function classes (e.g., the covering number), and $\widetilde{\mathcal{O}}(\cdot)$ hides logarithmic factors. When specified to linear function classes, the suboptimality of

|  | Offline | Partial Obs. | Confounded Data | Policy Opt. |
|---|:---:|:---:|:---:|:---:|
| Xie et al. (2021) | ✓ | ✗ | ✗ | ✓ |
| Uehara and Sun (2021) | ✓ | ✗ | ✗ | ✓ |
| Jin et al. (2020) | ✗ | ✓ | ✗ | ✓ |
| Efroni et al. (2022) | ✗ | ✓ | ✗ | ✓ |
| Liu et al. (2022) | ✗ | ✓ | ✗ | ✓ |
| Bennett and Kallus (2021) | ✓ | ✓ | ✓ | ✗ |
| Shi et al. (2021) | ✓ | ✓ | ✓ | ✗ |
| P3O (ours) | ✓ | ✓ | ✓ | ✓ |

Table 1: We compare with most related representative works in closely related lines of research. The first line of research studies offline RL in standard MDPs without any partial observability. The second line of research studies online RL in POMDPs where the actions are specified by history-dependent policies. Thus, the actions does not directly depends on the latent states and thus these works do not involve the challenge due to confounded data. The third line of research studies OPE in POMDPs where the goal is to learn the value of the target policy as opposed to learning the optimal policy. As a result, these works do not to need to handle the challenge of distributional shift via pessimism.

P3O becomes $\widetilde{\mathcal{O}}(\sqrt{H^3 d/n})$, where $d$ is the dimension of the feature mapping. To our best knowledge, we establish the first provably efficient offline RL algorithm for POMDP with a confounded dataset.

## 1.1 OVERVIEW OF TECHNIQUES

To deal with the coupled challenges of confounding bias, distributional shift, and large observational spaces, our algorithm and analysis rely on the following technical ingredients.

**Confidence regions based on minimax estimation via proximal causal inference.** To handle the confounded offline dataset, we use the proxy variables from proximal causal inference (Lipsitch et al., 2010; Tchetgen et al., 2020; Miao et al., 2018a;b), which allows us to identify the value of each policy by a set of confounding bridge functions. These bridge functions only depend on observed variables and satisfy a set of backward conditional moment equations. We then estimate these bridge functions via minimax estimation (Dikkala et al., 2020; Chernozhukov et al., 2020; Uehara et al., 2021). More importantly, to handle the distributional shift, we propose a sequence of novel confidence regions for the bridge functions, which quantifies the uncertainty of minimax estimation based on finite data. This sequence of new confidence regions has not been considered in the previous works on off-policy evaluation (OPE) in POMDPs (Bennett and Kallus, 2021; Shi et al., 2021) as pessimism seems unnecessary in these works. Meanwhile, the confidence regions are constructed as a level set w.r.t. the loss functions of the minimax estimation for bridge functions. Such a construction contrasts with previous works on offline RL with confidence regions via maximum likelihood estimation (Uehara and Sun, 2021; Liu et al., 2022) or least square regression (Xie et al., 2021). Furthermore, we develop a novel theoretical analysis to show that any function in the confidence regions enjoys a fast statistical rate of convergence. Finally, leveraging the backwardly inductive nature of the bridge functions, our proposed confidence regions and analysis take the temporal structure into consideration, which might be of independent interest to the study on dynamic causal inference (Friston et al., 2003).

**Pessimism principle for learning POMDPs.** To learn the optimal policy in the face of distributional shift, we adopt the pessimism principle which is shown to be effective in offline RL in MDPs (Liu et al., 2020; Jin et al., 2021; Rashidinejad et al., 2021; Uehara and Sun, 2021; Xie et al., 2021; Yin and Wang, 2021; Zanette et al., 2021; Yin et al., 2022; Yan et al., 2022). Specifically, the newly proposed confidence regions, combined with the identification result based on proximal causal inference, allows us to construct a novel pessimistic estimator for the value of each target policy. From a theoretical perspective, the identification result and the backward induction property of the bridge functions provide a way of decomposing the suboptimality of the learned policy in terms of statistical errors of the bridge functions. When combined with the pessimism and the fast statistical rates enjoyed by any functions in the confidence regions, we show that our proposed P3O algorithm efficiently learns the optimal policy under only a partial coverage assumption of the confounded dataset. We highlight that our work firstly extend the pessimism principle to offline RL in POMDPs with confounded data.

## 1.2 Related Work

Our work is closely related to the bodies of literature on (i) reinforcement learning POMDPs, (ii) offline reinforcement learning (in MDPs), and (iii) OPE via causal inference. Compared to the literature, our work simultaneously involve partial observability, confounded data, and offline policy optimization simultaneously, and thus involves the challenges faced by (i)–(iii). We summarize and contrast with most related existing works in Table 1. We defer the detailed discussion to Appendix B.1.

## 2 Preliminaries

**Notations.** In the sequel, we use lower case letters (i.e., $s$, $a$, $o$, and $\tau$) to represent dummy variables and upper case letters (i.e., $S$, $A$, $O$, and $\Gamma$) to represent random variables. We use the variables in the calligraphic font (i.e., $\mathcal{S}$, $\mathcal{A}$, and $\mathcal{O}$) to represent the spaces of variables, and the blackboard bold font (i.e., $\mathbb{P}$ and $\mathbb{O}$) to represent probability kernels.

### 2.1 Episodic Partially Observable Markov Decision Process

We consider an episodic, finite-horizon POMDP, specified by a tuple $(\mathcal{S}, \mathcal{O}, \mathcal{A}, H, \mu_1, \mathbb{P}, \mathbb{O}, R)$. Here we let $\mathcal{S}$, $\mathcal{A}$, and $\mathcal{O}$ denote the state, action, and observation spaces, respectively. The integer $H \in \mathbb{N}$ denotes the length of each episode. The distribution $\mu_1 \in \Delta(\mathcal{S})$ denotes the distribution of the initial state. The set $\mathbb{P} = \{\mathbb{P}_h\}_{h \in [H]}$ denotes the collection of state transition kernels where each kernel $\mathbb{P}_h(\cdot | s, a) : \mathcal{S} \times \mathcal{A} \mapsto \Delta(\mathcal{S})$ characterizes the distribution of the next state $s_{h+1}$ given that the agent takes action $a_h = a \in \mathcal{A}$ at state $s_h = s \in \mathcal{S}$ and step $h \in [H]$. The set $\mathbb{O} = \{\mathbb{O}_h\}_{h=1}^H$ denotes the observation emission kernels where each kernel $\mathbb{O}_h(\cdot | s) : \mathcal{S} \mapsto \Delta(\mathcal{O})$ characterizes the distribution over observations given the current state $s \in \mathcal{S}$ at step $h \in [H]$. Finally, the set $R = \{R_h\}_{h=1}^H$ denotes the collection of reward functions where each function $R_h(s, a) : \mathcal{S} \times \mathcal{A} \mapsto [0, 1]$ specifies the reward the agent receives when taking action $a \in \mathcal{A}$ at state $s \in \mathcal{S}$ and step $h \in [H]$.

Different from an MDP, in a POMDP, only the observation $o$, the action $a$, and the reward $r$ are observable, while the state variable $s$ is unobservable. In each episode, the environment first samples an initial state $S_1$ from $\mu_1(\cdot)$. At each step $h \in [H]$, the environment emits an observation $O_h$ from $\mathbb{O}_h(\cdot | S_h)$. If an action $A_h$ is taken, then the environment samples the next state $S_{h+1}$ from $\mathbb{P}_h(\cdot | S_h, A_h)$ and assign a reward $R_h$ given by $R_h(S_h, A_h)$. In our setting, we also let $O_0 \in \mathcal{O}$ denote the prior observation before step $h = 1$. We assume that $O_0$ is independent of other random variables in this episode given the first state $S_1$.

### 2.2 Offline Data Generation: Confounded Dataset

Now we describe the data generation process. Motivated by real-world examples such as precision medicine and autonomous driving discussed in Section 1, we assume that the offline data is generated by some behavior policy $\pi^b$ which has access to the latent states. Specifically, we let $\pi^b = \{\pi_h^b\}_{h=1}^H$ denote a collection of policies such that $\pi_h^b(\cdot | s) : \mathcal{S} \mapsto \Delta(\mathcal{A})$ specifies the probability of taking action each $a \in \mathcal{A}$ at state $s$ and step $h$. This behavior policy induces a set of probability distribution $\mathcal{P}^b = \{\mathcal{P}_h^b\}_{h=1}^H$ on the trajectories of the POMDP, where $\mathcal{P}_h^b$ is the distribution of the variables at step $h$ when following the policy $\pi^b$. Formally, we assume that the offline data is denoted by $\mathbb{D} = \{(o_0^k, (o_1^k, a_1^k, r_1^k), \cdots, (o_H^k, a_H^k, r_H^k))\}_{k=1}^n$, where $n$ is the number of trajectories, and for each $k \in [n]$, $(o_0^k, (o_1^k, a_1^k, r_1^k), \cdots, (o_H^k, a_H^k, r_H^k))$ is independently sampled from $\mathcal{P}^b$. We highlight that such an offline dataset is confounded since the latent state $S_h$, which is not stored in the dataset, simultaneously affects the control variables (i.e., action $A_h$) and the outcome variables (i.e., observation $O_h$ and reward $R_h$). Such a setting is prohibitive for existing offline RL algorithms for MDPs as directly applying them will nevertheless incur a confounding bias that is not negligible.

### 2.3 Learning Objective

The goal of offline RL is to learn an optimal policy from the offline dataset which maximizes the expected cumulative reward. For POMDPs, the learned policy can only depend on the observable information mentioned in Section 2.1. To formally define the set policies of interest, we first define the

space of observable history as $\mathcal{H} = \{\mathcal{H}_h\}_{h=0}^{H-1}$, where each element $\tau_h \in \mathcal{H}_h$ is a (partial) trajectory such that $\tau_h \subseteq \{(o_1, a_1), \cdots, (o_h, a_h)\}$. We use $\Gamma_h$ to denote the corresponding random variable.

We denote by $\Pi(\mathcal{H})$ the class of policies which make decisions based on the current observation $o_h \in \mathcal{O}$ and the history information $\tau_{h-1} \in \mathcal{H}_{h-1}$. That means, a policy $\pi = \{\pi_h\}_{h=1}^{H} \in \Pi(\mathcal{H})$ satisfies $\pi_h(\cdot | o, \tau) : \mathcal{O} \times \mathcal{H}_{h-1} \mapsto \Delta(\mathcal{A})$. The choice of $\mathcal{H}$ induces the policy set $\Pi(\mathcal{H})$ by specifying the input of the policies. We now introduce three examples of $\mathcal{H}$ and the corresponding $\Pi(\mathcal{H})$.

**Example 2.1** (Reactive policy (Azizzadenesheli et al., 2018))**.** *The policy only depends on the current observation $O_h$. Formally, we have $\mathcal{H}_{h-1} = \{\varnothing\}$ and therefore $\tau_{h-1} = \varnothing$ for each $h \in [H]$.*

**Example 2.2** (Finite-history policy (Efroni et al., 2022))**.** *The policy depends on the current observation and the history of length at most $k$. Formally, we have $\mathcal{H}_{h-1} = (\mathcal{O} \times \mathcal{A})^{\otimes \min\{k, h-1\}}$ and $\tau_{h-1} = ((o_l, a_l), \cdots, (o_{h-1}, a_{h-1}))$ for some $k \in \mathbb{N}$, where the index $l = \max\{1, h - k\}$.*

**Example 2.3** (Full-history policy (Liu et al., 2022))**.** *The policy depends on the current observation and the full history. Formally, $\mathcal{H}_{h-1} = (\mathcal{O} \times \mathcal{A})^{\otimes(h-1)}$, and $\tau_{h-1} = ((o_1, a_1), \cdots, (o_{h-1}, a_{h-1}))$.*

We illustrate these examples with causal graphs and a more detailed discussion in Appendix C.1. Now given a policy $\pi \in \Pi(\mathcal{H})$, we denote by $J(\pi)$ the value of $\pi$ that characterizes the expected cumulative rewards the agent receives by following $\pi$. Formally, $J(\pi)$ is defined as

$$J(\pi) := \mathbb{E}_\pi \left[ \sum_{h=1}^{H} \gamma^{h-1} R_h(S_h, A_h) \right], \tag{2.1}$$

where $\gamma \in (0, 1]$ is the discount factor, $\mathbb{E}_\pi$ denotes the expectation w.r.t. $\mathcal{P}^\pi = \{\mathcal{P}_h^\pi\}_{h=1}^{H}$ which is the distribution of the trajectories induced by $\pi$. We define the suboptimality gap of any policy $\widehat{\pi}$ as

$$\text{SubOpt}(\widehat{\pi}) := J(\pi^\star) - J(\widehat{\pi}), \quad \text{where} \quad \pi^\star \in \arg\max_{\pi \in \Pi(\mathcal{H})} J(\pi). \tag{2.2}$$

Here $\pi^\star$ is the optimal policy within $\Pi(\mathcal{H})$. Our goal is to find some policy $\widehat{\pi} \in \Pi(\mathcal{H})$ that minimizes the suboptimality gap in (2.2) based on the offline dataset $\mathbb{D}$.

## 3 ALGORITHM: PROXY VARIABLE PESSIMISTIC POLICY OPTIMIZATION

As is previously discussed, the offline RL problem introduced in Section 2 for POMDPs suffers from three coupled challenges — (i) the confounding bias, (ii) distributional shift, and (iii) large observation spaces. In the sequel, we introduce an algorithm that addresses all three challenges simultaneously. We first introduce the high-level idea for combating these challenges.

Offline RL for POMDPs is known to be intractable in the worst case (Krishnamurthy et al., 2016). So we first identify a benign class of POMDPs where the causal structure involving latent states can be captured by only the observable variables available in the dataset $\mathbb{D}$. For such a class of POMDPs, by leveraging tools from proximal causal inference (Lipsitch et al., 2010; Tchetgen et al., 2020), we then seek to identify the value $J(\pi)$ of the policy $\pi \in \Pi(\mathcal{H})$ via some confounding bridge functions $\mathbf{b}^\pi$ (Assumption 3.2) which only depend on the observable variables and thus can be estimated using $\mathbb{D}$ (Theorem 3.3). Identification via proximal causal inference will be discussed in Section 3.1.

In addition, to estimate these confounding bridge functions, we utilize the the fact that these functions satisfy a sequence of conditional moment equations which resembles the Bellman equations in classical MDPs (Bellman and Kalaba, 1965). Then we adopt the idea of minimax estimation (Dikkala et al., 2020; Kallus et al., 2021; Uehara et al., 2021; Duan et al., 2021) which formulates the bridge functions as the solution to a series of minimax optimization problems in (3.11). Additionally, the loss function in minimax estimation readily incorporates function approximators and thus addresses the challenge of large observation spaces.

To further handle the distributional shift, we extend the pessimism principle (Liu et al., 2020; Jin et al., 2021; Rashidinejad et al., 2021; Uehara and Sun, 2021; Xie et al., 2021; Yin and Wang, 2021; Zanette et al., 2021) to POMDPs with the help of the confounding bridge functions. In specific, based on the confounded dataset, we first construct a novel confidence region $\text{CR}^\pi(\xi)$ for $\mathbf{b}^\pi$ based on level sets with respect to the loss functions of the minimax estimation (See (3.12) for details). Our algorithm, Proxy variable Pessimistic Policy Optimization (P3O), outputs the policy that maximizes pessimistic estimates the values of the policies within $\Pi(\mathcal{H})$. The details of P3O is summarized by Algorithm 1 in Section 3.3.

### 3.1 POLICY VALUE IDENTIFICATION VIA PROXIMAL CAUSAL INFERENCE

To handle the confounded dataset $\mathbb{D}$, we first identify the policy value $J(\pi)$ for each $\pi \in \Pi(\mathcal{H})$ using the idea of proxy variables. Following the notions of proximal causal inference (Tennenholtz et al., 2020; Lipsitch et al., 2010), we assume that there exists negative control actions $\{Z_h\}_{h=1}^H$ and negative control outcomes $\{W_h\}_{h=1}^H$ satisfying the following independence assumption.

**Assumption 3.1** (Negative control). *We assume there exist negative control variables $\{W_h\}_{h=1}^H$ and $\{Z_h\}_{h=1}^H$ measurable with respect to the observed trajectories, such that under $\mathcal{P}^b$, it holds that*

$$Z_h \perp O_h, R_h, W_h, W_{h+1} \,|\, A_h, S_h, \Gamma_{h-1}, \quad W_h \perp A_h, \Gamma_{h-1}, S_{h-1} \,|\, S_h. \tag{3.1}$$

We explain in detail the existence of such negative control variables for all the three different choices of history $\mathcal{H}$ in Examples 2.1, 2.2, and 2.3 respectively in Appendix C.1. Besides Assumption 3.1, our identification of policy value also relies on the notion of confounding bridge functions (Kallus et al., 2021; Shi et al., 2021), for which we make the following assumption.

**Assumption 3.2** (Confounding bridge functions). *For any history-dependent policy $\pi \in \Pi(\mathcal{H})$, we assume the existence of the value bridge functions $\{b_h^\pi : \mathcal{A} \times \mathcal{W} \mapsto \mathbb{R}\}_{h=1}^H$ and the weight bridge functions $\{q_h^\pi : \mathcal{A} \times \mathcal{Z} \mapsto \mathbb{R}\}_{h=1}^H$ which are defined as the solution to the following equations almost surely with respect to the measure $\mathcal{P}^b$:*

$$\mathbb{E}_{\pi^b}\left[b_h^\pi(A_h, W_h)|A_h, Z_h\right] = \mathbb{E}_{\pi^b}\Big[R_h \pi_h(A_h|O_h, \Gamma_{h-1})$$
$$+ \gamma \sum_{a'} b_{h+1}^\pi(a', W_{h+1})\pi_h(A_h|O_h, \Gamma_{h-1})\Big|A_h, Z_h\Big], \tag{3.2}$$

$$\mathbb{E}_{\pi^b}\left[q_h^\pi(A_h, Z_h)|A_h, S_h, \Gamma_{h-1}\right] = \frac{\mu_h(S_h, \Gamma_{h-1})}{\pi_h^b(A_h|S_h)}. \tag{3.3}$$

*Here $b_{H+1}^\pi$ is a zero function and $\mu_h(S_h, \Gamma_{h-1})$ in (3.3) is defined as the importance sampling ratio $\mu_h(S_h, \Gamma_{h-1}) := \mathcal{P}_h^\pi(S_h, \Gamma_{h-1})/\mathcal{P}_h^b(S_h, \Gamma_{h-1})$.*

We use "confounding bridge function" and "bridge function" interchangeably throughout the paper. We remark that in the proximal causal inference literature, the existence of such bridge functions bears more generality than assuming certain complicated completeness conditions (Cui et al., 2020), as discussed by Kallus et al. (2021). The existence of such bridge functions is justified by conditions on the the rank of certain conditional probabilities or singular values of certain conditional expectation linear operators. See Appendix C.2 for examples of Assmuption 3.2 under previous choice of $\mathcal{H}$.

Now given Assumption 3.1 and 3.2 on the existence of proxy variables and bridge functions, we are ready to present the main identification result. It represents the true policy value $J(\pi)$ via the value bridge functions, as is concluded in the following theorem.

**Theorem 3.3** (Identification of policy value). *Under Assumption 3.1 and 3.2, for any history-dependent policy $\pi \in \Pi(\mathcal{H})$, it holds that*

$$J(\pi) = F(\mathbf{b}^\pi), \quad \text{where} \quad F(\mathbf{b}^\pi) := \mathbb{E}_{\pi^b}\left[\sum_{a \in \mathcal{A}} b_1^\pi(a, W_1)\right]. \tag{3.4}$$

See Appendix E for a detailed proof. Note that although we have assumed the existence of both the value bridge functions in (3.2) and the weight bridge functions in (3.3), Theorem 3.3 represents $J(\pi)$ using only the value bridge functions. In (3.2) all the random variables involved are observed by the learner and distributed according to the data distribution $\mathcal{P}^b$, which means that the value bridge functions can be estimated from data. This overcomes the confounding issue.

### 3.2 POLICY EVALUATION VIA MINIMAX ESTIMATION WITH UNCERTAINTY QUANTIFICATION

According to Theorem 3.3 and Assumption 3.2, to estimate the value $J(\pi)$ of $\pi \in \Pi(\mathcal{H})$, it suffices to estimate the value bridge functions $\{b_h^\pi\}_{h=1}^H$ by solving (3.2), which is a conditional moment equation. To this end, we adopt the method of minimax estimation (Dikkala et al., 2020; Uehara et al., 2021; Duan et al., 2021). Furthermore, in order to handle the distributional shift between behavior

policy and target policies, we construct a sequence of confidence regions for $\{b_h^\pi\}_{h=1}^H$ based on minimax estimation, which allows us to apply the pessimism principle by finding the most pessimistic estimates within the confidence regions.

Specifically, minimax estimation involves two function classes $\mathbb{B} \subseteq \{b : \mathcal{A} \times \mathcal{W} \mapsto \mathbb{R}\}$ and $\mathbb{G} \subseteq \{g : \mathcal{A} \times \mathcal{Z} \mapsto \mathbb{R}\}$, interpreted as the primal and dual function classes, respectively. Theoretical assumptions on $\mathbb{B}$ and $\mathbb{G}$ are presented in Section 4. In order to find functions that satisfy (3.2), it suffices to find $\mathbf{b} = (b_1, \cdots, b_H) \in \mathbb{B}^{\otimes H}$ such that the following conditional moment

$$
\begin{aligned}
\ell_h^\pi(b_h, b_{h+1})(A_h, Z_h) := \mathbb{E}_{\pi^b}\Big[ & b_h(A_h, W_h) - R_h \pi_h(A_h | O_h, \Gamma_{h-1}) \\
& - \gamma \sum_{a' \in \mathcal{A}} b_{h+1}(a', W_{h+1}) \pi_h(A_h | O_h, \Gamma_{h-1}) \Big| A_h, Z_h \Big]
\end{aligned}
\tag{3.5}
$$

is equal to zero almost surely for all $h \in [H]$, where $b_{H+1}$ is a zero function. Intuitively, the quantity (3.5) can be interpreted as the "Bellman residual" of the value bridge functions $\mathbf{b}$. Notice that (3.5) being zero almost surely for all $h \in [H]$ is actually a conditional moment equation, which is equivalent to finding $\mathbf{b} \in \mathbb{B}^{\otimes H}$ such that the following residual mean squared error (RMSE) is minimized for all $h$:

$$
\mathcal{L}_h^\pi(b_h, b_{h+1}) := \mathbb{E}_{\pi^b}\big[ \big( \ell_h^\pi(b_h, b_{h+1})(A_h, Z_h) \big)^2 \big].
\tag{3.6}
$$

It might seem tempting to directly minimize the empirical version of (3.6). However, this is not viable as one would obtain a biased estimator due to an additional variance term. The reason is that the quantity defined by (3.5) is a conditional expectation and therefore RMSE defined by (3.6) cannot be directly unbiasedly estimated from data (Farahmand et al., 2016). In the sequel, we adopt the technique of minimax estimation to circumvent this issue. In particular, we first use Fenchel duality to write (3.6) as

$$
\mathcal{L}_h^\pi(b_h, b_{h+1}) = 4\lambda \mathbb{E}_{\pi^b}\left[ \max_{g \in \mathbb{G}} \ell_h^\pi(b_h, b_{h+1})(A_h, Z_h) \cdot g(A_h, Z_h) - \lambda g(A_h, Z_h)^2 \right], \ \lambda > 0,
\tag{3.7}
$$

which holds when the dual function class $\mathbb{G}$ is expressive enough such that $\ell_h^\pi(b_h, b_{h+1})/2\lambda \in \mathbb{G}$. Then thanks to the interchangeability principle (Rockafellar and Wets, 2009; Dai et al., 2017; Shapiro et al., 2021), we can interchange the order of maximization and expectation and derive that

$$
\mathcal{L}_h^\pi(b_h, b_{h+1}) = 4\lambda \max_{g \in \mathbb{G}} \mathbb{E}_{\pi^b}\left[ \ell_h^\pi(b_h, b_{h+1})(A_h, Z_h) \cdot g(A_h, Z_h) - \lambda g(A_h, Z_h)^2 \right].
\tag{3.8}
$$

The core idea of minimax estimation is to minimize the empirical version of (3.8) instead of (3.6), and the benefit of doing so is a fast statistical rate of $\widetilde{\mathcal{O}}(n^{-1/2})$ (Dikkala et al., 2020; Uehara et al., 2021), as we can see in the sequel. For simplicity, in the following, we define $\Phi_{\pi,h}^\lambda : \mathbb{B} \times \mathbb{B} \times \mathbb{G} \mapsto \mathbb{R}$ with parameter $\lambda > 0$ as

$$
\Phi_{\pi,h}^\lambda(b_h, b_{h+1}; g) := \mathbb{E}_{\pi^b}\big[ \ell_h^\pi(b_h, b_{h+1})(A_h, Z_h) \cdot g(A_h, Z_h) - \lambda g(A_h, Z_h)^2 \big],
\tag{3.9}
$$

and we denote by $\widehat{\Phi}_{\pi,h}^\lambda : \mathbb{B} \times \mathbb{B} \times \mathbb{G} \mapsto \mathbb{R}$ the empirical version of $\Phi_{\pi,h}^\lambda$, i.e.,

$$
\begin{aligned}
\widehat{\Phi}_{\pi,h}^\lambda(b_h, b_{h+1}; g) := \widehat{\mathbb{E}}_{\pi^b}\Big[ \Big( & b_h(A_h, W_h) - R_h \pi_h(A_h | O_h, \Gamma_{h-1}) \\
& - \gamma \sum_{a' \in \mathcal{A}} b_{h+1}(a', W_{h+1}) \pi_h(A_h | O_h, \Gamma_{h-1}) \Big) \cdot g(A_h, Z_h) - \lambda g(A_h, Z_h)^2 \Big],
\end{aligned}
\tag{3.10}
$$

where $\widehat{\mathbb{E}}_{\pi^b}$ denotes the empirical version of $\mathbb{E}_{\pi^b}$ based on dataset $\mathbb{D}$ described in Section 2.2.

Furthermore, note that the value bridge functions $(b_1^\pi, \cdots, b_h^\pi)$ admit a sequential dependence structure. To handle such dependency, for any $\pi \in \Pi(\mathcal{H})$, $h \in [H]$, and $b_{h+1} \in \mathbb{B}$, we first define the minimax estimator $\widehat{b}_h(b_{h+1})$ as

$$
\widehat{b}_h(b_{h+1}) := \arg\min_{b \in \mathbb{B}} \Big\{ \max_{g \in \mathbb{G}} \widehat{\Phi}_{\pi,h}^\lambda(b_h, b_{h+1}; g) \Big\}.
\tag{3.11}
$$

Based on (3.11), we propose a confidence region for $\mathbf{b}^\pi := (b_1^\pi, \cdots, b_H^\pi) \in \mathbb{B}^{\otimes H}$ as

$$
\mathrm{CR}^\pi(\xi) := \left\{ \mathbf{b} \in \mathbb{B}^{\otimes H} \,\Big|\, \max_{g \in \mathbb{G}} \widehat{\Phi}_{\pi,h}^\lambda(b_h, b_{h+1}; g) - \max_{g \in \mathbb{G}} \widehat{\Phi}_{\pi,h}^\lambda(\widehat{b}_h(b_{h+1}), b_{h+1}; g) \leq \xi, \forall h \right\}.
\tag{3.12}
$$

---

**Algorithm 1** Proxy variable Pessimistic Policy Optimization (P3O)

1: **Input**: confidence parameter $\xi > 0$, regularization parameter $\lambda > 0$, dataset $\mathbb{D}$, classes $\mathbb{B}$ and $\mathbb{G}$.
2: Construct minimax estimator confidence region $\mathrm{CR}^\pi(\xi)$ for each $\pi \in \Pi(\mathcal{H})$ by (3.12).
3: **Policy evaluation:** pessimistically estimate $\widehat{J}_{\mathrm{Pess}}(\pi)$ for each $\pi \in \Pi(\mathcal{H})$ by (3.13).
4: **Policy optimization:** set $\widehat{\pi}$ by (3.14).
5: **Output**: $\widehat{\pi} = \{\widehat{\pi}_h\}_{h=1}^H$.

---

From the above definition[1], one can see that $\mathrm{CR}^\pi(\xi)$ is actually a coupled sequence of $H$ confidence regions, where each single confidence region aims to cover a function $b_h^\pi$. For notational simplicity, we use a single notation $\mathrm{CR}^\pi(\xi)$ to denote all the $H$ confidence regions. Intuitively, the confidence region $\mathrm{CR}^\pi(\xi)$ contains all $\mathbf{b} \in \mathbb{B}^{\otimes H}$ whose RMSE does not exceed that of $(\widehat{b}_h(b_{h+1}),\ b_{h+1})$ by too much at each $h \in [H]$. The confidence region takes the sequential dependence of confounding bridge functions into consideration in the sense that each $\mathbf{b} \in \mathrm{CR}^\pi(\xi)$ is restricted through the minimax estimation loss between continuous steps. As we show in Section D, with high probability, the confidence region $\mathrm{CR}^\pi(\xi)$ contains the true bridge value functions $\mathbf{b}^\pi$. More importantly, every $\mathbf{b} \in \mathrm{CR}^\pi(\xi)$ enjoys a fast statistical rate of $\widetilde{\mathcal{O}}(n^{-1/2})$.

Now combining the confidence region (3.12) and the identification formula (3.4), for any policy $\pi \in \Pi(\mathcal{H})$, we adopt an pessimistic estimate of the value of $J(\pi)$ as

$$\widehat{J}_{\mathrm{Pess}}(\pi) := \min_{\mathbf{b} \in \mathrm{CR}^\pi(\xi)} \widehat{F}(\mathbf{b}), \quad \text{where} \ \ \widehat{F}(\mathbf{b}) := \widehat{\mathbb{E}}_{\pi^b}\left[\sum_{a \in \mathcal{A}} b_1(a, W_1)\right]. \tag{3.13}$$

### 3.3 POLICY OPTIMIZATION

Given the pessimistic value estimate (3.13), P3O chooses $\widehat{\pi}$ which maximizes $\widehat{J}_{\mathrm{Pess}}(\pi)$, that is,

$$\widehat{\pi} := \arg\max_{\pi \in \Pi(\mathcal{H})} \widehat{J}_{\mathrm{Pess}}(\pi). \tag{3.14}$$

We summarize the entire P3O algorithm in Algorithm 1. In Section 4, we show that under some mild assumptions on the function classes $\mathbb{B}$ and $\mathbb{G}$ and under only a partial coverage assumption on the dataset $\mathbb{D}$, the suboptimality (2.2) of Algorithm 1 decays at the fast statistical rate of $\widetilde{\mathcal{O}}(n^{-1/2})$, where $\widetilde{\mathcal{O}}(\cdot)$ omits $H$ and factors that characterize the complexity of the function classes.

## 4 THEORETICAL RESULTS

In this section, we present our theoretical results. For ease of presentation, we first assume that both the primal function class $\mathbb{B}$ and the policy class $\Pi(\mathcal{H})$ are finite sets with cardinality $|\mathbb{B}|$ and $|\Pi(\mathcal{H})|$, respectively. But we allow the dual function class $\mathbb{G}$ to be an infinite set. Our results can be easily extended to infinite $\mathbb{B}$ and $\Pi(\mathcal{H})$ using the notion of covering numbers (Wainwright, 2019), which we demonstrate with linear function approximation in Section H.1.

We first introduce some necessary assumptions for efficient learning of the optimal policy. To begin with, the following Assumption 4.1 ensures that the offline data generated by $\pi^b$ has a good coverage over $\pi^\star$. The problem would become intractable without such an assumption (Chen and Jiang, 2019).

**Assumption 4.1** (Partial coverage). *We assume that the concentrability coefficient for the optimal policy $\pi^\star$, defined as $C^{\pi^\star} := \max_{h \in [H]} \mathbb{E}_{\pi^b}\left[(q_h^{\pi^\star}(A_h, Z_h))^2\right]$, satisfies that $C^{\pi^\star} < +\infty$.*

Very importantly, Assumption 4.1 only assumes the partial coverage, i.e., the optimal policy $\pi^\star$ is well covered by $\pi^b$ (Jin et al., 2021; Uehara and Sun, 2021), which is significantly weaker than the uniform coverage, i.e., the entire policy class $\Pi(\mathcal{H})$ is covered by $\pi^b$ (Chen and Jiang, 2019) in the sense that $\max_{\pi \in \Pi(\mathcal{H})} C^\pi < +\infty$. See Appendix B.1 for more about partial coverage in POMDP.

The next assumption is on the functions classes $\mathbb{B}$ and $\mathbb{G}$. We require that $\mathbb{B}$ and $\mathbb{G}$ are uniformly bounded, and that $\mathbb{G}$ is symmetric, star-shaped, and has bounded localized Rademacher complexity.

---

[1]We refer the readers to Appendix B.1 for a detailed comparison of the minimax-typed loss (and confidence region) with the least-square-typed loss (and confidence region) used by Xie et al. (2021).

**Assumption 4.2** (Function classes $\mathbb{B}$ and $\mathbb{G}$). *We assume the classes $\mathbb{B}$ and $\mathbb{G}$ satisfy that: i) There exist $M_{\mathbb{B}}, M_{\mathbb{G}} < +\infty$ such that $\mathbb{B}$, $\mathbb{G}$ are bounded by $\sup_{b \in \mathbb{B}} \sup_{w \in \mathcal{W}} |\sum_{a \in \mathcal{A}} b(a, w)| \leq M_{\mathbb{B}}$[2], $\sup_{g \in \mathbb{G}} \sup_{(a,z) \in \mathcal{A} \times \mathcal{Z}} |g(a, z)| \leq M_{\mathbb{G}}$; ii) $\mathbb{G}$ is star-shaped, i.e., for any $g \in \mathbb{G}$ and $\lambda \in [0, 1]$, it holds that $\lambda g \in \mathbb{G}$; iii) $\mathbb{G}$ is symmetric, i.e., for any $g \in \mathbb{G}$, it holds that $-g \in \mathbb{G}$; iv) For any step $h \in [H]$, $\mathbb{G}$ has bounded critical radius $\alpha_{\mathbb{G},h,n}$ which solves inequality $\mathcal{R}_n(\mathbb{G}; \alpha) \leq \alpha^2 / M_{\mathbb{G}}$, where $\mathcal{R}_n(\mathbb{G}, \alpha)$ is the localized population Rademacher complexity of $\mathbb{G}$ under the distribution of $(A_h, Z_h)$ induced by $\pi^b$, that is,*

$$\mathcal{R}_n(\mathbb{G}, \alpha) = \mathbb{E}_{\pi^b, \epsilon_i} \left[ \sup_{g \in \mathbb{G}: \|g\|_2 \leq \alpha} \left| \frac{1}{n} \sum_{i=1}^{n} \epsilon_i g(A_h, Z_h) \right| \right],$$

*with $\|g\|_2$ defined as $(\mathbb{E}_{\pi^b}[g^2(A_h, Z_h)])^{1/2}$, random variables $\{\epsilon_i\}_{i=1}^{n}$ independent of $(A_h, Z_h)$ and independently uniformly distributed on $\{+1, -1\}$. Also, we denote $\alpha_{\mathbb{G},n} := \max_{h \in [H]} \alpha_{\mathbb{G},h,n}$.*

Finally, to ensure that the minimax estimation in Section 3.2 learns the value bridge functions unbiasedly, we make the following completeness and realizability assumptions on the function classes $\mathbb{B}$ and $\mathbb{G}$ which are standard in the literature (Dikkala et al., 2020; Xie et al., 2021; Shi et al., 2021).

**Assumption 4.3** (Completeness and realizability). *We assume that, i) completeness: for any $h \in [H]$, any $\pi \in \Pi(\mathcal{H})$, and any $b_h, b_{h+1} \in \mathbb{B}$, it holds that $\frac{1}{2\lambda} \ell_h^\pi(b_h, b_{h+1}) \in \mathbb{G}$ where $\ell_h^\pi$ is defined in (3.5); ii) realizability: for any $h \in [H]$, any $\pi \in \Pi(\mathcal{H})$, and any $b_{h+1} \in \mathbb{B}$, there exists $b^\star \in \mathbb{B}$ such that $\mathcal{L}_h^\pi(b^\star, b_{h+1}) \leq \epsilon_{\mathbb{B}}$ for some $\epsilon_{\mathbb{B}} < +\infty$, i.e., we assume that*

$$0 \leq \epsilon_{\mathbb{B}} := \max_{h \in [H], \pi \in \Pi(\mathcal{H}), b_{h+1} \in \mathbb{B}} \min_{b_h \in \mathbb{B}} \mathbb{E}_{\pi^b} \left[ \ell_h^\pi(b_h, b_{h+1})(A_h, Z_h)^2 \right] < +\infty.$$

Here the completeness assumption means the dual function class $\mathbb{G}$ is rich enough which guarantees the equivalence between $\mathcal{L}_h^\pi(\cdot, \cdot)$ and $\max_{g \in \mathbb{G}} \Phi_{\pi,h}^\lambda(\cdot, \cdot; g)$. The realizability assumption means that the primal function class $\mathbb{B}$ is rich enough such that (3.2) always admits an (approximate) solution.

With these technical assumptions, we can establish our main theoretical results in the following theorem, which gives an upper bound of the suboptimality (2.2) of the policy $\widehat{\pi}$ output by Algorithm 1.

**Theorem 4.4** (Suboptimality). *Under Assumptions 3.1, 3.2, 4.1, 4.2, and 4.3, by setting the regularization parameter $\lambda$ and the confidence parameter $\xi$ as $\lambda = 1$ and*

$$\xi = C_1 \cdot M_{\mathbb{B}}^2 M_{\mathbb{G}}^2 \cdot \log(|\mathbb{B}||\Pi(\mathcal{H})|H/\zeta)/n,$$

*then probability at least $1 - 3\delta$, it holds that*

$$\text{SubOpt}(\widehat{\pi}) \leq C_1' \sqrt{C^{\pi^\star}} H \cdot M_{\mathbb{B}} M_{\mathbb{G}} \cdot \sqrt{\log(|\mathbb{B}||\Pi(\mathcal{H})|H/\zeta)/n} + C_1' \sqrt{C^{\pi^\star} M_{\mathbb{G}}} H \epsilon_{\mathbb{B}}^{1/4},$$

*where $\zeta = \min\{\delta, 4c_1 \exp(-c_2 n \alpha_{\mathbb{G},n}^2)\}$. Here $C^{\pi^\star}$, $\alpha_{\mathbb{G},n}$, and $\epsilon_{\mathbb{B}}$ are defined in Assumption 4.1, 4.2, and 4.3, respectively. And $C_1$, $C_1'$, $c_1$, and $c_2$ are some problem-independent universal constants.*

We introduce all the key technical lemmas and sketch the proof of Theorem 4.4 in Section D. We refer to Appendix G for a detailed proof. When it holds that $\alpha_{\mathbb{G},n} = \widetilde{\mathcal{O}}(n^{-1/2})$ and $\epsilon_{\mathbb{B}} = 0$, Theorem 4.4 implies that $\text{SubOpt}(\widehat{\pi}) \leq \widetilde{\mathcal{O}}(n^{-1/2})$, which corresponds to a "fast statistical rate" for minimax estimation (Uehara et al., 2021). The derivation of such a fast rate relies on a novel analysis for the risk of functions in the confidence region, which is shown by Lemma D.3 in Section D. Meanwhile, for many choices of the dual function class $\mathbb{G}$, it holds that $\alpha_{\mathbb{G},n}$ scales with $\sqrt{\log \mathcal{N}_{\mathbb{G}}}$ where $\mathcal{N}_{\mathbb{G}}$ denotes the complexity measure of the class $\mathbb{G}$. In such cases, the suboptimality also scales with $\sqrt{\log \mathcal{N}_{\mathbb{G}}}$, without explicit dependence on the cardinality of the spaces $\mathcal{S}$, $\mathcal{A}$, or $\mathcal{O}$. Finally, we highlight that, thanks to the principle of pessimism, the suboptimality of P3O depends only on the partial coverage concentrability coefficient $C^{\pi^\star}$, which can be significantly smaller than the uniform coverage concentrability coefficient $\sup_{\pi \in \Pi(\mathcal{H})} C^\pi$. In conclusion, when $\alpha_{\mathbb{G},n} = \widetilde{\mathcal{O}}(\sqrt{\log \mathcal{N}_{\mathbb{G}}/n})$ and $\epsilon_{\mathbb{B}} = 0$, the P3O algorithm enjoys a $\widetilde{\mathcal{O}}(H\sqrt{C^{\pi^\star} \log \mathcal{N}_{\mathbb{G}}/n})$ suboptimality.

**Linear function approximation.** Theorem 4.4 can be readily extended to the case of linear function approximation (LFA) with infinite-cardinality $\mathbb{B}$ and $\Pi(\mathcal{H})$, which yields an $\widetilde{\mathcal{O}}(\sqrt{H^3 d/n})$ suboptimality. Due to space limit, we defer the detailed setup and main results of LFA to Appendix H.1.

---

[2]The constant $M_{\mathbb{B}}$ might seem to be proportional to $|\mathcal{A}|$ due to the summation $\sum_{a \in \mathcal{A}} b(a, w)$, but it is not. The reason is that the definition of the true confounding bridge function in (3.2) involves a product with $\pi_h(\cdot|O_h, \Gamma_{h-1})$ which is a distribution over $\mathcal{A}$. Thus the summation over $\mathcal{A}$ is essentially an average over $\mathcal{A}$.

ACKNOWLEDGEMENTS

Zhaoran Wang acknowledges National Science Foundation (Awards 2048075, 2008827, 2015568, 1934931), Simons Institute (Theory of Reinforcement Learning), Amazon, J.P. Morgan, and Two Sigma for their support. Zhuoran Yang acknowledges Simons Institute (Theory of Reinforcement Learning) for the support. The authors would like to thank Zhihan Liu for helpful discussions on minimax estimators.

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

CONTENTS

## A TABLE OF NOTATIONS

Table 2: Table of Notations

| Notation | Meaning |
|:---:|:---:|
| $J(\pi)$ | Policy value $\mathbb{E}_\pi[\sum_{h=1}^H \gamma^{h-1} R_h]$ |
| $b_h^\pi, q_h^\pi$ | value bridge function, weight bridge function of $\pi$ at step $h$ |
| $\mathbf{b}^\pi, \mathbf{q}^\pi$ | value bridge function vector, weight bridge function vector of $\pi$ |
| $\mathrm{CR}^\pi(\xi)$ | confidence region of $\mathbf{b}^\pi$, according to (3.12) |
| $\mathbf{b}$ | an element in the confidence region $\mathrm{CR}^\pi(\xi)$ |
| $F(\mathbf{b}), \widehat{F}(\mathbf{b})$ | a mapping for identification with $J(\pi) = F(\mathbf{b}^\pi)$, according to (3.4) |
| $\ell_h^\pi$ | "Bellman residual" for bridge functions, according to (3.5) |
| $\mathcal{L}_h^\pi$ | residual mean square loss for $\ell_h^\pi$, according to (3.6) |
| $\Phi_{\pi,h}^\lambda, \widehat{\Phi}_{\pi,h}^\lambda$ | a mapping for minimax estimation, according to (3.9) |
| $\widehat{b}_h(b_{h+1})$ | minimax estimator of $b_h^\pi$ given $b_{h+1}$, according to (3.11) |
| $\widehat{J}_{\mathrm{pess}}(\pi)$ | pessimistic estimator of $J(\pi)$, according to (3.13) |
| $\widehat{\pi}$ | policy returned by P3O algorithm, according to (3.14) |

In this section, we provide a comprehensive clarification on the use of notation in this paper.

We use lower case letters (i.e., $s$, $a$, $o$, and $\tau$) to represent dummy variables and upper case letters (i.e., $S$, $A$, $O$, and $\Gamma$) to represent random variables. We use the variables in the calligraphic font (i.e., $\mathcal{S}$, $\mathcal{A}$, $\mathcal{O}$, and $\mathcal{H}$) to represent the spaces of variables, and the blackboard bold font (i.e., $\mathbb{P}$ and $\mathbb{O}$) to represent probability kernels.

We use $\mathcal{H} = \{\mathcal{H}_h\}_{h=0}^{H-1}$ to denote the space of observable history, where each element $\tau_h \in \mathcal{H}_h$ is a (partial) trajectory such that $\tau_h \subseteq \{(o_1, a_1), \cdots, (o_h, a_h)\}$. We use $\pi^b = \{\pi_h^b\}_{h=1}^H$ to denote the behavior policy, where $\pi_h^b : \mathcal{S} \mapsto \Delta(\mathcal{A})$. We use $\pi = \{\pi_h\}_{h=1}^H \in \Pi(\mathcal{H})$ to denote a history-dependent policy with $\pi_h : \mathcal{O} \times \mathcal{H}_{h-1} \mapsto \Delta(\mathcal{A})$. Also, we use $\pi^\star = \{\pi_h^\star\}_{h=1}^H$ to denote the optimal history-dependent policy. Offline data $\mathbb{D}$ is collected by $\pi^b$, as described in Section 2.2.

We use $\mathcal{P}^b = \{\mathcal{P}_h^b\}_{h=1}^H$ and $\mathcal{P}^\pi = \{\mathcal{P}_h^\pi\}_{h=1}^H$ to denote the distribution of trajectories under the policy $\pi^b$ and $\pi$, respectively, where $\mathcal{P}_h^b$ and $\mathcal{P}_h^\pi$ denote the density of corresponding variables at step $h$. Also, we use $\mathbb{E}_{\pi^b}$ and $\mathbb{E}_\pi$ to denote the expectation w.r.t. the distribution $\mathcal{P}^b$ and $\mathcal{P}^\pi$. We use $\widehat{\mathbb{E}}_{\pi^b}$ to denote the empirical version of $\mathbb{E}_{\pi^b}$, which is calculated on data $\mathbb{D}$.

Through out the paper, we use $\mathcal{O}(\cdot)$ to hide problem-independent constants and use $\widetilde{\mathcal{O}}(\cdot)$ to hide problem-independent constants plus logarithm factors. The following table summaries the notations we used in our proposed algorithm design and theory.

## B  FURTHER DISCUSSION

### B.1  FURTHER DISCUSSION ON RELATED WORK

**Reinforcement learning in POMDPs.** Our work is related to the recent line of research on developing provably efficient online RL methods for POMDPs (Guo et al., 2016; Krishnamurthy et al., 2016; Jin et al., 2020; Xiong et al., 2021; Jafarnia-Jahromi et al., 2021; Efroni et al., 2022; Liu et al., 2022). In the online setting, the actions are specified by history-dependent policies and thus the latent state does not directly affect the actions. Thus, the actions and observations in the online setting are not confounded by latent states. Consequently, although these work also conduct uncertainty quantification to encourage exploration, the confidence regions are not based on confounded data and are thus constructed differently.

**Offline reinforcement learning and pessimism.** Our work is also related to the literature on offline RL and particularly related to the works based on the pessimism principle (Antos et al., 2007; Munos and Szepesvári, 2008; Chen and Jiang, 2019; Buckman et al., 2020; Liu et al., 2020; Min et al., 2021; Jin et al., 2021; Zanette, 2021; Jin et al., 2021; Xie et al., 2021; Uehara and Sun, 2021; Yin and Wang, 2021; Rashidinejad et al., 2021; Zhan et al., 2022; Yin et al., 2022; Yan et al., 2022). Offline RL faces the challenge of the distributional shift between the behavior policy and the family of target policies. Without any coverage assumption on the offline data, the number of data needed to find a near-optimal policy can be exponentially large (Buckman et al., 2020; Zanette, 2021). To circumvent this problem, a few existing works study offline RL under a uniform coverage assumption, which requires the concentrability coefficients between the behavior and target policies are uniformly bounded. See, e.g., Antos et al. (2007); Munos and Szepesvári (2008); Chen and Jiang (2019) and the references therein. Furthermore, a more recent line of work aims to weaken the uniform coverage assumption by adopting the pessimism principle in algorithm design (Liu et al., 2020; Jin et al., 2021; Rashidinejad et al., 2021; Uehara and Sun, 2021; Xie et al., 2021; Yin and Wang, 2021; Zanette et al., 2021; Yin et al., 2022; Yan et al., 2022). In particular, these works proves theoretically that pessimism is effective in tackling the distributional shift of the offline dataset. In particular, by constructing pessimistic value function estimates, these works establish upper bounds on the suboptimality of the proposed methods based on significantly weaker partial coverage assumption. That is, these methods can find a near-optimal policy as long as the dataset covers the optimal policy. The efficacy of pessimism has also been validated empirically in Kumar et al. (2020); Kidambi et al. (2020); Yu et al. (2021); Janner et al. (2021). Compared with these works on pessimism, we focus on the more challenging setting of POMDP with a confounded dataset. To perform pessimism in the face of confounders, we conduct uncertainty quantification for the minimax estimation regarding the confounding bridge functions. Our work complements this line of research by successfully applying pessimism to confounded data.

**OPE via causal inference.** Our work is closely related to the line of research that employing tools from causal inference (Pearl, 2009) for studying OPE with unobserved confounders (Oberst and Sontag, 2019; Kallus and Zhou, 2020; Bennett et al., 2021; Kallus and Zhou, 2021; Mastouri et al., 2021; Shi et al., 2021; Bennett and Kallus, 2021; Shi et al., 2022). Among them, Bennett and Kallus (2021); Shi et al. (2021) are most relevant to our work. In particular, these works also leverage proximal causal inference (Lipsitch et al., 2010; Miao et al., 2018a;b; Cui et al., 2020; Tchetgen et al., 2020; Singh, 2020) to identify the value of the target policy in POMDPs. See Tchetgen et al. (2020) for a detailed survey of proximal causal inference. In comparison, this line of research only focuses on evaluating a single policy, whereas we focus on learning the optimal policy within a class of target policies. As a result, we need to handle a more challenging distributional shift problem between the behavior policy and an entire class of target policies, as opposed to a single target policy in OPE. However, thanks to the pessimism, we establish theory based on a partial coverage assumption that is similar to that in the OPE literature. To achieve such a goal, we conduct uncertainty quantification for the bridge function estimators, which is absent in the the works on OPE. As a result, our analysis is different from that in Bennett and Kallus (2021); Shi et al. (2021).

**Relations between minimax-typed loss and least-square-typed loss (Xie et al., 2021).** During the preparation of this paper, we find that *in the MDP setting*, the least-square-typed loss considered by (Xie et al., 2021) can be reformulated to the minimax-typed loss that we consider in this paper with a different dual function class. To see this, consider the MDP setting with a single transition tuple $(S_h, A_h, S_{h+1})$. The goal is to estimate the Bellman target $(\mathcal{B}V_{h+1})\colon \mathcal{S} \times \mathcal{A} \to \mathbb{R}$, where $\mathcal{B}$ is the Bellman operator and $V_{h+1}\colon \mathcal{S} \to \mathbb{R}$ is a fixed state-value function. For each $(s,a) \in \mathcal{S} \times \mathcal{A}$, $(\mathcal{B}^\pi f_{h+1})(s,a)$ is given by

$$(\mathcal{B}f_{h+1})(s,a) = R_h(s,a) + \int_{\mathcal{S}} P_h(\mathrm{d}s'|s,a)V_{h+1}(s').$$

Here $R_h$ is the reward function and we can assume it is known for now, and $P_h : \mathcal{S} \times \mathcal{A} \mapsto \Delta(\mathcal{S})$ is the unknown transition kernel. We use function class $\mathcal{F}$ to approximate the bellman target. Then based on the offline transition data $\mathbb{D} = \{(s_h^\tau, a_h^\tau, s_{h+1}^\tau)\}_{\tau=1}^N$, the least-square-typed loss function given in Equation (3.1) of (Xie et al., 2021) becomes

$$\widehat{\mathcal{L}}_h^{\mathrm{ls}}(f_h) = \widehat{\mathbb{E}}_{\mathbb{D}}\left[\left(f_h(S_h, A_h) - R_h - V_{h+1}(S_{h+1})\right)^2\right]$$
$$- \min_{f_h' \in \mathcal{F}} \widehat{\mathbb{E}}_{\mathbb{D}}\left[\left(f_h'(S_h, A_h) - R_h - V_{h+1}(S_{h+1})\right)^2\right], \tag{B.1}$$

where $R_h$ is an abbreviation for $R_h(S_h, A_h)$. Using the equality $x^2 - y^2 = (x+y)(x-y)$, we can rewrite the least-square-typed loss (B.1) as

$$\widehat{\mathcal{L}}_h^{\mathrm{ls}}(f_h) = \sup_{f_h' \in \mathcal{F}} \widehat{\mathbb{E}}_{\mathbb{D}}\left[\left((f_h + f_h')(S_h, A_h) - 2R_h - 2V_{h+1}(S_{h+1})\right)\left((f_h - f_h')(S_h, A_h)\right)\right].$$

For derivation, we further rewrite first term as

$$(f_h + f_h')(S_h, A_h) - 2R_h - 2V_{h+1}(S_{h+1})$$
$$= \left(2f_h(S_h, A_h) - 2R_h - 2V_{h+1}(S_{h+1})\right) - \left((f_h - f_h')(S_h, A_h)\right).$$

With this, we can then rewrite the least-square-typed loss (B.1) as

$$\widehat{\mathcal{L}}_h^{\mathrm{ls}}(f_h) = \sup_{f_h' \in \mathcal{F}} \widehat{\mathbb{E}}_{\mathbb{D}}\Big[\left(2f_h(S_h, A_h) - 2R_h - 2V_{h+1}(S_{h+1})\right)\left((f_h - f_h')(S_h, A_h)\right)$$
$$- \left((f_h - f_h')(S_h, A_h)\right)^2\Big].$$

Now by defining a new function class $\mathcal{G}_f$ *depending on f* as $\mathcal{G}_f = \{f - f' : f' \in \mathcal{F}\}$, we arrive that

$$\frac{1}{2}\widehat{\mathcal{L}}_h^{\mathrm{ls}}(f_h) = \sup_{g_h \in \mathcal{G}_{f_h}} \widehat{\mathbb{E}}_{\mathbb{D}}\left[\left(f_h(S_h, A_h) - R_h - V_{h+1}(S_{h+1})\right)g_h(S_h, A_h) - \frac{1}{2}g_h(S_h, A_h)^2\right]. \tag{B.2}$$

This shares the same form as the minimax-typed loss $\sup_{g_h \in \mathbb{G}} \widehat{\Phi}_{\pi,h}^{1/2}(b_h, b_{h+1}; g_h)$ we consider in our work, see (3.10) in the main text. But still there are differences. In (B.2), the dual function $g_h$ lies in a dual function class $\mathcal{G}_{f_h}$ which depends on the primal function $f_h$. While in our minimax-typed loss, the dual function class does not depends on the primal function.

Finally, we need to point out that even the two losses share the same form, the form of the confidence region considered by our work is different from that considered by Xie et al. (2021). To see this, still using the previous notations, the confidence region in Xie et al. (2021) (Equation (3.2)) becomes

$$\mathrm{CR}_h(\xi) = \left\{f_h \in \mathcal{F} : \widehat{\mathcal{L}}_h^{\mathrm{ls}}(f_h) \leq \xi\right\}.$$

Meanwhile, if we reduce our confidence region to the above MDP setting, our confidence region should be in the form of

$$\mathrm{CR}_h(\xi) = \left\{f_h \in \mathcal{F} : \widehat{\mathcal{L}}_h^{\mathrm{mm}}(f_h) - \min_{f_h \in \mathcal{F}} \widehat{\mathcal{L}}_h^{\mathrm{mm}}(f_h) \leq \xi\right\},$$

where $\mathcal{L}_h^{\mathrm{mm}}(f_h)$ denotes the minimax-typed-loss. Our algorithm and theoretical analysis are based on the second form of confidence region, which is key to the derivation of fast statistical rates for elements in the confidence region based on minimax estimation.

### B.2 DISCUSSION ABOUT THE PARTIAL COVERAGE

**More about the partial coverage (Assumption 4.1).** Our work assumes the partial coverage of $\mathbb{D}$ according to Assumption 4.1, where we implicitly requires that $\mathcal{P}_h^\pi \left(S_h, \Gamma_{h-1}\right) / \mathcal{P}_h^b \left(S_h, \Gamma_{h-1}\right) < +\infty$ for all $\pi \in \Pi(\mathcal{H})$ (we call it the finite-ratio condition from here). We note that this finite-ratio condition can NOT be regarded as the full coverage assumption. Instead, this is a regularity condition that arises from causal inference.

First of all, the finite-ratio condition is different from the full coverage assumption in standard MDPs. The Full coverage assumption in standard MDPs usually takes the form that

$$\max_{\pi \in \Pi} \frac{\mathcal{P}_h^\pi(s, a)}{\mathcal{P}_h^b(s, a)} < C,$$

for some fixed $C > 0$. This condition means the density ratio of the marginal distributions of $(s, a)$ between any target policy $\pi$ and the behavior policy $\pi^b$ is uniformly bounded by a constant. This condition (or some similar form) is a common and widely accepted form of full coverage in the MDP literature, e.g. (Chen and Jiang, 2019; Xie and Jiang, 2020). Note that this constant $C$ is a uniform upper bound over the candidate policy class. Very importantly, this constant $u'$ appears in the final error bound. The partial coverage assmuption in MDP, on the other hand, is commonly formulated as

$$\frac{\mathcal{P}_h^{\pi^\star}(s, a)}{\mathcal{P}_h^b(s, a)} < C,$$

This condition means the density ratio of the marginal distributions of $(s, a)$ between only the optimal policy $\pi^\star$ and the behavior policy $\pi^b$, is bounded by a constant. The form of this assumption is very close to Assumption 4.1 (Partial coverage) in our paper. In other words, our Assumption 4.1 is a version of the partial coverage assumption that is tailored to the POMDP case. Notably, this constant $C$ in the partial coverage assumption also appears in the final error bound.

As a sharp comparison to both the full coverage and partial coverage assumptions, the finite-ratio condition that the quantity $\mathcal{P}_h^\pi \left(S_h, \Gamma_{h-1}\right) / \mathcal{P}_h^b \left(S_h, \Gamma_{h-1}\right) < +\infty$ for all $\pi \in \Pi(\mathcal{H})$ does not result in any constant factor that appears in the final error bound. In the case of infinite policy class $\Pi(\mathcal{H})$, we can allow the ratio to be arbitrarily large and that won't hurt our final error bound. Therefore, this is not a coverage assumption. Our finite-ratio condition is a regularity condition that arises from causal inference. This condition is needed to deal with the extra challenge of the confounding issue in our POMP setting. In related works studying OPE under confounded POMDP (Shi et al., 2021), this finite-ratio condition is also needed. Overall, our paper is indeed under partial coverage and the finite ration condition is not a kind of coverage assumption.

### B.3 POTENTIAL APPLICATION: REAL-WORLD EXAMPLE OF PROXIMAL CAUSAL INFERENCE IN RL.

Let us consider the real-world example of applying the POMDP model to sepsis treatment studied by Tsoukalas et al. (2015). In such an example, the state, action, observation, and reward of the POMDP are given by the following:

- State variable $S_h$ refers to the clinical state of the patient, e.g., sepsis, SIRS, Bacteremia, etc.

- Observable variable $O_h$ refers to all the information one can read from a medical device, such as the heart rate, the respiratory rate, blood pressure, blood test result of infection, etc.

- Action $A_h$ refers to certain treatment given to the patient. For example, each antibiotic combination can be considered as an action. As mentioned in Tsoukalas et al. (2015), a total of 48 antibiotics have been included in the patient's remedy.

- Reward/cost values need to be provided empirically by physicians, based on the severity of each state. In the example of Tsoukalas et al. (2015), the states and their corresponding rewards/costs include: Healthy (100,000), No SIRS (50,000), Probable Sepsis (PS, 5000), SIRS (-50), Bacteremia (-10,000), etc.

- Finally, a history trajectory is the record of antibiotic treatment received by the patient. The behavior policy is some treatment plans that have been applied to some patients to generate the dataset.

When using reactive policies (Example 2.1), the negative control action variable ($Z_h$) is just the observation variable $O_{h-1}$ which reflects the patient's clinical state at the last treatment time step, and the negative control outcome variable ($W_h$) is just the observation variable $O_h$ at the current time step. Furthermore, when the observation $O$ contains enough information to reflect the underlying state $S$, which basically implies a certain full rank assumption, we can then use Example C.1 to guarantee the existence of the bridge functions (See Appendix C).

## C PROXIMAL CAUSAL INFERENCE

In this Section, we complement the discussion of proximal causal inference in Section 3.1.

### C.1 ILLUSTRATION OF EXAMPLES

In this subsection, we give detailed discussions for the three examples of history-dependent policies mentioned in Section 2.3. In particular, we give causal graphs of the POMDP when adopting these policies. Also, we explain the choice of negative control variables for these policies in Section 3.1.

#### C.1.1 REACTIVE POLICY (EXAMPLE 2.1 REVISITED)

When the target policy is a reactive policy, it only depends on the current observation $O_h$. That is, $\mathcal{H}_{h-1} = \{\varnothing\}$ and $\Gamma_{h-1} = \varnothing$ for each $h \in [H]$. The causal graph for such a target policy is shown in Figure 2. In this case, we choose the negative control action as $Z_h = O_{h-1}$ (node in **green**) and the negative control outcome as $W_h = O_h$ (node in **yellow**). By this choice, we can check the independence condition in Assumption 3.1 via Figure 2, i.e., under $\mathcal{P}^b$,

$$O_{h-1} \perp O_h, R_h, O_{h+1} \mid S_h, A_h \quad O_h \perp A_h, S_{h-1} \mid S_h.$$

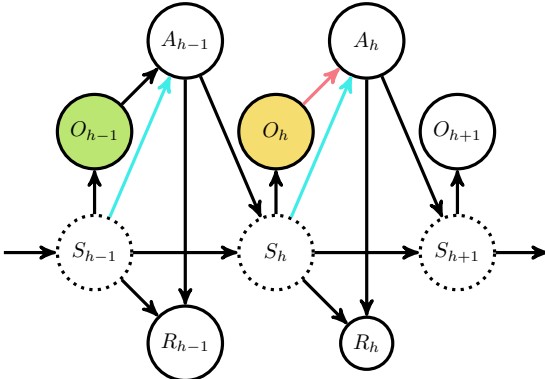

Figure 2: Causal graph for reactive policy. The dotted nodes indicate that the variables are not stored in the offline dataset. Solid arrows indicate the dependency among the variables. Specifically, The **red** arrows depict the dependence of the target policy on the observable variables. The **blue** arrows depict the dependence of the behavior policy on the latent state. The negative control action and outcome variables at the $h$-th step are filled in **green** and **yellow**, respectively.

#### C.1.2 FINITE-HISTORY POLICY (EXAMPLE 2.2 REVISITED)

When the target policy a is finite-length history policy, it depends on the current observation and history of length at most $k$. That is, $\mathcal{H}_{h-1} = (\mathcal{O} \times \mathcal{A})^{\otimes \min\{k, h-1\}}$ for some $k \in \mathbb{N}$, $\Gamma_{h-1} = ((O_l, A_l), \cdots, (O_{h-1}, A_{h-1}))$ where the index $l = \max\{1, h-k\}$. The causal graph for such a target policy is shown in Figure 3. In this case, we choose the negative control action as $Z_h = O_{l-1}$ (node in **green**) and the negative control outcome as $W_h = O_h$ (node in **yellow**). By this choice, we can check the independence condition in Assumption 3.1 via Figure 3, i.e., under $\mathcal{P}^b$,

$$O_{l-1} \perp O_h, R_h, O_{h+1} \mid S_h, A_h, O_{h-1}, A_{h-1}, \cdots, O_l, A_l,$$
$$O_h \perp A_h, S_{h-1}, O_{h-1}, A_{h-1}, \cdots, O_l, A_l \mid S_h.$$

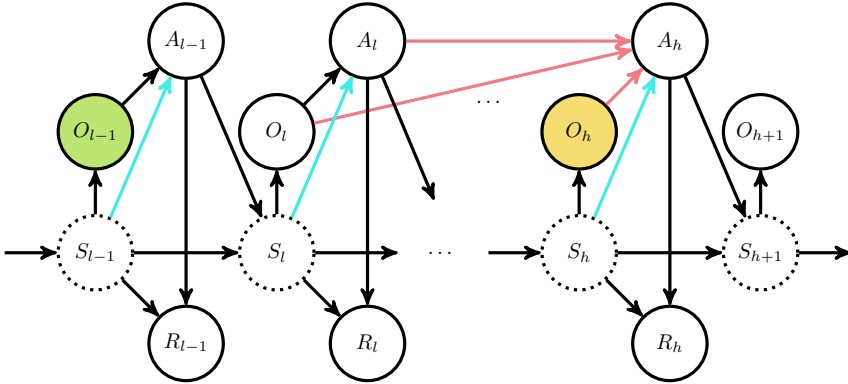

Figure 3: Causal graph for finite-length history policy. Index $l = \max\{1, h - k\}$. The dotted nodes indicate that the variables are not stored in the offline dataset. Solid arrows indicate the dependency among the variables. Specifically, The **red** arrows depict the dependence of the target policy on the observable variables. The **blue** arrows depict the dependence of the behavior policy on the latent state. The negative control action and outcome variables at step $h$ are filled in **green** and **yellow**.

### C.1.3 FULL-HISTORY POLICY (EXAMPLE 2.3 REVISITED)

When the target policy is a full-history policy, it depends on the current observation and the full history. That is, $\mathcal{H}_{h-1} = (\mathcal{O} \times \mathcal{A})^{\otimes(h-1)}$ and $\Gamma_{h-1} = ((O_1, A_1), \cdots, (O_{h-1}, A_{h-1}))$. The causal graph for such a target policy is shown in Figure 4. In this case, we choose the negative control action as $Z_h = O_0$ (node in **green**) and the negative control outcome as $W_h = O_h$ (node in **yellow**). By this choice, we can check the independence condition in Assumption 3.1 via Figure 4, i.e., under $\mathcal{P}^b$,

$$O_0 \perp O_h, R_h, O_{h+1} \mid S_h, A_h, O_{h-1}, A_{h-1}, \cdots, O_1, A_1,$$
$$O_h \perp A_h, S_{h-1}, O_{h-1}, A_{h-1}, \cdots, O_1, A_1 \mid S_h.$$

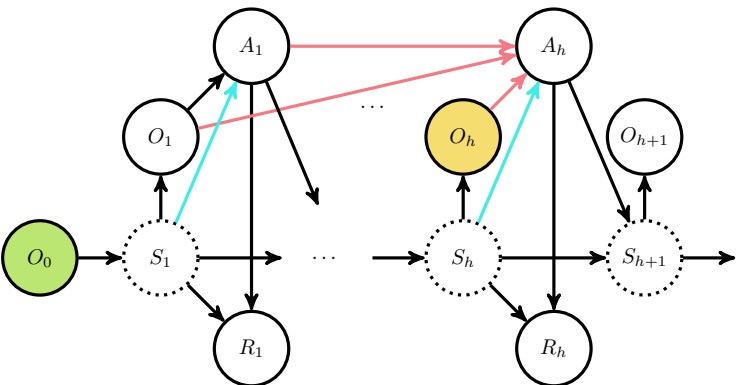

Figure 4: Causal graph for full-length history policy. The dotted nodes indicate that the variables are not stored in the offline dataset. Solid arrows indicate the dependency among the variables. Specifically, The **red** arrows depict the dependence of the target policy on the observable variables. The **blue** arrows depict the dependence of the behavior policy on the latent state. The negative control action and outcome variables at step $h$ are filled in **green** and **yellow**, respectively.

### C.2 EXAMINATION OF ASSUMPTION 3.2

In this subsection, we give concrete examples when the Assumption 3.2 holds, i.e., the confounding bridge functions exist.

**Example C.1** (Example 2.1 revisited). *For the tabular setting (i.e., $\mathcal{S}$, $\mathcal{A}$, and $\mathcal{O}$ are finite spaces) and reactive policies (i.e., $\pi_h : \mathcal{O} \mapsto \Delta(\mathcal{A})$), the sufficient condition under which Assumption 3.2 holds is that*

$$\mathbf{rank}(\mathcal{P}_h^b(O_h|S_h)) = |\mathcal{S}|, \quad \mathbf{rank}(\mathcal{P}_h^b(O_{h-1}|S_h)) = |\mathcal{S}|, \tag{C.1}$$

*where $\mathcal{P}_h(O_h|S_h)$ denote an $|\mathcal{S}| \times |\mathcal{O}|$ matrix whose $(s,o)$-th element is $\mathcal{P}_h^b(O_h = o|S_h = s)$, and $\mathcal{P}_h^b(O_{h-1}|S_h)$ is defined similarly.*

*Proof of Example C.1.* Recall that for reactive policies, the history information $\Gamma_{h-1} = \varnothing$. We first show that under condition (C.1), there exist functions $\{b_h^\pi\}_{h=1}^H$ and $\{q_h^\pi\}_{h=1}^H$ which solve the following equations

$$\mathbb{E}_{\pi^b}\left[b_h^\pi(A_h, O_h)|A_h, S_h\right]$$
$$= \mathbb{E}_{\pi^b}\left[R_h\pi_h(A_h|O_h) + \gamma\sum_{a'} b_{h+1}^\pi(a', O_{h+1})\pi_h(A_h|O_h)\Big|A_h, S_h\right], \tag{C.2}$$

$$\mathbb{E}_{\pi^b}\left[q_h^\pi(A_h, O_{h-1})|A_h, S_h\right] = \frac{\mu_h(S_h)}{\pi_h^b(A_h|S_h)}, \tag{C.3}$$

Then we show that the solutions to (C.2) and (C.3) also solve (3.2) and (3.3). The difference between (C.2) and (3.2) is that in (C.2) we condition on the latent state $S_h$ rather than the observable negative control variable $Z_h$. In related literature (Bennett and Kallus, 2021; Shi et al., 2021), the solutions to (C.2) and (C.3) are referred to as *unlearnable bridge functions*.

We first show the existence of $\{b_h^\pi\}_{h=1}^H$ in a backward manner. Denote by $b_{h+1}^\pi$ a zero function. Suppose that $b_{h+1}^\pi$ exists, we show that $b_h^\pi$ also exists. Since now spaces $\mathcal{S}$, $\mathcal{A}$, and $\mathcal{O}$ are discrete, we adopt the notation of matrix. In particular, we denote by

$$\mathbf{B} \in \mathbb{R}^{|\mathcal{A}| \times |\mathcal{O}|}, \qquad \mathbf{B}(a,o) = b_h(a,o),$$
$$\mathbf{O} \in \mathbb{R}^{|\mathcal{S}| \times |\mathcal{O}|}, \qquad \mathbf{O}(s,o) = \mathcal{P}_h^b(O_h = o|S_h = s),$$

$$\mathbf{R} \in \mathbb{R}^{|\mathcal{A}| \times |\mathcal{S}|}, \ \mathbf{R}(s,a) = \mathbb{E}_{\pi^b}\left[R_h\pi_h(A_h|O_h) + \gamma\sum_{a'} b_{h+1}^\pi(a', O_{h+1})\pi_h(A_h|O_h)\Big|A_h = a, S_h = s\right].$$

The existence of $b_h^\pi$ satisfying (C.2) is equivalent to the existence of $\mathbf{B}$ solving the matrix equation

$$\mathbf{B}\,\mathbf{O}^\top = \mathbf{R}. \tag{C.4}$$

By condition (C.1), we known that the matrix $\mathbf{O}^\top$ is of full column rank, which indicates that (C.4) admits a solution $\mathbf{B}$. This proves the existence of $b_h^\pi$. For $\{q_h^\pi\}_{h=1}^H$, we use a similar method by considering

$$\mathbf{Q} \in \mathbb{R}^{|\mathcal{A}| \times |\mathcal{O}|}, \qquad \mathbf{Q}(a,o) = q_h(a,o),$$
$$\mathbf{O}_- \in \mathbb{R}^{|\mathcal{S}| \times |\mathcal{O}|}, \qquad \mathbf{O}_-(s,o) = \mathcal{P}_h^b(O_{h-1} = o|S_h = s),$$
$$\mathbf{C} \in \mathbb{R}^{|\mathcal{A}| \times |\mathcal{S}|}, \qquad \mathbf{C}(s,a) = \frac{\mu_h(S_h = s)}{\pi_h^b(A_h = a|S_h = s)}.$$

The existence of $q_h^\pi$ satisfying (C.3) is equivalent to the existence of $\mathbf{Q}$ solving the matrix equation

$$\mathbf{Q}\,\mathbf{O}_-^\top = \mathbf{C} \tag{C.5}$$

By condition (C.1), we known that the matrix $\mathbf{O}_-^\top$ is of full column rank, which indicates that (C.5) admits a solution $\mathbf{Q}$. This proves the existence of $q_h^\pi$. Thus we have shown that there exists $\{b_h^\pi\}_{h=1}^H$ and $\{q_h^\pi\}_{h=1}^H$ which solve equation (C.2) and (C.3). Finally, it holds that any solution to (C.2) and (C.3) also forms a solution to (3.2) and (3.3), which has been shown in Theorem 11 in Shi et al. (2021). This finishes the proof of Example C.1. $\qquad\square$

**Example C.2** (Example 2.2 revisited). *For the tabular setting and finite length policies (i.e., $\pi_h : \mathcal{O} \times (\mathcal{O} \times \mathcal{A})^{\min\{k, h-1\}} \mapsto \Delta(\mathcal{A})$), the sufficient condition under which Assumption 3.2 holds is that, for any action $a \in \mathcal{A}$,*

$$\mathbf{rank}(\mathcal{P}_h^b(O_h|A_h = a, O_{h-k-1})) = |\mathcal{O}|, \quad \mathbf{rank}(\mathcal{P}_h^b(O_{h-k-1}|A_h = a, S_h, \Gamma_{h-1})) = |\mathcal{O}|, \tag{C.6}$$

*where $\mathcal{P}_h^b(O_h|A_h = a, O_{h-k-1})$ is a $|\mathcal{O}| \times |\mathcal{O}|$ matrix with $(o,o')$-th element is $\mathcal{P}_h^b(O_h = o|A_h = a, O_{h-k-1} = o')$ and $\mathcal{P}_h^b(O_{h-k-1}|A_h = a, S_h, \Gamma_{h-1})$ is a $|\mathcal{S}||\mathcal{H}_{h-1}| \times |\mathcal{O}|$ matrix defined similarly.*

*Proof of Example C.2.* To see this, we first prove the existence of $\{b_n^\pi\}$. For simplicity, we denote by

$$\mathbf{P}_a = \left(\mathcal{P}_h^b\left(O_h \mid A_h = a, O_{h-k-1}\right)\right) \in \mathbb{R}^{|\mathcal{O}| \times |\mathcal{O}|}$$

for each $a \in \mathcal{A}$. Also, we denote that

$$\mathbf{B}_a = \left(b_h\left(a, O_h\right)\right) \in \mathbb{R}^{|\mathcal{O}| \times 1},$$

$$\mathbf{R}_a = \left(\mathbb{E}_{\pi^b}\left[R_h \pi_h\left(A_h \mid O_h\right) + \gamma \sum_{a'} b_{h+1}^\pi\left(a', O_{h+1}\right) \pi_h\left(A_h \mid O_h\right) \mid A_h = a, O_{h-k-1}\right]\right) \in \mathbb{R}^{|\mathcal{O}| \times 1}.$$

Then for each $a \in \mathcal{A}$, the existence of $b_n^\pi(a, \cdot)$ is equivalent to the existence of the solution to

$$\mathbf{P}^a \mathbf{B}^a = \mathbf{R}^a.$$

Such a linear equation admits a solution due to our assumption on the matrix $\mathbf{P}_a$. This shows the existence of $\{b_h^\pi\}$. For $\{q_h^\pi\}$, the deduction is similar by considering for each $a \in \mathcal{A}$,

$$\mathbf{T}_a = \left(\mathcal{P}_h^b\left(O_{h-k-1} \mid A_h = a, S_h, \Gamma_{h-1}\right)\right) \in \mathbb{R}^{|\mathcal{S}||\mathcal{H}_{h-1}| \times |\mathcal{O}|},$$

$$\mathbf{Q}_a = \left(q_h\left(a, O_{h-k-1}\right)\right) \in \mathbb{R}^{|\mathcal{O}| \times 1},$$

$$\mathbf{C}_a = \left(\frac{\mu_h\left(S_h, \Gamma_{h-1}\right)}{\pi^b\left(a \mid S_h\right)}\right) \in \mathbb{R}^{|\mathcal{S}||\mathcal{H}_{h-1}| \times 1}.$$

By considering the equation that

$$\mathbf{T}_a \mathbf{Q}_a = \mathbf{C}_a$$

and using the full rank assumption on matrix $\mathbf{T}_a$, we can obtain the existence of $\{q_h^\pi\}$. This finishes the proof of Example C.2. $\qquad\square$

# D    PROOF SKETCHES OF MAIN THEORETICAL RESULT

In this section, we sketch the proof of the main theoretical result Theorem 4.4, and we refer to Appendix G for a detailed proof. For simplicity, we denote that for any $\pi \in \Pi(\mathcal{H})$ and $\mathbf{b} \in \mathbb{B}^{\otimes H}$,

$$F(\mathbf{b}) := \mathbb{E}_{\pi^b}\left[\sum_{a \in \mathcal{A}} b_1(a, W_1)\right], \quad \widehat{F}(\mathbf{b}) := \widehat{\mathbb{E}}_{\pi^b}\left[\sum_{a \in \mathcal{A}} b_1(a, W_1)\right]. \tag{D.1}$$

By the definition (D.1) and Theorem 3.3, for any policy $\pi \in \Pi(\mathcal{H})$, it holds that $J(\pi) = F(\mathbf{b}^\pi)$, where we have denoted by $\mathbf{b}^\pi = (b_1^\pi, \cdots, b_H^\pi)$ the vector of true value bridge functions of $\pi$ which are given in (3.2).

Our proof to Theorem 4.4 relies on the following three key lemmas. The first lemma relates the different values of mapping $F(\cdot)$ induced by a true value bridge function $\mathbf{b}^\pi$ and any other functions $\mathbf{b} \in \mathbb{B}^{\otimes H}$ to the RMSE loss which we aim to minimize by algorithm design. This indeed decomposes the suboptimality (2.2).

**Lemma D.1** (Suboptimality decomposition). *Under Assumption 3.1, 3.2, for any policy $\pi \in \Pi(\mathcal{H})$ and $\mathbf{b} \in \mathbb{B}^{\otimes H}$, it holds that*

$$F(\mathbf{b}^\pi) - F(\mathbf{b}) \le \sum_{h=1}^H \gamma^{h-1} \sqrt{C^\pi} \cdot \sqrt{\mathcal{L}_h^\pi(b_h, b_{h+1})},$$

*where the concentrability coefficient $C^\pi$ is defined as $C^\pi := \sup_{h \in [H]} \mathbb{E}_{\pi^b}\left[(q_h^\pi(A_h, Z_h))^2\right].$*

*Proof of Lemma D.1.* See Appendix F.1 for a detailed proof. $\qquad\square$

The following two lemmas characterize the theoretical properties of the confidence region $\mathrm{CR}^\pi(\xi)$. Specifically, Lemma D.2 shows that with high probability the confidence region of $\pi$ contains the true value bridge function $\mathbf{b}^\pi$. Besides, Lemma D.3 shows that each bridge function vector $\mathbf{b} \in \mathrm{CR}^\pi(\xi)$ enjoys a fast statistical rate (Uehara et al., 2021) for its RMSE loss $\mathcal{L}_h^\pi$ defined in (3.6). To obtain such a fast rate, we develop novel proof techniques in Appendix F.3.

**Lemma D.2** (Validity of confidence regions). *Under Assumption 3.2 and 4.2, for any $0 < \delta < 1$, by setting*

$$\xi = C_1(\lambda + 1/\lambda) \cdot M_{\mathbb{B}}^2 \cdot M_{\mathbb{G}}^2 \cdot \log(|\mathbb{B}||\Pi(\mathcal{H})|H/\zeta)/n,$$

*for some problem-independent universal constant $C_1 > 0$ and $\zeta = \min\{\delta, 4c_1 \exp(-c_2 n \alpha_{\mathbb{G},n}^2)\}$, it holds with probability at least $1 - \delta$ that $\mathbf{b}^\pi \in \mathrm{CR}^\pi(\xi)$ for any policy $\pi \in \Pi(\mathcal{H})$.*

*Proof of Lemma D.2.* See Appendix F.2 for a detailed proof. $\square$

**Lemma D.3** (Accuracy of confidence regions). *Under Assumption 3.2, 4.2, and 4.3, by setting the same $\xi$ as in Lemma D.2, with probability at least $1 - \delta/2$, for any policy $\pi \in \Pi(\mathcal{H})$, $\mathbf{b} \in \mathrm{CR}^\pi(\xi)$, and step $h$,*

$$\sqrt{\mathcal{L}_h^\pi(b_h, b_{h+1})} \le \widetilde{C}_1 M_{\mathbb{B}} M_{\mathbb{G}} \sqrt{(\lambda + 1/\lambda) \cdot \log(|\mathbb{B}||\Pi(\mathcal{H})|H/\zeta)/n} + \widetilde{C}_1 \epsilon_{\mathbb{B}}^{1/4} M_{\mathbb{G}}^{1/2},$$

*for some problem-independent universal constant $\widetilde{C}_1 > 0$, and $\zeta = \min\{\delta, 4c_1 \exp(-c_2 n \alpha_{\mathbb{G},n}^2)\}$.*

*Proof of Lemma D.3.* See Appendix F.3 for a detailed proof. $\square$

When $\alpha_{\mathbb{G},n} \in \mathcal{O}(n^{-1/2})$ and $\epsilon_{\mathbb{B}} = 0$, Lemma D.3 implies that $\mathcal{L}_h^\pi(b_h, b_{h+1}) \le \widetilde{\mathcal{O}}(n^{-1})$. Now with Lemma D.1, Lemma D.2, and Lemma D.3, by the choice of $\widehat{\pi}$ in P3O, we can show that

$$
\begin{aligned}
J(\pi^\star) - J(\widehat{\pi}) &\le \widetilde{\mathcal{O}}(n^{-1/2}) + \max_{\mathbf{b} \in \mathrm{CR}^{\pi^\star}(\xi)} F(\mathbf{b}) - \min_{\mathbf{b} \in \mathrm{CR}^{\widehat{\pi}}(\xi)} F(\mathbf{b}) \\
&\le \widetilde{\mathcal{O}}(n^{-1/2}) + \max_{\mathbf{b} \in \mathrm{CR}^{\pi^\star}(\xi)} F(\mathbf{b}) - \min_{\mathbf{b} \in \mathrm{CR}^{\pi^\star}(\xi)} F(\mathbf{b}) \\
&\le \widetilde{\mathcal{O}}(n^{-1/2}) + 2 \max_{\mathbf{b} \in \mathrm{CR}^{\pi^\star}(\xi)} \left| F(\mathbf{b}) - F(\mathbf{b}^{\pi^\star}) \right| \\
&\le \widetilde{\mathcal{O}}(n^{-1/2}) + 2 \max_{\mathbf{b} \in \mathrm{CR}^{\pi^\star}(\xi)} \sum_{h=1}^H \gamma^{h-1} \sqrt{C^{\pi^\star}} \cdot \sqrt{\mathcal{L}_h^{\pi^\star}(b_h, b_{h+1})}, \quad \text{(D.2)}
\end{aligned}
$$

where the first inequality holds by Lemma D.2, the second inequality holds from the optimality of $\widehat{\pi}$ in Algorithm 1, the third inequality holds directly, and the last inequality holds by Lemma D.1. Finally, by applying Lemma D.3 to the right hand side of (D.2), we conclude the proof of Theorem 4.4.

## E    PROOF OF THEOREM 3.3

*Proof of Theorem 3.3.* For any step $h$, we denote $J_h(\pi) := \mathbb{E}_\pi[R_h(S_h, A_h)]$. We have that

$$
\begin{aligned}
J_h(\pi) &= \mathbb{E}_\pi[R_h(S_h, A_h)] \\
&= \mathbb{E}_\pi\big[\mathbb{E}_\pi[R_h(S_h, A_h)|O_h, S_h, \Gamma_{h-1}]\big] \\
&= \mathbb{E}_\pi\left[\sum_{a \in \mathcal{A}} R_h(S_h, a)\pi_h(a|O_h, \Gamma_{h-1})\right] \\
&= \mathbb{E}_\pi\left[\mathbb{E}_\pi\left[\sum_{a \in \mathcal{A}} R_h(S_h, a)\pi_h(a|O_h, \Gamma_{h-1}) \bigg| S_h, \Gamma_{h-1}\right]\right],
\end{aligned}
$$

where the second and the last equality follows from the tower property of conditional expectation. Using the definition of density ratio $\mu_h(S_h, \Gamma_{h-1})$ in Assumption 3.2, we can change the outer

expectation to $\mathbb{E}_{\pi^b}$ by

$$J_h(\pi) = \mathbb{E}_{\pi^b}\left[\mu_h(S_h, \Gamma_{h-1}) \cdot \mathbb{E}_\pi\left[\sum_{a \in \mathcal{A}} R_h(S_h, a)\pi_h(a|O_h, \Gamma_{h-1})\Big|S_h, \Gamma_{h-1}\right]\right],$$

$$= \mathbb{E}_{\pi^b}\left[\sum_{a \in \mathcal{A}} R_h(S_h, a) \cdot \pi_h(a|O_h, \Gamma_{h-1}) \cdot \mu_h(S_h, \Gamma_{h-1})\right]$$

$$= \mathbb{E}_{\pi^b}\left[\sum_{a \in \mathcal{A}} \pi_h^b(a|S_h) \cdot R_h(S_h, a) \cdot \frac{\pi_h(a|O_h, \Gamma_{h-1})}{\pi_h^b(a|S_h)} \cdot \mu_h(S_h, \Gamma_{h-1})\right]$$

$$\overset{(a)}{=} \mathbb{E}_{\pi^b}\left[\mathbb{E}_{\pi^b}\left[R_h(S_h, A_h) \cdot \frac{\pi_h(A_h|O_h, \Gamma_{h-1})}{\pi_h^b(A_h|S_h)} \cdot \mu_h(S_h, \Gamma_{h-1})\Big|S_h, O_h, \Gamma_{h-1}\right]\right]$$

$$= \mathbb{E}_{\pi^b}\left[R_h(S_h, A_h) \cdot \pi_h(A_h|O_h, \Gamma_{h-1}) \cdot \frac{\mu_h(S_h, \Gamma_{h-1})}{\pi_h^b(A_h|S_h)}\right],$$

where step (a) follows from the fact that $A_h \sim \pi_h^b(\cdot|S_h)$ and satisfies $A_h \perp O_h, \Gamma_{h-1}|S_h$ under $\pi^b$. Now using the definition (3.3) of weight bridge function $q_h^\pi$ in Assumption 3.2, we have that

$$J_h(\pi) = \mathbb{E}_{\pi^b}\left[R_h(S_h, A_h) \cdot \pi_h(A_h|O_h, \Gamma_{h-1}) \cdot \mathbb{E}_{\pi^b}[q_h^\pi(A_h, Z_h)|S_h, A_h, \Gamma_{h-1}]\right]$$

$$\overset{(a)}{=} \mathbb{E}_{\pi^b}[R_h(S_h, A_h) \cdot \pi_h(A_h|O_h, \Gamma_{h-1}) \cdot q_h^\pi(A_h, Z_h)]$$

$$= \mathbb{E}_{\pi^b}[\mathbb{E}_{\pi^b}[R_h(S_h, A_h) \cdot \pi_h(A_h|O_h, \Gamma_{h-1}) \cdot q_h^\pi(A_h, Z_h)|A_h, Z_h]]$$

$$= \mathbb{E}_{\pi^b}[\mathbb{E}_{\pi^b}[R_h(S_h, A_h) \cdot \pi_h(A_h|O_h, \Gamma_{h-1})\cdot|A_h, Z_h]\, q_h^\pi(A_h, Z_h)],$$

where step (a) follows from the assumption that $Z_h \perp O_h, R_h|S_h, A_h, \Gamma_{h-1}$ by Assumption 3.1. Now using the definition (3.2) of value bridge function $b_h^\pi$ in Assumption 3.2, we have that

$$J_h(\pi) = \mathbb{E}_{\pi^b}\left[\mathbb{E}_{\pi^b}\left[b_h^\pi(A_h, W_h) - \gamma \sum_{a' \in \mathcal{A}} b_{h+1}^\pi(a', W_{h+1})\pi_h(A_h|O_h, \Gamma_{h-1})\Big|A_h, Z_h\right] q_h^\pi(A_h, Z_h)\right]$$

$$= \mathbb{E}_{\pi^b}[f(S_h, A_h, O_h, W_h, W_{h+1}, \Gamma_{h-1}) \cdot q_h^\pi(A_h, Z_h)]$$

$$= \mathbb{E}_{\pi^b}[\mathbb{E}_{\pi^b}[f(S_h, A_h, O_h, W_h, W_{h+1}, \Gamma_{h-1}) \cdot q_h^\pi(A_h, Z_h)|S_h, A_h, O_h, W_h, W_{h+1}, \Gamma_{h-1}]],$$

$$= \mathbb{E}_{\pi^b}[f(S_h, A_h, O_h, W_h, W_{h+1}, \Gamma_{h-1}) \cdot \mathbb{E}_{\pi^b}[q_h^\pi(A_h, Z_h)|S_h, A_h, O_h, W_h, W_{h+1}, \Gamma_{h-1}]],$$

$$\overset{(a)}{=} \mathbb{E}_{\pi^b}[f(S_h, A_h, O_h, W_h, W_{h+1}, \Gamma_{h-1}) \cdot \mathbb{E}_{\pi^b}[q_h^\pi(A_h, Z_h)|S_h, A_h, \Gamma_{h-1}]],$$

where for simplicity we have denoted that

$$f(S_h, A_h, O_h, W_h, W_{h+1}, \Gamma_{h-1}) = b_h^\pi(A_h, W_h) - \gamma \sum_{a' \in \mathcal{A}} b_{h+1}^\pi(a', W_{h+1})\pi_h(A_h|O_h, \Gamma_{h-1}),$$

and step (a) follows from the assumption that $Z_h \perp O_h, W_h, W_{h+1}|S_h, A_h, \Gamma_{h-1}$ by Assumption 3.1. By the definition (3.3) of weight bridge function $q_h^\pi$ in Assumption 3.2 again, we have that

$$J_h(\pi) = \mathbb{E}_{\pi^b}\left[f(S_h, A_h, O_h, W_h, W_{h+1}, \Gamma_{h-1}) \cdot \frac{\mu_h(S_h, \Gamma_{h-1})}{\pi_h^b(A_h|S_h)}\right]$$

$$\overset{(a)}{=} \mathbb{E}_{\pi^b}\left[\left(b_h^\pi(A_h, W_h) - \gamma \sum_{a' \in \mathcal{A}} b_{h+1}^\pi(a', W_{h+1})\pi_h(A_h|O_h, \Gamma_{h-1})\right) \cdot \frac{\mu_h(S_h, \Gamma_{h-1})}{\pi_h^b(A_h|S_h)}\right],$$

where step (a) just applies the definition of $f$. Now sum $J_h(\pi)$ over $h \in [H]$, we have that

$$J(\pi) = \sum_{h=1}^H \gamma^{h-1}J_h(\pi) = \underbrace{\mathbb{E}_{\pi^b}\left[\frac{\mu_1(S_1, \Gamma_0)}{\pi_1^b(A_1|S_1)}b_1^\pi(A_1, W_1)\right]}_{(A)} + \underbrace{\sum_{h=2}^H \gamma^{h-1}\Delta_h}_{(B)}, \tag{E.1}$$

where for simplicity we define $\Delta_h$ for $h = 2, \cdots, H$ as

$$\Delta_h = \mathbb{E}_{\pi^b}\left[\frac{\mu_h(S_h, \Gamma_{h-1})}{\pi_h^b(A_h|S_h)}b_h^\pi(A_h, W_h) - \frac{\mu_{h-1}(S_{h-1}, \Gamma_{h-2})}{\pi_{h-1}^b(A_{h-1}|S_{h-1})} \cdot \sum_{a' \in \mathcal{A}} b_h^\pi(a', W_h)\pi_{h-1}(A_{h-1}|O_{h-1}, \Gamma_{h-1})\right].$$

In the sequel, we deal with term (A) and (B) respectively. On the one hand, we have that

$$
\begin{aligned}
(A) &\overset{(a)}{=} \mathbb{E}_{\pi^b}\left[\frac{\mathcal{P}_1^\pi(S_1, \Gamma_0)}{\mathcal{P}_1^b(S_1, \Gamma_0)\pi_1^b(A_1|S_1)}b_1^\pi(A_1, W_1)\right]\\
&\overset{(b)}{=} \mathbb{E}_{\pi^b}\left[\frac{1}{\pi_1^b(A_1|S_1)}b_1^\pi(A_1, W_1)\right]\\
&= \mathbb{E}_{\pi^b}\left[\mathbb{E}_{\pi^b}\left[\frac{1}{\pi_1^b(A_1|S_1)}b_1^\pi(A_1, W_1)\Big|S_1, W_1\right]\right]\\
&\overset{(c)}{=} \mathbb{E}_{\pi^b}\left[\sum_{a\in\mathcal{A}}\frac{\pi_1^b(a|S_1)}{\pi_1^b(a|S_1)}b_1^\pi(a, W_1)\right]\\
&= \mathbb{E}_{\pi^b}\left[\sum_{a\in\mathcal{A}}b_1^\pi(a, W_1)\right],
\end{aligned}
$$

where step (a) follows from the definition of $\mu_1(S_1, \Gamma_0)$ in Assumption 3.2, step (b) follows from the fact that at $h = 1$, $\mathcal{P}_1^b(S_1, \Gamma_0) = \mathcal{P}_1^\pi(S_1, \Gamma_0)$, and step (c) follows from the assumption that $A_1 \perp W_1|S_1$ by Assumption 3.1. On the other hand, term (b) in (E.1) is actually 0, which we show by proving that $\Delta_h = 0$ for any $h \geq 2$. We denote by $\Delta_h = \Delta_h^1 - \Delta_h^2$ and we consider $\Delta_h^1$ and $\Delta_h^2$ respectively, where

$$
\Delta_h^1 = \mathbb{E}_{\pi^b}\left[\frac{\mu_h(S_h, \Gamma_{h-1})}{\pi_h^b(A_h|S_h)}\cdot b_h^\pi(A_h, W_h)\right],
$$

$$
\Delta_h^2 = \mathbb{E}_{\pi^b}\left[\frac{\mu_{h-1}(S_{h-1}, \Gamma_{h-2})}{\pi_{h-1}^b(A_{h-1}|S_{h-1})}\cdot\sum_{a'\in\mathcal{A}}b_h^\pi(a', W_h)\pi_{h-1}(A_{h-1}|O_{h-1}, \Gamma_{h-1})\right].
$$

In the sequel, we prove that $\Delta_h^1 = \Delta_h^2$ for the three cases of $T_h$ in Example 2.1, 2.2, and 2.3, respectively.

**Case 1: Reactive policy (Example 2.1).** We first focus on the simple case when policy $\pi$ is reactive. Since for reactive policies $T_h = \varnothing$, we can equivalently write $\mu_h(S_h, \Gamma_{h-1})$ as $\mu_h(S_h) = \mathcal{P}_h^\pi(S_h)/\mathcal{P}_h^b(S_h)$. Now for $\Delta_h^1$, we can rewrite it as

$$
\begin{aligned}
\Delta_h^1 &= \mathbb{E}_{\pi^b}\left[\frac{\mathcal{P}_h^\pi(S_h)}{\mathcal{P}_h^b(S_h)\pi_h^b(A_h|S_h)}\cdot b_h^\pi(A_h, W_h)\right]\\
&\overset{(a)}{=} \int_\mathcal{S}\cancel{\mathcal{P}_h^b(s_h)}\mathrm{d}s_h\sum_{a_h\in\mathcal{A}}\cancel{\pi_h^b(a_h|s_h)}\int_\mathcal{W}\mathcal{P}_h^b(w_h|s_h, a_h)\mathrm{d}w_h\cdot\frac{\mathcal{P}_h^\pi(s_h)}{\cancel{\mathcal{P}_h^b(s_h)}\cancel{\pi_h^b(a_h|s_h)}}b_h^\pi(a_h, w_h)\\
&\overset{(b)}{=} \sum_{a_h\in\mathcal{A}}\int_\mathcal{S}\mathcal{P}_h^\pi(s_h)\mathrm{d}s_h\int_\mathcal{W}\mathcal{P}_h^b(w_h|s_h)\mathrm{d}w_h\cdot b_h^\pi(a_h, w_h).
\end{aligned}
$$

Here step (a) expands the expectation by using integral against corresponding density functions, and step (b) follows from cancelling the same terms and the fact that $W_h \perp A_h|S_h$ under Assumption 3.1. For $\Delta_h^2$, we can also rewrite it as

$$
\begin{aligned}
\Delta_h^2 &= \mathbb{E}_{\pi^b}\left[\frac{\mathcal{P}_{h-1}^\pi(S_{h-1})\pi_{h-1}(A_{h-1}|O_{h-1})}{\mathcal{P}_{h-1}^b(S_{h-1})\pi_{h-1}^b(A_{h-1}|S_{h-1})}\cdot\sum_{a'\in\mathcal{A}}b_h^\pi(a', W_h)\right]\\
&\overset{(a)}{=} \int_\mathcal{S}\cancel{\mathcal{P}_{h-1}^b(s_{h-1})}\mathrm{d}s_{h-1}\int_\mathcal{O}\mathbb{O}_{h-1}(o_{h-1}|s_{h-1})\mathrm{d}o_{h-1}\sum_{a_{h-1}\in\mathcal{A}}\cancel{\pi_{h-1}^b(a_{h-1}|s_{h-1})}\int_\mathcal{S}\mathbb{P}_h(s_h|s_{h-1}, a_{h-1})\mathrm{d}s_h\\
&\qquad\int_\mathcal{W}\mathcal{P}_h^b(w_h|s_h, s_{h-1}, a_{h-1}, o_{h-1})\cdot\frac{\mathcal{P}_{h-1}^\pi(s_{h-1})\pi_{h-1}(a_{h-1}|o_{h-1})}{\cancel{\mathcal{P}_{h-1}^b(s_{h-1})}\cancel{\pi_{h-1}^b(a_{h-1}|s_{h-1})}}\sum_{a_h\in\mathcal{A}}b_h^\pi(a_h, w_h)\mathrm{d}w_h.
\end{aligned}
$$

Here step (a) follows from expanding the expectation. It follows that

$$
\begin{aligned}
\Delta_h^2 &\stackrel{(b)}{=} \sum_{a_h \in \mathcal{A}} \int_{\mathcal{S}} \mathcal{P}_{h-1}^{\pi}(s_{h-1}) \mathrm{d}s_{h-1} \int_{\mathcal{O}} \mathbb{O}_{h-1}(o_{h-1}|s_{h-1}) \mathrm{d}o_{h-1} \sum_{a \in \mathcal{A}} \pi_{h-1}(a_{h-1}|o_{h-1}) \\
&\qquad \int_{\mathcal{S}} \mathbb{P}_h(s_h|s_{h-1}, a_{h-1}) \mathrm{d}s' \int_{\mathcal{W}} \mathcal{P}_h^b(w_h|s_h) \cdot b_h^{\pi}(a_h, w_h) \\
&\stackrel{(c)}{=} \sum_{a_h \in \mathcal{A}} \int_{\mathcal{S}} \mathcal{P}_h^{\pi}(s_h) \mathrm{d}s_h \int_{\mathcal{W}} \mathcal{P}_h^b(w_h|s_h) \cdot b_h^{\pi}(a_h, w_h) \mathrm{d}w_h.
\end{aligned}
$$

Here step (b) follows from cancelling the same terms and using the fact that $W_h \perp S_{h-1}, A_{h-1}, O_{h-1}|S_h$ by Assumption 3.1, and step (d) follows by marginalizing over $S_{h-1}, A_{h-1}, O_{j-1}$. Thus we have proved that $\Delta_h^1 = \Delta_h^2$ for reactive policies and consequently $\Delta_h = \Delta_h^1 - \Delta_h^2 = 0$.

**Case 2: Finite-history policy (Example 2.2).** Now we have that $\Gamma_{h-1} \cup \{A_h, O_h\} = \{A_{l-1}, O_{l-1}\} \cup T_h$, where the index $l = \max\{0, h-k\}$. Similarly, we can first rewrite $\Delta_h^1$ as

$$
\begin{aligned}
\Delta_h^1 &= \mathbb{E}_{\pi^b}\left[ \frac{\mathcal{P}_h^{\pi}(S_h, \Gamma_{h-1})}{\mathcal{P}_h^b(S_h, \Gamma_{h-1})\pi_h^b(A_h|S_h)} b_h^{\pi}(A_h, W_h) \right] \\
&\stackrel{(a)}{=} \int_{\mathcal{S} \times \mathcal{H}_{h-1}} \mathcal{P}_h^b(s_h, \tau_{h-1}) \mathrm{d}s_h \mathrm{d}\tau_{h-1} \sum_{a_h \in \mathcal{A}} \pi_h^b(a_h|s_h) \int_{\mathcal{W}} \mathcal{P}_h^b(w_h|s_h, a_h, \tau_{h-1}) \mathrm{d}w_h \\
&\qquad \cdot \frac{\mathcal{P}_h^{\pi}(s_h, \tau_{h-1})}{\mathcal{P}_h^b(s_h, \tau_{h-1})\pi_h^b(a_h|s_h, \tau_{h-1})} b_h^{\pi}(a_h, w_h) \\
&\stackrel{(b)}{=} \sum_{a_h \in \mathcal{A}} \int_{\mathcal{S} \times \mathcal{H}_{h-1}} \mathcal{P}_h^{\pi}(s_h, \tau_{h-1}) \mathrm{d}s_h \mathrm{d}\tau_{h-1} \int_{\mathcal{W}} \mathcal{P}_h^b(w_h|s_h) \mathrm{d}w_h \cdot b_h^{\pi}(a_h, w_h).
\end{aligned}
$$

Here step (a) follows from expanding the expectation, and step (b) follows from cancelling the same terms and using the fact that $W_h \perp A_h, \Gamma_{h-1}|S_h$ under Assumption 3.1. For $\Delta_h^2$, we can also rewrite it as

$$
\begin{aligned}
\Delta_h^2 &= \mathbb{E}_{\pi^b}\left[ \frac{\mathcal{P}_{h-1}^{\pi}(S_{h-1}, \Gamma_{h-2})\pi_{h-1}(A_{h-1}|O_{h-1})}{\mathcal{P}_{h-1}^b(S_{h-1}, \Gamma_{h-2})\pi_{h-1}^b(A_{h-1}|S_{h-1}, \Gamma_{h-2})} \sum_{a' \in \mathcal{A}} b_h^{\pi}(a', W_h) \right] \\
&\stackrel{(a)}{=} \int_{\mathcal{S} \times \mathcal{H}_{h-2}} \mathcal{P}_{h-1}^b(s_{h-1}, \tau_{h-2}) \mathrm{d}s_{h-1} \mathrm{d}\tau_{h-2} \int_{\mathcal{O}} \mathbb{O}_{h-1}(o_{h-1}|s_{h-1}) \mathrm{d}o_{h-1} \sum_{a_{h-1} \in \mathcal{A}} \pi_{h-1}^b(a_{h-1}|s_{h-1}) \\
&\qquad \int_{\mathcal{S}} \mathbb{P}_h(s_h|s_{h-1}, a_{h-1}) \mathrm{d}s_h \int_{\mathcal{W}} \mathcal{P}_h^b(w_h|s_h, s_{h-1}, a_{h-1}, o_{h-1}, \tau_{h-2}) \\
&\qquad \cdot \frac{\mathcal{P}_{h-1}^{\pi}(s_{h-1}, \tau_{h-2})\pi_{h-1}(a_{h-1}|o_{h-1}, \tau_{h-2})}{\mathcal{P}_{h-1}^b(s_{h-1}, \tau_{h-2})\pi_{h-1}^b(a_{h-1}|s_{h-1})} \sum_{a_h \in \mathcal{A}} b_h^{\pi}(a_h, w_h) \\
&\stackrel{(b)}{=} \sum_{a_h \in \mathcal{A}} \int_{\mathcal{S} \times \mathcal{H}_{h-2}} \mathcal{P}_{h-1}^{\pi}(s_{h-1}, \widetilde{\tau}_{h-2}, a_l, o_l) \mathrm{d}s_{h-1} \mathrm{d}\widetilde{\tau}_{h-2} \mathrm{d}a_l \mathrm{d}o_l \int_{\mathcal{O}} \mathbb{O}_{h-1}(o_{h-1}|s_{h-1}) \mathrm{d}o_{h-1} \\
&\qquad \sum_{a_{h-1} \in \mathcal{A}} \pi_{h-1}(a_{h-1}|o_{h-1}, \tau_{h-2}) \int_{\mathcal{S}} \mathbb{P}_h(s_h|s_{h-1}, a_{h-1}) \mathrm{d}s_h \int_{\mathcal{W}} \mathcal{P}_h^b(w_h|s_h) \cdot b_h^{\pi}(a_h, w_h) \\
&\stackrel{(c)}{=} \sum_{a_h \in \mathcal{A}} \int_{\mathcal{S} \times \mathcal{H}_{h-1}} \mathcal{P}_h^{\pi}(s_h, \tau_{h-1}) \mathrm{d}s_h \mathrm{d}\tau_{h-1} \int_{\mathcal{W}} \mathcal{P}_h^b(w_h|s_h) \cdot b_h^{\pi}(a_h, w_h),
\end{aligned} \tag{E.2}
$$

where the index $l = \max\{1, h-1-k\}$. In step (b), we have denoted by $\widetilde{\tau}_{h-2} = \tau_{h-2} \setminus \{a_l, o_l\}$ and it holds that $\tau_{h-1} = \widetilde{\tau}_{h-2} \cup \{o_{h-1}, a_{h-1}\}$. Here step (a) follows from expanding the expectation, step (b) follows from cancelling the same terms and using the fact that $W_h \perp S_{h-1}, A_{h-1}, \Gamma_{h-1}|S_h$ under Assumption 3.1, and step (c) follows by marginalizing $S_{h-1}, A_l, O_l$. Thus we have proved that $\Delta_h^1 = \Delta_h^2$ for finite-length history policies and consequently $\Delta_h = \Delta_h^1 - \Delta_h^2 = 0$.

**Case 3: Full-history policy (Example 2.3).** For full history information $T_h$, we have that $\Gamma_{h-1} \cup \{A_h, O_h\} = T_h$. Following the same argument as in Case 2 (Example 2.2), we can first show that

$$\Delta_h^1 = \sum_{a_h \in \mathcal{A}} \int_{\mathcal{S} \times \mathcal{H}_{h-1}} \mathcal{P}_h^\pi(s_h, \tau_{h-1}) \mathrm{d}s_h \mathrm{d}\tau_{h-1} \int_{\mathcal{W}} \mathcal{P}_h^b(w_h|s_h) \mathrm{d}w_h \cdot b_h^\pi(a_h, w_h).$$

Besides, for $\Delta_2$, by a similar argument as in Case 2 except that we don't need marginalize over $A_l, O_l$ in (E.2), we can show that

$$\Delta_h^2 = \sum_{a_h \in \mathcal{A}} \int_{\mathcal{S} \times \mathcal{H}_{h-2}} \mathcal{P}_{h-1}^\pi(s_{h-1}, \tau_{h-2}) \mathrm{d}s_{h-1} \mathrm{d}\tau_{h-2} \int_{\mathcal{O}} \mathbb{O}_{h-1}(o_{h-1}|s_{h-1}) \mathrm{d}o_{h-1} \sum_{a_{h-1} \in \mathcal{A}} \pi_{h-1}(a_{h-1}|o_{h-1}, \tau_{h-2})$$

$$\int_{\mathcal{S}} \mathbb{P}_h(s_h|s_{h-1}, a_{h-1}) \mathrm{d}s_h \int_{\mathcal{W}} \mathcal{P}_h^b(w_h|s_h, s_{h-1}, a_{h-1}, o_{h-1}, \tau_{h-2}) \cdot b_h^\pi(a_h, w_h)$$

$$= \sum_{a_h \in \mathcal{A}} \int_{\mathcal{S} \times \mathcal{H}_{h-1}} \mathcal{P}_h^\pi(s_h, \tau_{h-1}) \mathrm{d}s_h \mathrm{d}\tau_{h-1} \int_{\mathcal{W}} \mathcal{P}_h^b(w_h|s_h) \mathrm{d}w_h \cdot b_h^\pi(a_h, w_h).$$

Therefore, we show that $\Delta_h^1 = \Delta_h^2$ for full history policies and consequently $\Delta_h = \Delta_h^1 - \Delta_h^2 = 0$.

Now we have shown that term (B) in (E.1) is actually $0$ for Example 2.1, Example 2.2, and Example 2.3, respectively, which allows us to conclude that

$$J(\pi) = (A) = \mathbb{E}_{\pi^b} \left[ \sum_{a \in \mathcal{A}} b_1^\pi(a, W_1) \right].$$

This finishes the proof of Theorem 3.3. $\qquad \square$

## F  PROOF OF LEMMAS IN SECTION D

We first review and define several notations and quantities that are useful in the proof of the lemmas in Section D. Firstly, we define mapping $\ell_h^\pi : \mathbb{B} \times \mathbb{B} \mapsto \{\mathcal{A} \times \mathcal{Z} \mapsto \mathbb{R}\}$ as

$$\ell_h^\pi(b_h, b_{h+1})(A_h, Z_h) := \mathbb{E}_{\pi^b} \Big[ b_h(A_h, W_h) - R_h \pi_h(A_h|O_h, \Gamma_{h-1})$$
$$- \gamma \sum_{a' \in \mathcal{A}} b_{h+1}(a', W_{h+1}) \pi_h(A_h|O_h, \Gamma_{h-1}) \Big| A_h, Z_h \Big]. \qquad \text{(F.1)}$$

Furthermore, for each step $h \in [H]$, we define a joint space $\mathcal{I}_h = \mathcal{A} \times \mathcal{W} \times \mathcal{O} \times \mathcal{H}_{h-1} \times \mathcal{W}$ and define mapping $\varsigma_h^\pi : \mathbb{B} \times \mathbb{B} \mapsto \{\mathcal{I}_h \mapsto \mathbb{R}\}$ as

$$\varsigma_h^\pi(b_h, b_{h+1})(A_h, W_h, O_h, \Gamma_{h-1}, W_{h+1}) := b_h(A_h, W_h) - R_h \pi_h(A_h|O_h, \Gamma_{h-1})$$
$$- \gamma \sum_{a' \in \mathcal{A}} b_{h+1}(a', W_{h+1}) \pi_h(A_h|O_h, \Gamma_{h-1}). \qquad \text{(F.2)}$$

When appropriate, we abbreviate $I_h = (A_h, W_h, O_h, \Gamma_{h-1}, W_{h+1}) \in \mathcal{I}_h$ in the sequel. Using definition (F.1) and (F.2), we further introduce two mappings $\Phi_{\pi,h}^\lambda, \Phi_{\pi,h} : \mathbb{B} \times \mathbb{B} \times \mathbb{G} \mapsto \mathbb{R}$ as defined by (3.9),

$$\Phi_{\pi,h}^\lambda(b_h, b_{h+1}; g) := \mathbb{E}_{\pi^b} \big[ \ell_h^\pi(b_h, b_{h+1})(A_h, Z_h) \cdot g(A_h, Z_h) - \lambda g(A_h, Z_h)^2 \big],$$

$$\Phi_{\pi,h}(b_h, b_{h+1}; g) := \Phi_{\pi,h}^0(b_h, b_{h+1}; g) = \mathbb{E}_{\pi^b} \big[ \ell_h^\pi(b_h, b_{h+1})(A_h, Z_h) \cdot g(A_h, Z_h) \big],$$

where we define that $\Phi_{\pi,h} = \Phi_{\pi,h}^0$. Also, recall from (3.10) that the empirical version of $\Phi_{\pi,h}^\lambda, \Phi_{\pi,h}$ are defined by $\widehat{\Phi}_{\pi,h}^\lambda, \widehat{\Phi}_{\pi,h}$ as

$$\widehat{\Phi}_{\pi,h}^\lambda(b_h, b_{h+1}; g) := \widehat{\mathbb{E}}_{\pi^b} \big[ \varsigma_h^\pi(b_h, b_{h+1})(I_h) \cdot g(A_h, Z_h) - \lambda g(A_h, Z_h)^2 \big],$$

$$\widehat{\Phi}_{\pi,h}(b_h, b_{h+1}; g) := \widehat{\Phi}_{\pi,h}^0(b_h, b_{h+1}; g) = \widehat{\mathbb{E}}_{\pi^b} \big[ \varsigma_h^\pi(b_h, b_{h+1})(I_h) \cdot g(A_h, Z_h) \big].$$

Recall from (3.11) that given function $b_{h+1} \in \mathbb{B}$, the minimax estimator $\widehat{b}_h(b_{h+1})$ is defined as

$$\widehat{b}_h(b_{h+1}) := \arg\min_{b \in \mathbb{B}} \max_{g \in \mathbb{G}} \widehat{\Phi}^\lambda_{\pi,h}(b, b_{h+1}; g).$$

Meanwhile, we define the following quantity for ease of theoretical analysis as

$$b_h^\star(b_{h+1}) := \arg\min_{b \in \mathbb{B}} \max_{g \in \mathbb{G}} \Phi^\lambda_{\pi,h}(b, b_{h+1}; g). \tag{F.3}$$

By the boundedness assumption on $\mathbb{B}$ in Assumption 4.2, we have that $|\ell^\pi_h|, |\varsigma^\pi_h| \leq 2M_{\mathbb{B}}$. By the completeness assumption on $\mathbb{G}$ in Assumption 4.3, we also know that $\ell^\pi_h(b_h, b_{h+1})/2\lambda \in \mathbb{G}$ for any $b_h, b_{h+1} \in \mathbb{B}$. Finally, for notational simplicity, we define for each $g \in \mathbb{G}$ that,

$$\|g\|_2^2 := \mathbb{E}_{\pi^b}[g(A_h, Z_h)^2],$$

and we denote by $\|g\|_{2,n}^2$ its empirical version, i.e.,

$$\|g\|_{2,n}^2 := \widehat{\mathbb{E}}_{\pi^b}[g(A_h, Z_h)^2].$$

We remark that we have dropped the dependence of $\|g\|_2^2$ on step $h$ since it is clear from the context when used in the proofs and does not make any confusion.

## F.1 PROOF OF LEMMA D.1

*Proof of Lemma D.1.* By definition (D.1) of $F(\mathbf{b})$, for any policy $\pi \in \Pi(\mathcal{H})$ and vector of functions $\mathbf{b} \in \mathbb{B}^{\otimes H}$, it holds that

$$
\begin{aligned}
F(\mathbf{b}^\pi) - F(\mathbf{b}) &\overset{(a)}{=} \mathbb{E}_{\pi^b}\left[\sum_{a \in \mathcal{A}} b_1^\pi(a, W_1) - b_1(a, W_1)\right] \\
&= \mathbb{E}_{\pi^b}\left[\sum_{a \in \mathcal{A}} \frac{\pi_1^b(a|S_1)}{\pi_1^b(a|S_1)}\left(b_1^\pi(a, W_1) - b_1(a, W_1)\right)\right] \\
&\overset{(b)}{=} \mathbb{E}_{\pi^b}\left[\mathbb{E}_{\pi^b}\left[\frac{1}{\pi_1^b(A_1|S_1)}\left(b_1^\pi(a, W_1) - b_1(a, W_1)\right)\Big| S_1, W_1\right]\right] \\
&= \mathbb{E}_{\pi^b}\left[\frac{1}{\pi_1^b(A_1|S_1)}\left(b_1^\pi(A_1, W_1) - b_1(A_1, W_1)\right)\right]
\end{aligned}
$$

where step (a) follows from Theorem 3.3 and (D.1), and step (b) holds since $A_1 \perp W_1 \,|\, S_1$ by Assumption 3.1. Notice that by definition (3.3), at step $h = 1$, the weight bridge function $q_h^\pi$ satisfies equation

$$\mathbb{E}_{\pi^b}[q_1^\pi(A_1, Z_1)|A_1, S_1, \Gamma_0] = \frac{\mathcal{P}_h^\pi(S_1, \Gamma_0)}{\mathcal{P}_h^\pi(S_1, \Gamma_0)\pi_1^b(A_1|S_1)} = \frac{1}{\pi_1^b(A_1|S_1)},$$

which further gives that

$$
\begin{aligned}
F(\mathbf{b}^\pi) - F(\mathbf{b}) &= \mathbb{E}_{\pi^b}\left[\mathbb{E}_{\pi^b}\left[q_1^\pi(A_1, Z_1)|A_1, S_1, \Gamma_0\right]\left(b_1^\pi(A_1, W_1) - b_1(A_1, W_1)\right)\right] \\
&\overset{(a)}{=} \mathbb{E}_{\pi^b}\left[\mathbb{E}_{\pi^b}\left[q_1^\pi(A_1, Z_1)|A_1, S_1, W_1, \Gamma_0\right] \cdot \left(b_1^\pi(A_1, W_1) - b_1(A_1, W_1)\right)\right] \\
&= \mathbb{E}_{\pi^b}\left[q_1^\pi(A_1, Z_1)\left(b_1^\pi(A_1, W_1) - b_1(A_1, W_1)\right)\right],
\end{aligned}
$$

where step (a) holds since $Z_1 \perp W_1 \,|\, A_1, S_1, \mathcal{H}_0$ by Assumption 3.1. Now we can further obtain that,

$$
\begin{aligned}
F(\mathbf{b}^\pi) - F(\mathbf{b}) &= \mathbb{E}_{\pi^b}\left[q_1^\pi(A_1, Z_1)\mathbb{E}_{\pi^b}\left[b_1^\pi(A_1, W_1) - b_1(A_1, W_1)|A_1, Z_1\right]\right] \\
&\overset{(a)}{=} \mathbb{E}_{\pi^b}\left[q_1^\pi(A_1, Z_1)\left\{\mathbb{E}_{\pi^b}\left[R_1\pi_1(A_1|O_1, \Gamma_0) + \gamma\sum_{a'} b_2^\pi(a', W_2)\pi_1(A_1|O_1, \Gamma_0)\Big|A_1, Z_1\right]\right.\right. \\
&\qquad\left.\left. - \mathbb{E}_{\pi^b}\left[b_1(A_1, W_1)|A_1, Z_1\right]\right\}\right],
\end{aligned}
$$

where step (a) follows from the definition in (3.2) of value bridge function $b_1^\pi$ in Assumption 3.2. Now to relate the difference between $F(\mathbf{b}^\pi)$ and $F(\mathbf{b})$ with the RMSE loss $\mathcal{L}_1^\pi$ defined in (3.6), we rewrite the above equation as the following,

$$
\begin{aligned}
&F(\mathbf{b}^\pi) - F(\mathbf{b}) \\
&= \mathbb{E}_{\pi^b}\bigg[q_1^\pi(A_1, Z_1)\bigg\{\mathbb{E}_{\pi^b}\bigg[R_1\pi_h(A_1|O_1,\Gamma_0) + \gamma\sum_{a'} b_2^\pi(a', W_2)\pi_1(A_1|O_1,\Gamma_0)\bigg|A_1, Z_1\bigg] \\
&\qquad - \mathbb{E}_{\pi^b}\bigg[R_h\pi_1(A_1|O_1,\Gamma_0) + \gamma\sum_{a'} b_2(a', W_{h+1})\pi_1(A_1|O_1,\Gamma_0)\bigg|A_1, Z_1\bigg] \\
&\qquad + \mathbb{E}_{\pi^b}\bigg[R_1\pi_1(A_1|O_1,\Gamma_0) + \gamma\sum_{a'} b_2(a', W_2)\pi_1(A_1|O_1,\Gamma_0)\bigg|A_1, Z_1\bigg] \\
&\qquad - \mathbb{E}_{\pi^b}\bigg[b_1(A_1, W_1)\bigg|A_1, Z_1\bigg]\bigg\}\bigg] \\
&= \mathbb{E}_{\pi^b}\bigg[q_1^\pi(A_1, Z_1)\bigg\{\gamma\mathbb{E}_{\pi^b}\bigg[\sum_{a'}\Big(b_2^\pi(a', W_2) - b_2(a, W_2)\Big)\pi_1(A_1|O_1,\Gamma_0)\bigg|A_1, Z_1\bigg] \\
&\qquad + \mathbb{E}_{\pi^b}\bigg[R_1\pi_h(A_1|O_1,\Gamma_0) + \gamma\sum_{a'} b_2(a', W_2)\pi_h(A_1|O_1,\Gamma_0) - b_1(A_1, W_1)\bigg|A_1, Z_1\bigg]\bigg\}\bigg].
\end{aligned}
$$
(F.4)

We deal with the two terms in the right-hand side of (F.4) respectively. On the one hand, the first term equals to

$$
\begin{aligned}
&\gamma\mathbb{E}_{\pi^b}\bigg[q_1^\pi(A_1, Z_1)\mathbb{E}_{\pi^b}\bigg[\sum_{a'}\Big(b_2^\pi(a', W_2) - b_2(a, W_2)\Big)\pi_1(A_1|O_1,\Gamma_0)\bigg|A_1, Z_1\bigg]\bigg] \\
&= \gamma\mathbb{E}_{\pi^b}\bigg[q_1^\pi(A_1, Z_1)\sum_{a'}\Big(b_2^\pi(a', W_2) - b_2(a, W_2)\Big)\pi_1(A_1|O_1,\Gamma_0)\bigg] \\
&= \gamma\mathbb{E}_{\pi^b}\bigg[\mathbb{E}_{\pi^b}\bigg[q_1^\pi(A_1, Z_1)\bigg|S_1, A_1, \Gamma_0, O_1, W_2\bigg]\sum_{a'}\Big(b_2^\pi(a', W_2) - b_2(a, W_2)\Big)\pi_1(A_1|O_1,\Gamma_0)\bigg] \\
&\overset{(a)}{=} \gamma\mathbb{E}_{\pi^b}\bigg[\mathbb{E}_{\pi^b}\bigg[q_1^\pi(A_1, Z_1)\bigg|S_1, A_1, \Gamma_0\bigg]\sum_{a'}\Big(b_2^\pi(a', W_2) - b_2(a, W_2)\Big)\pi_1(A_1|O_1,\Gamma_0)\bigg] \\
&\overset{(b)}{=} \gamma\mathbb{E}_{\pi^b}\bigg[\frac{\mu_1(S_1, \Gamma_0)}{\pi_1^b(A_1|S_1)}\sum_{a'}\Big(b_2^\pi(a', W_2) - b_2(a, W_2)\Big)\pi_1(A_1|O_1,\Gamma_0)\bigg],
\end{aligned}
$$

where step (a) follows from the fact that $Z_1 \perp O_1, W_2|S_1, A_1, \Gamma_0$ according to Assumption 3.1, and step (b) follows from the definition (3.3) of weight bridge function $q_1^\pi$ in Assumption 3.2. Now following the same argument as in showing $\Delta_h = 0$ in the proof of Theorem 3.3, we can show that

$$
\begin{aligned}
&\mathbb{E}_{\pi^b}\bigg[\frac{\mu_1(S_1, \Gamma_0)}{\pi_1^b(A_1|S_1)}\sum_{a'}\Big(b_2^\pi(a', W_2) - b_2(a, W_2)\Big)\pi_1(A_1|O_1,\Gamma_0)\bigg] \\
&\qquad = \mathbb{E}_{\pi^b}\bigg[q_2^\pi(A_2, Z_2)\Big(b_2^\pi(A_2, W_2) - b_2(A_2, W_2)\Big)\bigg].
\end{aligned}
$$
(F.5)

On the other hand, the second term in the R.H.S. of (F.4) can be rewritten and bounded by

$$
\begin{aligned}
&\mathbb{E}_{\pi^b}\bigg[q_1^\pi(A_1, Z_1)\mathbb{E}_{\pi^b}\bigg[R_1\pi_1(A_1|O_1,\Gamma_0) + \gamma\sum_{a'} b_2(a', W_2)\pi_1(A_1|O_1,\Gamma_0) - b_1(A_1, W_1)\bigg|A_1, Z_1\bigg]\bigg] \\
&\leq \sqrt{C^\pi}\mathbb{E}_{\pi^b}\bigg[\bigg\{\mathbb{E}_{\pi^b}\bigg[R_1\pi_1(A_1|O_1,\Gamma_0) + \gamma\sum_{a'} b_2(a', W_2)\pi_1(A_1|O_1,\Gamma_0) - b_1(A_1, W_1)\bigg|A_1, Z_1\bigg]\bigg\}^{1/2}\bigg] \\
&= \sqrt{C^\pi} \cdot \sqrt{\mathcal{L}_1^\pi(b_1, b_2)},
\end{aligned}
$$
(F.6)

where $C^\pi$ is defined as $C^\pi := \sup_{h \in [H]} \mathbb{E}_{\pi^b} \left[ (q_h^\pi(A_h, Z_h))^2 \right]$, the inequality follows from Cauchy-Schwarz inequality, and the equality follows from the definition of $\mathcal{L}_1^\pi$ in (3.6). Combining (F.4), (F.5) with (F.6), we can obtain that

$$F(\mathbf{b}^\pi) - F(\mathbf{b})$$
$$\leq \sqrt{C^\pi} \cdot \sqrt{\mathcal{L}_1^\pi(b_1, b_2)} + \gamma \mathbb{E}_{\pi^b} \left[ q_2^\pi(A_2, Z_2) \Big( b_2^\pi(A_2, W_2) - b_2(A_2, W_2) \Big) \right]. \tag{F.7}$$

Now applying the above argument on the second term in the R.H.S. of (F.7) recursively, we can obtain that

$$F(\mathbf{b}^\pi) - F(\mathbf{b}) \leq \sum_{h=1}^{H} \gamma^{h-1} \sqrt{C^\pi} \cdot \sqrt{\mathcal{L}_h^\pi(b_h, b_{h+1})}.$$

This finishes the proof of Lemma D.1. $\qquad\square$

### F.2 PROOF OF LEMMA D.2

*Proof of Lemma D.2.* By the definition of the confidence region $\mathrm{CR}^\pi(\alpha)$ in (3.12), we need to show for any policy $\pi \in \Pi(\mathcal{H})$ and step $h \in [H]$, it holds that,

$$\max_{g \in \mathbb{G}} \widehat{\Phi}_{\pi,h}^\lambda(b_h^\pi, b_{h+1}^\pi; g) - \max_{g \in \mathbb{G}} \widehat{\Phi}_{\pi,h}^\lambda(\widehat{b}_h(b_{h+1}^\pi), b_{h+1}^\pi; g) \leq \xi. \tag{F.8}$$

Notice that by Assumption 4.2, the function class $\mathbb{G}$ is symmetric and star-shaped, which indicates that

$$\max_{g \in \mathbb{G}} \widehat{\Phi}_{\pi,h}^\lambda(\widehat{b}_h(b_{h+1}^\pi), b_{h+1}^\pi; g) \geq \widehat{\Phi}_{\pi,h}^\lambda(\widehat{b}_h(b_{h+1}^\pi), b_{h+1}^\pi; 0) = 0.$$

Therefore, in order to prove (F.8), it suffices to show that

$$\max_{g \in \mathbb{G}} \widehat{\Phi}_{\pi,h}^\lambda(b_h^\pi, b_{h+1}^\pi; g) \leq \xi. \tag{F.9}$$

To relate the empirical expectation $\widehat{\Phi}_{\pi,h}^\lambda(b_h^\pi, b_{h+1}^\pi; g) = \widehat{\Phi}_{\pi,h}(b_h^\pi, b_{h+1}^\pi; g) - \lambda \|g\|_{2,n}^2$ to its population version, we need two localized uniform concentration inequalities. On the one hand, to relate $\|g\|_2^2$ and $\|g\|_{2,n}^2$, by Lemma I.1 (Theorem 14.1 of Wainwright (2019)), for some absolute constants $c_1, c_2 > 0$, it holds with probability at least $1 - \delta/2$ that,

$$\left| \|g\|_{2,n}^2 - \|g\|_2^2 \right| \leq \frac{1}{2} \|g\|_2^2 + \frac{M_\mathbb{G}^2 \log(2c_1/\zeta)}{2c_2 n}, \quad \forall g \in \mathbb{G}, \tag{F.10}$$

where $\zeta = \min\{\delta, 2c_1 \exp(-c_2 n \alpha_{\mathbb{G},n}^2 / M_\mathbb{G}^2)\}$ and $\alpha_{\mathbb{G},n}$ is the critical radius of function class $\mathbb{G}$ defined in Assumption 4.2. On the other hand, to relate $\widehat{\Phi}_{\pi,h}(b_h, b_{h+1}; g)$ and $\Phi_{\pi,h}(b_h, b_{h+1}; g)$ we invoke Lemma I.2 (Lemma 11 of (Foster and Syrgkanis, 2019)). Specifically, for any given $b_h, b_{h+1} \in \mathbb{B}$, $\pi \in \Pi(\mathcal{H})$, and $h \in [H]$, in Lemma I.2 we choose $\mathcal{F} = \mathbb{G}$, $\mathcal{X} = \mathcal{A} \times \mathcal{Z}$, $\mathcal{Y} = \mathcal{I}_h$, and loss function $\ell(g(A_h, Z_h), I_h) := \varsigma_h^\pi(b_h, b_{h+1})(I_h) \cdot g(A_h, Z_h)$ where $\varsigma_h^\pi$ is defined in (F.1), $I_h \in \mathcal{I}_h$ is defined in the beginning of Appendix F. It holds that $\ell$ is $L$-Lipschitz continuous in the first argument since for any $g, g' \in \mathbb{G}$, $(A_h, Z_h) \in \mathcal{A} \times \mathcal{Z}$, it holds that

$$\left| \ell(g(A_h, Z_h), I_h) - \ell(g'(A_h, Z_h), I_h) \right| = |\varsigma_h^\pi(b_h, b_{h+1})(I_h)| \cdot |g(A_h, Z_h) - g'(A_h, Z_h)|$$
$$\leq 2M_\mathbb{B} \cdot |g(A_h, Z_h) - g'(A_h, Z_h)|,$$

which indicates that $L = 2M_\mathbb{B}$. Now setting $f^\star = 0$ in Lemma I.2, we have that $\delta_n$ in Lemma I.2 coincides with $\alpha_{\mathbb{G},n}$ in Assumption 4.2. Then we can conclude that for some absolute constants $c_1, c_2 > 0$, it holds with probability at least $1 - \delta/(2|\mathbb{B}|^2|\Pi(\mathcal{H})|H)$ that

$$\left| \widehat{\Phi}_{\pi,h}(b_h, b_{h+1}; g) - \Phi_{\pi,h}(b_h, b_{h+1}; g) \right|$$
$$= \left| \widehat{\mathbb{E}}_{\pi^b}[\ell(g(A_h, Z_h), I_h)] - \mathbb{E}_{\pi^b}[\ell(g(A_h, Z_h), I_h)] \right|$$
$$\leq 18L\|g\|_2 \sqrt{\frac{M_\mathbb{G}^2 \log\left(2c_1|\mathbb{B}|^2|\Pi(\mathcal{H})|H/\zeta'\right)}{c_2 n}} + \frac{18LM_\mathbb{G}^2 \log\left(2c_1|\mathbb{B}|^2|\Pi(\mathcal{H})|H/\zeta'\right)}{c_2 n}, \quad \forall g \in \mathbb{G}, \tag{F.11}$$

where $\zeta' = \min\{\delta, 2c_1|\mathbb{B}|^2|\Pi(\mathcal{H})|H\exp(-c_2n\alpha_{\mathbb{G},n}^2/M_{\mathbb{G}}^2)\}$. Applying a union bound argument over $b_h, b_{h+1} \in \mathbb{B}$, $\pi \in \Pi(\mathcal{H})$, and $h \in [H]$, we then have that (F.11) holds for any $b_h, b_{h+1} \in \mathbb{B}$, $g \in \mathbb{G}$, $\pi \in \Pi(\mathcal{H})$, and $h \in [H]$ with probability at least $1 - \delta/2$. Now using these two concentration inequalities (F.10) and (F.11), we can further deduce that, for some absolute constants $c_1, c_2 > 0$, with probability at least $1 - \delta$,

$$\max_{g \in \mathbb{G}} \widehat{\Phi}_{\pi,h}^{\lambda}(b_h^{\pi}, b_{h+1}^{\pi}; g)$$

$$= \max_{g \in \mathbb{G}} \left\{ \widehat{\Phi}_{\pi,h}(b_h^{\pi}, b_{h+1}^{\pi}; g) - \lambda\|g\|_{2,n}^2 \right\}$$

$$\leq \max_{g \in \mathbb{G}} \left\{ \Phi_{\pi,h}(b_h^{\pi}, b_{h+1}^{\pi}; g) - \lambda\|g\|_2^2 + \frac{\lambda}{2}\|g\|_2^2 + \frac{\lambda M_{\mathbb{G}}^2\log(2c_1/\zeta)}{2c_2 n}, \right.$$

$$\left. + 18L\|g\|_2 \sqrt{\frac{M_{\mathbb{G}}^2\log(2c_1|\mathbb{B}|^2|\Pi(\mathcal{H})|H/\zeta')}{c_2 n}} + \frac{18LM_{\mathbb{G}}^2\log(2c_1|\mathbb{B}|^2|\Pi(\mathcal{H})|H/\zeta')}{c_2 n} \right\},$$

where $\zeta$ is given as $\zeta = \min\{\delta, 2c_1\exp(-c_2n\alpha_{\mathbb{G},n}^2/M_{\mathbb{G}}^2)\}$ and $\zeta'$ is given as $\zeta' = \min\{\delta, 2c_1|\mathbb{B}|^2|\Pi(\mathcal{H})|H\exp(-c_2n\alpha_{\mathbb{G},n}^2/M_{\mathbb{G}}^2)\}$ for any policy $\pi \in \Pi(\mathcal{H})$ and step $h$. Then we can further bound the right-hand side of the above inequality as

$$\max_{g \in \mathbb{G}} \widehat{\Phi}_{\pi,h}^{\lambda}(b_h^{\pi}, b_{h+1}^{\pi}; g)$$

$$\leq \max_{g \in \mathbb{G}} \Phi_{\pi,h}(b_h^{\pi}, b_{h+1}^{\pi}; g) + \max_{g \in \mathbb{G}} \left\{ -\frac{\lambda}{2}\|g\|_2^2 + 18L\|g\|_2 \sqrt{\frac{M_{\mathbb{G}}^2 \cdot \log(2c_1|\mathbb{B}|^2|\Pi(\mathcal{H})|H/\zeta')}{c_2 n}} \right\}$$

$$+ \frac{\lambda M_{\mathbb{G}}^2 \cdot \log(2c_1/\zeta)}{2c_2 n} + \frac{18LM_{\mathbb{G}}^2 \cdot \log(2c_1|\mathbb{B}|^2|\Pi(\mathcal{H})|H/\zeta')}{c_2 n}$$

$$\leq \frac{728L^2 \cdot M_{\mathbb{G}}^2 \cdot \log(2c_1|\mathbb{B}|^2|\Pi(\mathcal{H})|H/\zeta')}{\lambda n} + \frac{\lambda M_{\mathbb{G}}^2 \cdot \log(2c_1/\zeta)}{2c_2 n}$$

$$+ \frac{18LM_{\mathbb{G}}^2 \cdot \log(2c_1|\mathbb{B}|^2|\Pi(\mathcal{H})|H/\zeta')}{c_2 n}.$$

Here the last inequality holds from the fact that $\Phi_{\pi,h}(b_h^{\pi}, b_{h+1}^{\pi}; g) = 0$ since $b_h^{\pi}$ and $b_{h+1}^{\pi}$ are true bridge functions, and the fact that $\sup_{\|g\|_2}\{a\|g\|_2 - b\|g\|_2^2\} \leq a^2/4b$ for any $b > 0$. Now according to the choice of $\xi$ in Lemma D.2, using the fact that $\zeta < \zeta'$ and $L = 2M_{\mathbb{B}}$, we can conclude that, with probability at least $1 - \delta$,

$$\max_{g \in \mathbb{G}} \widehat{\Phi}_{\pi,h}^{\lambda}(b_h^{\pi}, b_{h+1}^{\pi}; g)$$

$$\leq \frac{728L^2 M_{\mathbb{G}}^2 \cdot \log(2c_1|\mathbb{B}|^2|\Pi(\mathcal{H})|H/\zeta')}{\lambda n} + \frac{\lambda M_{\mathbb{G}}^2 \cdot \log(2c_1/\zeta)}{2c_2 n} + \frac{18LM_{\mathbb{G}}^2 \cdot \log(2c_1|\mathbb{B}|^2|\Pi(\mathcal{H})|H/\zeta')}{c_2 n}$$

$$\lesssim \mathcal{O}\left( \frac{(\lambda + 1/\lambda) \cdot M_{\mathbb{B}}^2 M_{\mathbb{G}}^2 \cdot \log(|\mathbb{B}||\Pi(\mathcal{H})|H/\zeta)}{n} \right) \lesssim \xi.$$

This proves (F.9), and thus further indicates (F.8). Therefore, we finish the proof of Lemma D.2. $\square$

### F.3 Proof of Lemma D.3

We first give the high-level idea for proving Lemma D.3 as following. In order to achieve the fast rate for the whole confidence region, we took a series of novel proof steps.

We first introduce the following lemma, which claims that for any $b_{h+1} \in \mathbb{B}$, the $b^{\star}(b_{h+1})$ defined in (F.3) satisfies that $\max_{g \in \mathbb{G}} \widehat{\Phi}_{\pi,h}^{\lambda}(b^{\star}(b_{h+1}), b_{h+1}; g)$ is well-bounded. The proof of lemma follows the same argument as in the proof of Lemma D.2, which we defer to Appendix F.4.

Then given any bridge function in the confidence region, we identify a key term (term $(\star)$ in (F.12)) which is related to the RMSE of this bridge function. By carefully upper & lower bound this term,

where Lemma F.1 is applied, we eventually obtain a quadratic inequality that the RMSE of this bridge function satisfies. By solving this inequality, we can derive an upper bound on the RMSE loss which is *uniform* over the bridge functions in the confidence region, which is exactly the fast rate of the whole confidence region.

**Lemma F.1.** *For any function $b_{h+1} \in \mathbb{B}$, policy $\pi \in \Pi(\mathcal{H})$, and step $h \in [H]$, it holds with probability at least $1 - \delta/2$ that*

$$\max_{g \in \mathbb{G}} \widehat{\Phi}^\lambda_{\pi,h}(b^\star_h(b_{h+1}), b_{h+1}; g) \leq \xi + \epsilon^{1/2}_{\mathbb{B}} M_{\mathbb{G}},$$

*where $b^\star(b_{h+1})$ is defined in* (F.3) *and $\xi$ is defined in Lemma D.3.*

*Proof of Lemma F.1.* See Appendix F.4 for a detailed proof. $\square$

With Lemma F.1, we are now ready to give the proof of Lemma D.3.

*Proof of Lemma D.3.* Let's consider that for any $b_h, b_{h+1} \in \mathrm{CR}^\pi(\xi)$, we have that

$$\max_{g \in \mathbb{G}} \widehat{\Phi}^\lambda_{\pi,h}(b_h, b_{h+1}; g) = \max_{g \in \mathbb{G}} \Big\{ \widehat{\Phi}_{\pi,h}(b_h, b_{h+1}; g) - \widehat{\Phi}_{\pi,h}(b^\star_h(b_{h+1}), b_{h+1}; g) - 2\lambda \|g\|^2_{2,n}$$
$$+ \widehat{\Phi}_{\pi,h}(b^\star_h(b_{h+1}), b_{h+1}; g) + \lambda \|g\|^2_{2,n} \Big\}.$$

We further write the above as

$$\max_{g \in \mathbb{G}} \widehat{\Phi}^\lambda_{\pi,h}(b_h, b_{h+1}; g) \geq \max_{g \in \mathbb{G}} \Big\{ \widehat{\Phi}_{\pi,h}(b_h, b_{h+1}; g) - \widehat{\Phi}_{\pi,h}(b^\star_h(b_{h+1}), b_{h+1}; g) - 2\lambda \|g\|^2_{2,n} \Big\}$$
$$+ \min_{g \in \mathbb{G}} \Big\{ \widehat{\Phi}_{\pi,h}(b^\star_h(b_{h+1}), b_{h+1}; g) + \lambda \|g\|^2_{2,n} \Big\}$$
$$\overset{(a)}{=} \underbrace{\max_{g \in \mathbb{G}} \Big\{ \widehat{\Phi}_{\pi,h}(b_h, b_{h+1}; g) - \widehat{\Phi}_{\pi,h}(b^\star_h(b_{h+1}), b_{h+1}; g) - 2\lambda \|g\|^2_{2,n} \Big\}}_{(\star)}$$
$$- \max_{g \in \mathbb{G}} \widehat{\Phi}^\lambda_{\pi,h}(b^\star_h(b_{h+1}), b_{h+1}; g). \tag{F.12}$$

Here step (a) follows from that $\mathbb{G}$ is symmetric, $\widehat{\Phi}_{\pi,h}(b_h, h_{h+1}; -g) = -\widehat{\Phi}_{\pi,h}(b_h, h_{h+1}; g)$, and that

$$\min_{g \in \mathbb{G}} \Big\{ \widehat{\Phi}_{\pi,h}(b^\star_h(b_{h+1}), b_{h+1}; g) + \lambda \|g\|^2_{2,n} \Big\} = \min_{g \in \mathbb{G}} \Big\{ -\widehat{\Phi}_{\pi,h}(b^\star_h(b_{h+1}), b_{h+1}; -g) + \lambda \|g\|^2_{2,n} \Big\}$$
$$= \min_{g \in \mathbb{G}} \Big\{ -\widehat{\Phi}_{\pi,h}(b^\star_h(b_{h+1}), b_{h+1}; g) + \lambda \|g\|^2_{2,n} \Big\}$$
$$= -\max_{g \in \mathbb{G}} \Big\{ \widehat{\Phi}_{\pi,h}(b^\star_h(b_{h+1}), b_{h+1}; g) - \lambda \|g\|^2_{2,n} \Big\}$$
$$= -\max_{g \in \mathbb{G}} \widehat{\Phi}^\lambda_{\pi,h}(b^\star_h(b_{h+1}), b_{h+1}; g).$$

In the sequel, we upper and lower bound term $(\star)$ respectively.

**Upper bound of term $(\star)$.** By inequality (F.12), after rearranging terms, we can arrive that

$$(\star) \leq \max_{g \in \mathbb{G}} \widehat{\Phi}^\lambda_{\pi,h}(b^\star_h(b_{h+1}), b_{h+1}; g) + \max_{g \in \mathbb{G}} \widehat{\Phi}^\lambda_{\pi,h}(b_h, b_{h+1}; g)$$
$$\leq \max_{g \in \mathbb{G}} \widehat{\Phi}^\lambda_{\pi,h}(b^\star_h(b_{h+1}), b_{h+1}; g)$$
$$+ \max_{g \in \mathbb{G}} \widehat{\Phi}^\lambda_{\pi,h}(b_h, b_{h+1}; g) - \max_{g \in \mathbb{G}} \widehat{\Phi}^\lambda_{\pi,h}(\widehat{b}_h(b_{h+1}), b_{h+1}; g)$$
$$+ \max_{g \in \mathbb{G}} \widehat{\Phi}^\lambda_{\pi,h}(\widehat{b}_h(b_{h+1}), b_{h+1}; g)$$

On the one hand, by Lemma F.1, we have that with probability at least $1 - \delta/2$,

$$\max_{g \in \mathbb{G}} \widehat{\Phi}^\lambda_{\pi,h}(b^\star_h(b_{h+1}), b_{h+1}; g) \leq \xi + \epsilon^{1/2}_{\mathbb{B}} M_{\mathbb{G}}, \tag{F.13}$$

and by the definition of $\widehat{b}_h(b_{h+1})$ in (3.11), it holds simultaneously that

$$\max_{g\in\mathbb{G}}\widehat{\Phi}^\lambda_{\pi,h}(\widehat{b}_h(b_{h+1}),b_{h+1};g) \leq \max_{g\in\mathbb{G}}\widehat{\Phi}^\lambda_{\pi,h}(b^\star_h(b_{h+1}),b_{h+1};g) \leq \xi + \epsilon_\mathbb{B}^{1/2}M_\mathbb{G}. \tag{F.14}$$

On the other hand, by the choice of $\mathrm{CR}^\pi(\xi)$, it holds that

$$\max_{g\in\mathbb{G}}\widehat{\Phi}^\lambda_{\pi,h}(b_h,b_{h+1};g) - \max_{g\in\mathbb{G}}\widehat{\Phi}^\lambda_{\pi,h}(\widehat{b}_h(b_{h+1}),b_{h+1};g) \leq \xi. \tag{F.15}$$

Consequently, by combining (F.13), (F.14), and (F.15), we conclude that with probability at least $1-\delta/2$,

$$(\star) \leq 3\xi + 2\epsilon_\mathbb{B}^{1/2}M_\mathbb{G}. \tag{F.16}$$

**Lower bound of term $(\star)$.** For lower bound, we need two localized uniform concentration inequalities similar to (F.10) and (F.11) in the proof of Lemma D.2. On the one hand, by Lemma I.1, for some absolute constants $c_1, c_2 > 0$, it holds with probability at least $1-\delta/4$ that,

$$\left| \|g\|^2_{2,n} - \|g\|^2_2 \right| \leq \frac{1}{2}\|g\|^2_2 + \frac{M_\mathbb{G}^2\log(4c_1/\zeta)}{2c_2n}, \quad \forall g\in\mathbb{G}, \tag{F.17}$$

where $\zeta = \min\{\delta, 4c_1\exp(-c_2n\alpha^2_{\mathbb{G},n}/M_\mathbb{G}^2)\}$ and $\alpha_{\mathbb{G},n}$ is the critical radius of $\mathbb{G}$ defined in Assumption 4.2. On the other hand, following the same argument as in deriving (F.11), for any given $b_h, b'_h, b_{h+1}\in\mathbb{B}$, $\pi\in\Pi(\mathcal{H})$, and $h\in[H]$, in Lemma I.2 we choose $\mathcal{F}=\mathbb{G}$, $\mathcal{X}=\mathcal{A}\times\mathcal{Z}$, $\mathcal{Y}=\mathcal{I}$, and loss function

$$\ell(g(A_h,Z_h),I_h) := \varsigma^\pi_h(b_h,b_{h+1})(I_h)g(A_h,Z_h) - \varsigma^\pi_h(b'_h,b_{h+1})(I_h)g(A_h,Z_h),$$

where $\varsigma^\pi_h$ is defined in (F.1) and $I_h\in\mathcal{I}_h$ is defined in the beginning of Appendix F. It holds that $\ell$ is $L$-Lipschitz continuous in its first argument with $L = 2M_\mathbb{B}$. Now setting $f^\star = 0$ in Lemma I.2, we have that $\delta_n$ in Lemma I.2 coincides with $\alpha_{\mathbb{G},n}$ in Assumption 4.2. Then we have that for some absolute constants $c_1, c_2 > 0$, it holds with probability at least $1 - \delta/(4|\mathbb{B}|^3|\Pi(\mathcal{H})|H)$ that

$$\left| \left( \widehat{\Phi}_{\pi,h}(b_h,b_{h+1};g) - \widehat{\Phi}_{\pi,h}(b'_h,b_{h+1};g) \right) - \left( \Phi_{\pi,h}(b_h,b_{h+1};g) - \Phi_{\pi,h}(b'_h,b_{h+1};g) \right) \right|$$

$$= \left| \widehat{\mathbb{E}}_{\pi^b}[\ell(g(A_h,Z_h),I_h)] - \mathbb{E}_{\pi^b}[\ell(g(A_h,Z_h),I_h)] \right|$$

$$\leq 18L\|g\|_2\sqrt{\frac{M_\mathbb{G}^2\cdot\log(4c_1|\mathbb{B}|^3|\Pi(\mathcal{H})|H/\zeta')}{c_2n}} + \frac{18L\cdot M_\mathbb{G}^2\cdot\log\left(4c_1|\mathbb{B}|^3|\Pi(\mathcal{H})|H/\zeta'\right)}{c_2n}, \quad \forall g\in\mathbb{G}, \tag{F.18}$$

where $\zeta' = \min\{\delta, 4c_1|\mathbb{B}|^3|\Pi(\mathcal{H})|H\exp(-c_2n\alpha^2_{\mathbb{G},n}/M_\mathbb{G}^2)\}$. Applying a union bound argument over $b_h, b'_h, b_{h+1}\in\mathbb{B}$, $\pi\in\Pi(\mathcal{H})$, and $h\in[H]$, we have that (F.18) holds for any $b_h, b'_h, b_{h+1}\in\mathbb{B}$, $g\in\mathbb{G}$, $\pi\in\Pi(\mathcal{H})$, and $h\in[H]$ with probability at least $1-\delta/4$. Finally, for simplicity, we denote that

$$\iota_n := \sqrt{\frac{M_\mathbb{G}^2\cdot\log(4c_1|\mathbb{B}|^3|\Pi(\mathcal{H})|H/\zeta')}{c_2n}}, \quad \iota'_n := \sqrt{\frac{M_\mathbb{G}^2\cdot\log(4c_1/\zeta)}{2c_2n}} \tag{F.19}$$

Now we are ready to prove the lower bound on term $(\star)$. For simplicity, given fixed $b_h, b_{h+1}\in\mathbb{B}$, we denote

$$g^\pi_h := \frac{1}{2\lambda}\ell^\pi_h(b_h,b_{h+1})\in\mathbb{G},$$

where $\ell^\pi_h$ is defined in (F.1) and $g^\pi_h\in\mathbb{G}$ due to Assumption 4.3. Now consider that

$$(\star) = \max_{g\in\mathbb{G}}\left\{ \widehat{\Phi}_{\pi,h}(b_h,b_{h+1};g) - \widehat{\Phi}_{\pi,h}(b^\star_h(b_{h+1}),b_{h+1};g) - 2\lambda\|g\|^2_{2,n} \right\}$$

$$\geq \widehat{\Phi}_{\pi,h}(b_h,b_{h+1};g^\pi_h/2) - \widehat{\Phi}_{\pi,h}(b^\star_h(b_{h+1}),b_{h+1};g^\pi_h/2) - \frac{\lambda}{2}\|g^\pi_h\|^2_{2,n},$$

where the inequality follows from the fact that $\mathbb{G}$ is star-shaped and consequently $g_h^\pi/2 \in \mathbb{G}$. Then by applying concentration inequality (F.17) and (F.18), we have that

$$
\begin{aligned}
(\star) &\geq \Phi_{\pi,h}(b_h, b_{h+1}; g_h^\pi/2) - \Phi_{\pi,h}(b_h^\star(b_{h+1}), b_{h+1}; g_h^\pi/2) - 18L\iota_n \|g_h^\pi\|_2 - 18L\iota_n^2 \\
&\quad - \frac{\lambda}{2}\left(\frac{3}{2}\|g_h^\pi\|_2^2 + \iota_n'^2\right) \\
&\geq \lambda\|g_h^\pi\|_2^2 - 18L\iota_n\|g_h^\pi\|_2 - \epsilon_{\mathbb{B}}^{1/2}M_{\mathbb{G}} - 18L\iota_n^2 - \frac{\lambda}{2}\left(\frac{3}{2}\|g_h^\pi\|_2^2 + \iota_n'^2\right) \\
&= \frac{\lambda}{4}\|g_h^\pi\|_2^2 - 18L\iota_n\|g_h^\pi\|_2 - 18L\iota_n^2 - \frac{\lambda}{2}\iota_n'^2 - \epsilon_{\mathbb{B}}^{1/2}M_{\mathbb{G}}, \quad\quad\quad\quad (\text{F}.20)
\end{aligned}
$$

where the second inequality follows from that $\Phi_{\pi,h}(b^\star(b_{h+1}), b_{h+1}; g_h^\pi/2) \leq \epsilon_{\mathbb{B}}^{1/2}M_{\mathbb{G}}$ (we prove this inequality by (F.25) in the proof of Lemma F.1) and the fact that

$$
\Phi_{\pi,h}(b_h, b_{h+1}; g_h^\pi/2) = \frac{1}{4\lambda}\mathbb{E}_{\pi^b}[\ell_h^\pi(b_h, b_{h+1})(A_h, Z_h)^2] = \lambda\|g_h^\pi\|_2^2.
$$

**Combining upper bound and lower bound of term $(\star)$.** Now we are ready to combine the upper bound and lower bound of $(\star)$ to derive the bound on $\mathcal{L}_h^\pi(b_h, b_{h+1})$. By combining upper bound (F.16) and lower bound (F.20), we have that with probability at least $1 - \delta$, for any $b_h, b_{h+1} \in \mathbb{B}$, $\pi \in \Pi(\mathcal{H})$, and $h \in [H]$,

$$
\frac{\lambda}{4}\|g_h^\pi\|_2^2 - 18L\iota_n\|g_h^\pi\|_2 - 18L\iota_n^2 - \frac{\lambda}{2}\iota_n'^2 - \epsilon_{\mathbb{B}}^{1/2}M_{\mathbb{G}} \leq 3\xi + 2\epsilon_{\mathbb{B}}^{1/2}M_{\mathbb{G}}, \quad\quad\quad (\text{F}.21)
$$

This gives a quadratic inequality on $\|g_h^\pi\|_2$, i.e.,

$$
\lambda\|g_h^\pi\|_2^2 - \underbrace{72L\iota_n}_{(\text{A})}\|g_h^\pi\|_2 - \underbrace{4\left(18L\iota_n^2 + \frac{\lambda}{2}\iota_n'^2 + 3\xi + 3\epsilon_{\mathbb{B}}^{1/2}M_{\mathbb{G}}\right)}_{(\text{B})} \leq 0.
$$

By solving this quadratic equation, we have that

$$
\|g_h^\pi\|_2 \leq \frac{1}{2\lambda}\text{A} + \frac{1}{2\lambda}\sqrt{\text{A}^2 + 4\text{B}} \leq \frac{\text{A}}{\lambda} + \frac{\sqrt{\text{B}}}{\lambda}.
$$

Applying the definition of A and B, we conclude that, with probability at least $1 - \delta$,

$$
\begin{aligned}
\|g_h^\pi\|_2 &\leq \frac{72}{\lambda}L\iota_n + \frac{2}{\lambda}\left(18L\iota_n^2 + \frac{\lambda}{2}\iota_n'^2 + 3\xi + 3\epsilon_{\mathbb{B}}^{1/2}M_{\mathbb{G}}\right)^{1/2} \\
&\leq \frac{72}{\lambda}L\iota_n + \frac{6\sqrt{2}}{\lambda}L^{1/2}\iota_n + \frac{\sqrt{2}}{\sqrt{\lambda}}\iota_n' + \frac{2\sqrt{3}}{\lambda}\xi^{1/2} + \frac{2\sqrt{3}}{\lambda}\epsilon_{\mathbb{B}}^{1/4}M_{\mathbb{G}}^{1/2}
\end{aligned}
$$

Therefore, we can bound the RMSE loss $\mathcal{L}_h^\pi(b_h, b_{h+1})$ by

$$
\sqrt{\mathcal{L}_h^\pi(b_h, b_{h+1})} = 2\lambda\|g_h^\pi\|_2 \leq (144L + 12\sqrt{2}L^{1/2})\iota_n + 2\sqrt{2\lambda}\iota_n' + 4\sqrt{3}\xi^{1/2} + 4\sqrt{3}\epsilon_{\mathbb{B}}^{1/4}M_{\mathbb{G}}^{1/2}. \quad (\text{F}.22)
$$

Plugging in the definition of $\iota_n, \iota_n'$ in (F.19), $\xi$ in Lemma D.3, and that $L = 2M_{\mathbb{B}}$, we have that

$$
\begin{aligned}
&\sqrt{\mathcal{L}_h^\pi(b_h, b_{h+1})} \\
&\quad \leq (144L + 12\sqrt{2L}) \cdot \sqrt{\frac{M_{\mathbb{G}}^2 \cdot \log(4c_1|\mathbb{B}|^3|\Pi(\mathcal{H})|H/\zeta')}{c_2 n}} + 2\sqrt{2\lambda} \cdot \sqrt{\frac{M_{\mathbb{G}}^2 \cdot \log(4c_1/\zeta)}{2c_2 n}} \\
&\quad\quad + 4\sqrt{3} \cdot \sqrt{\frac{C_1(\lambda + 1/\lambda) \cdot M_{\mathbb{B}}^2 M_{\mathbb{G}}^2 \cdot \log(|\mathbb{B}||\Pi(\mathcal{H})|H/\zeta')}{n}} + 4\sqrt{3} \cdot \epsilon_{\mathbb{B}}^{1/4}M_{\mathbb{G}}^{1/2} \\
&\quad \leq \widetilde{C}_1 M_{\mathbb{B}} M_{\mathbb{G}} \sqrt{\frac{(\lambda + 1/\lambda) \cdot \log(|\mathbb{B}||\Pi(\mathcal{H})|H/\zeta)}{n}} + \widetilde{C}_1 \epsilon_{\mathbb{B}}^{1/4}M_{\mathbb{G}}^{1/2}.
\end{aligned}
$$

for some problem-independent constant $\widetilde{C}_1 > 0$ and $\zeta = \min\{\delta, 4c_1\exp(-c_2 n\alpha_{\mathbb{G},n}^2/M_{\mathbb{G}}^2)\}$. Here in the second inequality we have used the fact that $\zeta < \zeta'$. This finishes the proof of Lemma D.3. $\quad\square$

### F.4 PROOF OF LEMMA F.1

*Proof of Lemma F.1.* Following the proof of Lemma D.2, we first relate $\widehat{\Phi}^\lambda_{\pi,h}(b_h, b_{h+1}; g) = \widehat{\Phi}_{\pi,h}(b_h, b_{h+1}; g) - \lambda\|g\|^2_{2,n}$ and its population version $\Phi^\lambda_{\pi,h}(b_h, b_{h+1}; g)$ via two localized uniform concentration inequalities. On the one hand, to relate $\|g\|^2_2$ and $\|g\|^2_{2,n}$, by Lemma I.1 (Theorem 14.1 of Wainwright (2019)), for some absolute constants $c_1, c_2 > 0$, it holds with probability at least $1 - \delta/4$ that

$$\left|\|g\|^2_{2,n} - \|g\|^2_2\right| \leq \frac{1}{2}\|g\|^2_2 + \frac{M^2_{\mathbb{G}} \cdot \log(4c_1/\zeta)}{2c_2 n}, \quad \forall g \in \mathbb{G}, \tag{F.23}$$

where $\zeta = \min\{\delta, 4c_1 \exp(-c_2 n\alpha^2_{\mathbb{G},n}/M^2_{\mathbb{G}})\}$ and $\alpha_{\mathbb{G},n}$ is the critical radius of function class $\mathbb{G}$ defined in Assumption 4.2. On the other hand, to relate $\widehat{\Phi}_{\pi,h}(b_h, b_{h+1}; g)$ and $\Phi_{\pi,h}(b_h, b_{h+1}; g)$, we invoke Lemma I.2 (Lemma 11 of (Foster and Syrgkanis, 2019)). Specifically, for any given $b_h, b_{h+1} \in \mathbb{B}$, $\pi \in \Pi(\mathcal{H})$, and step $h$, in Lemma I.2 we choose $\mathcal{F} = \mathbb{G}$, $\mathcal{X} = \mathcal{A} \times \mathcal{Z}$, $\mathcal{Y} = \mathcal{I}_h$, and loss function $\ell(g(A_h, Z_h), I_h) := \varsigma^\pi_h(b_h, b_{h+1})(I_h)g(A_h, Z_h)$ where $\ell^\pi_h$ is defined in (F.1) and $I_h \in \mathcal{I}_h$ is defined in the beginning of Appendix F. We can see that $\ell$ is $L$-Lipschitz continuous in the first argument since for any $g, g' \in \mathbb{G}$, $(A_h, Z_h) \in \mathcal{A} \times \mathcal{Z}$, it holds that

$$\left|\ell(g(A_h, Z_h), I_h) - \ell(g'(A_h, Z_h), I_h)\right| = |\varsigma^\pi_h(b_h, b_{h+1})(I_h)| \cdot |g(A_h, Z_h) - g'(A_h, Z_h)|$$
$$\leq 2M_{\mathbb{B}} \cdot |g(A_h, Z_h) - g'(A_h, Z_h)|,$$

which indicates that $L = 2M_{\mathbb{B}}$. Now setting $f^\star = 0$ in Lemma I.2, we have that $\delta_n$ in Lemma I.2 coincides with $\alpha_{\mathbb{G},n}$ in Assumption 4.2. Then we can conclude that for some absolute constants $c_1, c_2 > 0$, it holds with probability at least $1 - \delta/(4|\mathbb{B}|^2|\Pi(\mathcal{H})|H)$ that, for all $g \in \mathbb{G}$,

$$\left|\widehat{\Phi}_{\pi,h}(b_h, b_{h+1}; g) - \Phi_{\pi,h}(b_h, b_{h+1}; g)\right|$$
$$= \left|\widehat{\mathbb{E}}_{\pi^b}[\ell(g(A_h, Z_h), A_h, Z_h)] - \mathbb{E}_{\pi^b}[\ell(g(A_h, Z_h), A_h, Z_h)]\right|$$
$$\leq 18L\|g\|_2\sqrt{\frac{M^2_{\mathbb{G}} \cdot \log\left(4c_1|\mathbb{B}|^2|\Pi(\mathcal{H})|H/\zeta\right)}{c_2 n}} + \frac{18L \cdot M^2_{\mathbb{G}} \cdot \log\left(4c_1|\mathbb{B}|^2|\Pi(\mathcal{H})|H/\zeta\right)}{c_2 n},$$
$$\tag{F.24}$$

where $\zeta = \min\{\delta, 4c_1|\mathbb{B}|^2|\Pi(\mathcal{H})|H \exp(-c_2 n\alpha^2_{\mathbb{G},n}/M^2_{\mathbb{G}})\}$. Applying a union bound argument over $b_h, b_{h+1} \in \mathbb{B}$, $\pi \in \Pi(\mathcal{H})$, and $h \in [H]$, we then have that (F.11) holds for any $b_h, b_{h+1} \in \mathbb{B}$, $g \in \mathbb{G}$, $\pi \in \Pi(\mathcal{H})$, and $h \in [H]$ with probability at least $1 - \delta/4$. Now using these two concentration inequalities (F.23) and (F.24), we can further deduce that, for some absolute constants $c_1, c_2 > 0$, with probability at least $1 - \delta/2$,

$$\max_{g \in \mathbb{G}} \widehat{\Phi}^\lambda_{\pi,h}(b^\star_h(b_{h+1}), b^\pi_{h+1}; g)$$

$$= \max_{g \in \mathbb{G}} \left\{\widehat{\Phi}_{\pi,h}(b^\star_h(b_{h+1}), b_{h+1}; g) - \lambda\|g\|^2_{2,n}\right\}$$

$$\leq \max_{g \in \mathbb{G}} \left\{\Phi_{\pi,h}(b^\star_h(b_{h+1}), b_{h+1}; g) - \lambda\|g\|^2_2 + \frac{\lambda}{2}\|g\|^2_2 + \frac{\lambda M^2_{\mathbb{G}}\log(4c_1/\zeta)}{2c_2 n}, \right.$$

$$\left. + 18L\|g\|_2\sqrt{\frac{M^2_{\mathbb{G}} \cdot \log(4c_1|\mathbb{B}|^2|\Pi(\mathcal{H})|H/\zeta')}{c_2 n}} + \frac{18L \cdot M^2_{\mathbb{G}} \cdot \log(4c_1|\mathbb{B}|^2|\Pi(\mathcal{H})|H/\zeta')}{c_2 n}\right\}$$

$$\leq \max_{g \in \mathbb{G}} \Phi_{\pi,h}(b^\star_h(b_{h+1}), b_{h+1}; g) + \max_{g \in \mathbb{G}} \left\{-\frac{\lambda}{2}\|g\|^2_2 + 18L\|g\|_2\sqrt{\frac{M^2_{\mathbb{G}} \cdot \log(4c_1|\mathbb{B}|^2|\Pi(\mathcal{H})|H/\zeta')}{c_2 n}}\right\}$$

$$+ \frac{\lambda M^2_{\mathbb{G}}\log(4c_1/\zeta)}{2c_2 n} + \frac{18L \cdot M^2_{\mathbb{G}} \cdot \log(4c_1|\mathbb{B}|^2|\Pi(\mathcal{H})|H/\zeta')}{c_2 n}$$

$$\leq \epsilon^{1/2}_{\mathbb{B}}M_{\mathbb{G}} + \frac{728L^2 \cdot M^2_{\mathbb{G}} \cdot \log(4c_1|\mathbb{B}|^2|\Pi(\mathcal{H})|H/\zeta')}{\lambda n} + \frac{\lambda M^2_{\mathbb{G}} \cdot \log(4c_1/\zeta)}{2c_2 n}$$

$$+ \frac{18L \cdot sM^2_{\mathbb{G}} \cdot \log(4c_1|\mathbb{B}|^2|\Pi(\mathcal{H})|H/\zeta')}{c_2 n},$$

where $\zeta$ is given as $\zeta = \min\{\delta, 4c_1 \exp(-c_2 n \alpha_{\mathbb{G},n}^2 / M_{\mathbb{G}}^2)\}$ and $\zeta'$ is given as $\zeta' = \min\{\delta, 4c_1 |\mathbb{B}|^2 |\Pi(\mathcal{H})| H \exp(-c_2 n \alpha_{\mathbb{G},n}^2 / M_{\mathbb{G}}^2)\}$ for any policy $\pi \in \Pi(\mathcal{H})$ and step $h \in [H]$. Here the last inequality holds from the fact that

$$\max_{g \in \mathcal{G}} \Phi_{\pi,h}(b_h^\star(b_{h+1}), b_{h+1}; g) \le \epsilon_{\mathbb{B}}^{1/2} M_{\mathbb{G}}, \tag{F.25}$$

and that $\sup_{\|g\|_2} \{a\|g\|_2 - b\|g\|_2^2\} \le a^2/4b$. Note that inequality (F.25) holds according to Assumption 4.3 and 4.3. In fact, by Assumption 4.3, we can first obtain by quadratic optimization that for $\lambda > 0$,

$$\max_{g \in \mathbb{G}} \Phi_{\pi,h}^\lambda(b_h, b_{h+1}) = \frac{1}{4\lambda} \mathcal{L}_h^\pi(b_h, b_{h+1}),$$

for any functions $b_h, b_{h+1} \in \mathbb{B}$. Thus we can equivalently express $b_h^\star(b_{h+1})$ as

$$b_h^\star(b_{h+1}) = \arg\min_{b \in \mathbb{B}} \frac{1}{4\lambda} \mathcal{L}_h^\pi(b, b_{h+1}) = \arg\min_{b \in \mathbb{B}} \mathcal{L}_h^\pi(b, b_{h+1}).$$

This further indicates the following bound on $\max_{g \in \mathbb{G}} \Phi_{\pi,h}(b_h^\star(b_{h+1}), b_{h+1}; g)$ that

$$\max_{g \in \mathbb{G}} \Phi_{\pi,h}(b_h^\star(b_{h+1}), b_{h+1}; g) \le \max_{g \in \mathbb{G}} \sqrt{\mathcal{L}_h(b_h^\star(b_{h+1}), b_{h+1}) \cdot \mathbb{E}_{\pi^b}[g(A_h, Z_h)^2]} \le \epsilon_{\mathbb{B}}^{1/2} M_{\mathbb{G}},$$

by Cauchy-Schwarz inequality and Assumption 4.3. Now according to the choice of $\xi$ in Lemma D.2, using the fact that $\zeta < \zeta'$ and $L = 2M_{\mathbb{B}}$, we can conclude that, with probability at least $1 - \delta/2$,

$$\max_{g \in \mathbb{G}} \widehat{\Phi}_{\pi,h}^\lambda(b_h^\star(b_{h+1}), b_{h+1}^\pi; g)$$

$$\le \frac{728 L^2 \cdot M_{\mathbb{G}}^2 \cdot \log(4c_1 |\mathbb{B}|^2 |\Pi(\mathcal{H})| H / \zeta')}{\lambda n} + \frac{\lambda M_{\mathbb{G}}^2 \cdot \log(4c_1/\zeta)}{2c_2 n}$$

$$+ \frac{18 L \cdot M_{\mathbb{G}}^2 \cdot \log(4c_1 |\mathbb{B}|^2 |\Pi(\mathcal{H})| H / \zeta')}{c_2 n} + \epsilon_{\mathbb{B}}^{1/2} M_{\mathbb{G}}$$

$$\lesssim \mathcal{O}\left( \frac{(\lambda + 1/\lambda) \cdot M_{\mathbb{B}}^2 M_{\mathbb{G}}^2 \cdot \log(|\mathbb{B}| |\Pi(\mathcal{H})| H / \zeta)}{n} \right) + \epsilon_{\mathbb{B}}^{1/2} M_{\mathbb{G}} \lesssim \xi + \epsilon_{\mathbb{B}}^{1/2} M_{\mathbb{G}}.$$

Therefore, we conclude the proof of Lemma F.1. $\qquad\square$

## G  PROOF OF THEOREM 4.4

*Proof of Theorem 4.4.* By the definition of $F(\mathbf{b})$ and $\widehat{F}(\mathbf{b})$ in (D.1) and the fact that $J(\pi) = F(\mathbf{b}^\pi)$ according to Theorem 3.3, we first have that

$$J(\pi^\star) - J(\widehat{\pi})$$
$$= F(\mathbf{b}^{\pi^\star}) - F(\mathbf{b}^{\widehat{\pi}})$$
$$= \underbrace{\left(F(\mathbf{b}^{\pi^\star}) - \widehat{F}(\mathbf{b}^{\pi^\star})\right)}_{\text{(i)}} + \underbrace{\left(F(\mathbf{b}^{\pi^\star}) - \widehat{F}(\mathbf{b}^{\widehat{\pi}})\right)}_{\text{(ii)}} + \underbrace{\left(\widehat{F}(\mathbf{b}^{\widehat{\pi}}) - F(\mathbf{b}^{\widehat{\pi}})\right)}_{\text{(iii)}}.$$

We can bound term (i) and term (iii) via uniform concentration inequalities, which we present latter. For term (ii), via Lemma D.2, with probability at least $1 - \delta$, $\mathbf{b}^{\pi^\star} \in \mathrm{CR}^{\pi^\star}(\xi)$ and $\mathbf{b}^{\widehat{\pi}} \in \mathrm{CR}^{\widehat{\pi}}(\xi)$, which indicates that

$$\text{(ii)} = \widehat{F}(\mathbf{b}^{\pi^\star}) - \widehat{F}(\mathbf{b}^{\widehat{\pi}}) \le \max_{\mathbf{b} \in \mathrm{CR}^{\pi^\star}(\xi)} \widehat{F}(\mathbf{b}) - \min_{\mathbf{b} \in \mathrm{CR}^{\widehat{\pi}}(\xi)} \widehat{F}(\mathbf{b}). \tag{G.1}$$

From (G.1), we can further bound term (ii) as

$$\text{(ii)} \le \max_{\mathbf{b} \in \mathrm{CR}^{\pi^\star}(\xi)} \widehat{F}(\mathbf{b}) - \max_{\pi \in \Pi(\mathcal{H})} \min_{\mathbf{b} \in \mathrm{CR}^{\pi}(\xi)} \widehat{F}(\mathbf{b})$$

$$\le \max_{\mathbf{b} \in \mathrm{CR}^{\pi^\star}(\xi)} \widehat{F}(\mathbf{b}) - \min_{\mathbf{b} \in \mathrm{CR}^{\pi^\star}(\xi)} \widehat{F}(\mathbf{b})$$

$$= \max_{\mathbf{b} \in \mathrm{CR}^{\pi^\star}(\xi)} \widehat{F}(\mathbf{b}) - \widehat{F}(\mathbf{b}^{\pi^\star}) + \widehat{F}(\mathbf{b}^{\pi^\star}) - \min_{\mathbf{b} \in \mathrm{CR}^{\pi^\star}(\xi)} \widehat{F}(\mathbf{b})$$

$$\le 2 \max_{\mathbf{b} \in \mathrm{CR}^{\pi^\star}(\xi)} \left| \widehat{F}(\mathbf{b}) - \widehat{F}(\mathbf{b}^{\pi^\star}) \right|. \tag{G.2}$$

Here the first inequality holds because $\max_{\pi\in\Pi(\mathcal{H})}\min_{\mathbf{b}\in\mathrm{CR}^\pi(\xi)}\widehat{F}(\mathbf{b})=\min_{\mathbf{b}\in\mathrm{CR}^{\widehat{\pi}}(\xi)}\widehat{F}(\mathbf{b})$ by the definition of $\widehat{\pi}$ from (3.14). The second inequality holds because by definition $\pi^\star$ is the optimal policy in $\Pi(\mathcal{H})$. The third inequality is trivial. Now to further bound (G.2) by the RMSE loss defined in (3.6), we consider

$$2\max_{\mathbf{b}\in\mathrm{CR}^{\pi^\star}(\xi)}\left|\widehat{F}(\mathbf{b})-\widehat{F}(\mathbf{b}^{\pi^\star})\right|$$

$$\leq 2\underbrace{\max_{\mathbf{b}\in\mathrm{CR}^{\pi^\star}(\xi)}\left|\widehat{F}(\mathbf{b})-F(\mathbf{b})\right|}_{\text{(iv)}}+2\underbrace{\max_{\mathbf{b}\in\mathrm{CR}^{\pi^\star}(\xi)}\left|F(\mathbf{b})-F(\mathbf{b}^{\pi^\star})\right|}_{\text{(v)}}+2\underbrace{\left|F(\mathbf{b}^{\pi^\star})-\widehat{F}(\mathbf{b}^{\pi^\star})\right|}_{\text{(vi)}},$$

where we can bound term (iv) and term (vi) via uniform concentration inequalities, which we present latter. For term (v), we invoke Lemma D.1 and obtain that

$$\text{(v)}\leq 2\max_{\mathbf{b}\in\mathrm{CR}^{\pi^\star}(\xi)}\sum_{h=1}^H\gamma^{h-1}\sqrt{C^{\pi^\star}}\cdot\sqrt{\mathcal{L}_h^{\pi^\star}(b_h,b_{h+1})}\leq 2\sqrt{C^{\pi^\star}}\sum_{h=1}^H\gamma^{h-1}\max_{\mathbf{b}\in\mathrm{CR}^{\pi^\star}(\xi)}\sqrt{\mathcal{L}_h^{\pi^\star}(b_h,b_{h+1})}.$$

Now invoking Lemma D.3, with probability at least $1-\delta$, $\max_{\mathbf{b}\in\mathrm{CR}^{\pi^\star}(\xi)}\sqrt{\mathcal{L}_h^{\pi^\star}(b_h,b_{h+1})}$ is bounded by

$$\max_{\mathbf{b}\in\mathrm{CR}^{\pi^\star}(\xi)}\sqrt{\mathcal{L}_h^\pi(b_h,b_{h+1})}\leq\widetilde{C}_1 M_{\mathbb{B}}M_{\mathbb{G}}\sqrt{\frac{(\lambda+1/\lambda)\log(|\mathbb{B}||\Pi(\mathcal{H})|H/\zeta)}{n}}+\widetilde{C}_1\epsilon_{\mathbb{B}}^{1/4}M_{\mathbb{G}}^{1/2},\quad\text{(G.3)}$$

for each step $h\in[H]$, where $\zeta=\min\{\delta,c_1\exp(-c_2 n\alpha_{\mathbb{G},n}^2)\}$. In the sequel, we turn to deal with term (i), (iii), (iv), and (vi), respectively. To this end, it suffices to apply uniform concentration inequalities to bound $F(\mathbf{b})$ and $\widehat{F}(\mathbf{b})$ uniformly over $\mathbf{b}\in\mathbb{B}^{\otimes H}$. By Hoeffding inequality, we have that, with probability at least $1-\delta$,

$$\left|J(\pi,\mathbf{b})-\widehat{J}(\pi,\mathbf{b})\right|\leq\sqrt{\frac{2M_{\mathbb{B}}^2\log(|\mathbb{B}|/\delta)}{n}},\quad\forall\pi\in\Pi(\mathcal{H}),\quad\forall\mathbf{b}\in\mathbb{B}^{\otimes H}.\qquad\text{(G.4)}$$

Consequently, all of (i), (iii), (iv), and (vi) are bounded by the right hand side of (G.4). Finally, by combining (G.3) and (G.4), with probability at least $1-3\delta$, it holds that

$$J(\pi^\star)-J(\widehat{\pi})\leq\text{(i)}+\text{(iii)}+\text{(iv)}+\text{(vi)}+\text{(v)}$$

$$\leq 2\sqrt{C^{\pi^\star}}\sum_{h=1}^H\gamma^{h-1}\left(\widetilde{C}_1 M_{\mathbb{B}}M_{\mathbb{G}}\sqrt{\frac{(\lambda+1/\lambda)\log(|\mathbb{B}||\Pi(\mathcal{H})|H/\zeta)}{n}}+\widetilde{C}_1\epsilon_{\mathbb{B}}^{1/4}M_{\mathbb{G}}^{1/2}\right)$$

$$+4\sqrt{\frac{2M_{\mathbb{B}}^2\log(|\mathbb{B}|/\delta)}{n}}$$

$$\leq C_1'\sqrt{C^{\pi^\star}}\left(\lambda+1/\lambda\right)^{1/2}HM_{\mathbb{B}}M_{\mathbb{G}}\sqrt{\frac{\log(|\mathbb{B}||\Pi(\mathcal{H})|H/\zeta)}{n}}+C_1'\sqrt{C^{\pi^\star}}H\epsilon_{\mathbb{B}}^{1/4}M_{\mathbb{G}}^{1/2},$$

for some problem-independent constant $C_1'>0$. We finish the proof of Theorem 4.4 by taking $\lambda=1$. $\qquad\square$

# H  DETAILS FOR LINEAR FUNCTION APPROXIMATION

## H.1  MAIN RESULT FOR LINEAR FUNCTION APPROXIMATION

In this subsection, we extend Theorem 4.4 to primal function class $\mathbb{B}$, dual function class $\mathbb{G}$, and policy class $\Pi(\mathcal{H})$ with linear structures. The linear structure assumption is commonly considered in the RL literature (Jin et al., 2021; Xie et al., 2021; Zanette et al., 2021; Duan et al., 2021; Min et al., 2022a;b; Fei and Xu, 2022; Huang et al., 2023), to mention a few. And it can be viewed as an extension of linear bandits (Auer, 2002; Dani et al., 2008; Li et al., 2010; Abbasi-Yadkori et al., 2011; He et al., 2022) to multiple-horizon setting. Note that the exact detail of the linear structure assumption might change across different works. In our case, we consider linear function classes $\mathbb{B}_{\mathrm{lin}}$, $\mathbb{G}_{\mathrm{lin}}$ and $\Pi_{\mathrm{lin}}$, which is characterized by the following definition.

**Definition H.1** (Linear function approximation). *Let $\phi : \mathcal{A} \times \mathcal{W} \to \mathbb{R}^d$ be a feature mapping for some integer $d \in \mathbb{N}$. We let the primal function class be $\mathbb{B} = \mathbb{B}_{\mathrm{lin}}$ where*

$$\mathbb{B}_{\mathrm{lin}} := \left\{ b \, \middle| \, b(\cdot, \cdot) = \langle \phi(\cdot, \cdot), \theta \rangle, \theta \in \mathbb{R}^d, \|\theta\|_2 \le L_b, \ \sup_{w \in \mathcal{W}} |\sum_{a \in \mathcal{A}} b(a, w)| \le M_{\mathbb{B}} \right\}.$$

*Let $\psi = \{\psi_h : \mathcal{A} \times \mathcal{O} \times \mathcal{H}_{h-1} \to \mathbb{R}^d\}_{h=1}^H$ be $H$ feature mappings. We let the policy function class be $\Pi(\mathcal{H}) = \Pi_{\mathrm{lin}}$ where $\Pi_{\mathrm{lin}} = \{\Pi_{\mathrm{lin},h}\}_{h=1}^H$ and each $\Pi_{\mathrm{lin},h}$ is defined as*

$$\Pi_{\mathrm{lin},h} := \left\{ \pi_h \, \middle| \, \pi_h(a|o, \tau) = \frac{\exp(\langle \psi_h(a, o, \tau), \beta \rangle)}{\sum_{a' \in \mathcal{A}} \exp(\langle \psi_h(a', o, \tau), \beta \rangle)}, \ \beta \in \mathbb{R}^d, \ \|\beta\|_2 \le L_\pi \right\}.$$

*Finally, let $\nu : \mathcal{A} \times \mathcal{Z} \to \mathbb{R}^d$ be another feature mapping. We let the dual function class be $\mathbb{G} = \mathbb{G}_{\mathrm{lin}}$ where*

$$\mathbb{G}_{\mathrm{lin}} := \left\{ g \, \middle| \, g(\cdot, \cdot) = \langle \nu(\cdot, \cdot), \omega \rangle, \omega \in \mathbb{R}^d, \|\omega\|_2 \le L_g \right\}.$$

*Assume without loss of generality that these feature mappings are normalized, i.e., $\|\phi\|_2, \|\psi\|_2, \|\nu\|_2 \le 1$.*

We note that Definition H.1 is consistent with Assumption 4.2. One can see that $\mathbb{B}_{\mathrm{lin}}$ and $\mathbb{G}_{\mathrm{lin}}$ is uniformly bounded, $\mathbb{G}_{\mathrm{lin}}$ is symmetric and star-shaped. And for other more detailed theoretical properties of $\mathbb{B}_{\mathrm{lin}}$, $\mathbb{G}_{\mathrm{lin}}$, and $\Pi_{\mathrm{lin}}$, we refer the readers to Appendix H.2 for corresponding results.

Under linear function approximation, we can extend Theorem 4.4 to the following corollary, which characterizes the suboptimality (2.2) of $\widehat{\pi}$ found by P3O when using $\mathbb{B}_{\mathrm{lin}}$, $\mathbb{G}_{\mathrm{lin}}$, and $\Pi_{\mathrm{lin}}$ as function classes.

**Corollary H.2** (Suboptimality analysis: linear function approximation). *With linear function approximation (Definition H.1), under Assumption 3.1, 3.2, 4.1, and 4.3, by setting the regularization parameter $\lambda$ and the confidence parameter $\xi$ as $\lambda = 1$ and*

$$\xi = C_2 M_{\mathbb{B}}^2 \cdot M_{\mathbb{G}}^2 \cdot dH \cdot \log(1 + L_b L_\pi H n / \delta) / n,$$

*then with probability at least $1 - \delta$, it holds that*

$$\mathrm{SubOpt}(\widehat{\pi}) \le C_2' \sqrt{C^{\pi^\star}} H M_{\mathbb{B}} L_g \sqrt{dH \log(1 + L_b L_\pi H n / \delta) / n} + C_2' \sqrt{C^{\pi^\star} L_g} H \epsilon_{\mathbb{B}}^{1/4}.$$

*Here $C_2$ and $C_2'$ are problem-independent universal constants.*

*Proof of Corollary H.2.* See Appendix H.3 for a detailed proof. $\qquad\square$

The guarantee of Corollary H.2 is structurally similar to that of Theorem 4.4, except that we can explicitly compute the complexity of the linear function classes and policy class. When $\epsilon_{\mathbb{B}} = 0$, P3O algorithm enjoys a $\widetilde{\mathcal{O}}(\sqrt{C^{\pi^\star} H^3 d / n})$ suboptimality under the linear function approximation. Compared to Theorem 4.4, Corollary H.2 does not explicitly assume Assumption 4.2 since it is implicitly satisfied by Definition H.1.

## H.2 Auxiliary Results for Linear Function Approximation

Here we present results that bound the complexity of certain functions classes in the case of linear function approximation (Definition H.1).

Recall the definition of the bridge function class $\mathbb{B}^{\otimes H}$ where $\mathbb{B} = \mathbb{B}_{\mathrm{lin}}$ is defined as

$$\mathbb{B}_{\mathrm{lin}} := \left\{ b \, \middle| \, b(\cdot, \cdot) = \langle \boldsymbol{\phi}(\cdot, \cdot), \theta \rangle, \theta \in \mathbb{R}^d, \|\theta\|_2 \le L_b, \ \sup_{w \in \mathcal{W}} |\sum_{a \in \mathcal{A}} b(a, w)| \le M_{\mathbb{B}} \right\}.$$

Denote by $\mathcal{N}_\epsilon^\infty(\mathbb{B})$ the $\epsilon$-covering number of $\mathbb{B}$ with respect to the $\ell_\infty$ norm. That is, there exists a collection of functions $\{b_i\}_{i=1}^N$ with $N \le \mathcal{N}_\epsilon^\infty(\mathbb{B})$ such that for any $b \in \mathbb{B}$, we can find some $b' \in \{b_i\}_{i=1}^N$ satisfying

$$\|b - b'\|_\infty := \sup_{a \in \mathcal{A}, w \in \mathcal{W}} |b(a, w) - b'(a, w)| \le \epsilon.$$

Recall the policy function class $\Pi(\mathcal{H}) = \Pi_{\mathrm{lin}}^{\otimes H}$ where $\Pi_{\mathrm{lin}}$ is defined as

$$\Pi_{\mathrm{lin}} := \left\{ \pi \ \middle| \ \pi(a|o,\tau) = \frac{e^{\langle \psi(a,o,\tau),\beta \rangle}}{\sum_{a' \in \mathcal{A}} e^{\langle \psi(a',o,\tau),\beta \rangle}}, \ \beta \in \mathbb{R}^d, \ \|\beta\|_2 \le L_\pi \right\}.$$

Denote by $\mathcal{N}_\epsilon^{\infty,1}(\Pi_{\mathrm{lin}})$ the $\epsilon$-covering number of $\Pi_{\mathrm{lin}}$ with respect to the $\ell_{\infty,1}$ norm, i.e.,

$$\|\pi - \pi'\|_{\infty,1} := \sup_{o \in \mathcal{O}, \tau \in \mathcal{H}} \sum_{a \in \mathcal{A}} |\pi(a|o,\tau) - \pi'(a|o,\tau)|.$$

The upper bounds for these covering numbers are given by the following lemma.

**Lemma H.3** (Lemma 6 in Zanette et al. 2021). *For any $\epsilon \in (0,1)$, we have*

$$\log \mathcal{N}_\epsilon^{\infty}(\mathbb{B}) \le d \log \left( 1 + \frac{2L_b}{\epsilon} \right),$$

$$\log \mathcal{N}_\epsilon^{\infty,1}(\Pi_{\mathrm{lin}}) \le d \log \left( 1 + \frac{16L_\pi}{\epsilon} \right).$$

**The $\epsilon$-nets for the product function classes**   In the rest of Appendix H, due to the proof, we need to consider $\epsilon$-nets defined for the product function classes $\mathbb{B}^{\otimes H}$ and $\Pi(\mathcal{H}) = \Pi_{\mathrm{lin}}^{\otimes H}$. Specifically, for $\mathbb{B}^{\otimes H}$, we consider an $\epsilon$-net of $\mathbb{B}^{\otimes H}$ defined in the following way: for any $\mathbf{b} = \{b_h\}_{h=1}^H \in \mathbb{B}^{\otimes H}$, there exists an $\mathbf{b}' = \{b_h'\}_{h=1}^H$ in the $\epsilon$-net, such that

$$\|b_h - b_h'\|_\infty \le \epsilon.$$

By Lemma H.3, the cardinality of this $\epsilon$-net is upper bounded by

$$\log \mathcal{N}_\epsilon^{\infty}(\mathbb{B}^{\otimes H}) \le dH \log \left( 1 + \frac{2L_b}{\epsilon} \right).$$

Similarly, we consider an $\epsilon$-net defined for $\Pi(\mathcal{H})$ defined as the following: for any $\pi = \{\pi_h\}_{h=1}^H \in \Pi(\mathcal{H})$, there exists an $\pi' = \{\pi_h'\}_{h=1}^H$ in the $\epsilon$-net such that

$$\|\pi_h - \pi_h'\|_{\infty,1} \le \epsilon.$$

Then by Lemma H.3, the cardinality of this $\epsilon$-net is upper bounded by

$$\log \mathcal{N}_\epsilon^{\infty,1}(\Pi(\mathcal{H})) \le dH \log \left( 1 + \frac{16L_\pi}{\epsilon} \right)$$

For the dual function class $\mathbb{G}_{\mathrm{lin}}$, recall the definition of the critical radius $\alpha_{\mathbb{G},n}$ in Assumption 4.2. The next lemma bound the critical radius of the linear dual function class $\mathbb{G} = \mathbb{G}_{\mathrm{lin}}$.

**Lemma H.4** (Lemma D.3 in Duan et al. 2021). *For the function class $\mathbb{G}_{\mathrm{lin}}$ defined in Definition H.1, its critical radius $\alpha_{\mathbb{G},n}$ satisfies*

$$\alpha_{\mathbb{G},n} = M_{\mathbb{G}} \sqrt{\frac{2d}{n}},$$

*where $M_{\mathbb{G}} := \sup_{g \in \mathbb{G}_{\mathrm{lin}}} \|g\|_\infty$.*

## H.3   Proof of Corollary H.2

We first introduce some lemmas needed for proving Corollary H.2. Their proof is deferred to Appendix H.4.1 and H.4.2.

**Lemma H.5** (Alternative of Lemma D.2 in the linear case). *Let the function, policy and dual function class $\mathbb{B} = \mathbb{B}_{\mathrm{lin}}$, $\Pi(\mathcal{H}) = \Pi_{\mathrm{lin}}$ and $\mathbb{G} = \mathbb{G}_{\mathrm{lin}}$ be defined as in Definition H.1. Then under Assumption 3.2, 4.2, and 4.3, by setting $\xi$ such that*

$$\xi = C_2 \cdot \left( \lambda + \frac{1}{\lambda} \right) \cdot \frac{M_{\mathbb{B}}^2 M_{\mathbb{G}}^2 dH \log \left( 1 + L_b L_\pi H n / \delta \right)}{n},$$

*for some problem-independent universal constant $C_2 > 0$, it holds with probability at least $1 - \delta$ that $\mathbf{b}^\pi \in \mathrm{CR}^\pi(\xi)$ for any policy $\pi \in \Pi(\mathcal{H})$.*

**Lemma H.6** (Alternative of Lemma D.3 in the linear case). *Under Assumption 3.2, 4.2, 4.3, and 4.3, by setting the same $\xi$ as in Lemma H.5, with probability at least $1 - \delta$, for any policy $\pi \in \Pi(\mathcal{H})$, $\mathbf{b} \in \mathrm{CR}^\pi(\xi)$, and step $h$,*

$$\sqrt{\mathcal{L}_h^\pi(b_h, b_{h+1})} \leq \widetilde{C}_2 \cdot (1 + \lambda) M_\mathbb{B} M_\mathbb{G} \cdot \sqrt{dH \log\left(1 + L_b L_\pi H n/\delta\right)/n} + \widetilde{C}_2 \cdot M_\mathbb{G}^{1/2} \epsilon_\mathbb{B}^{1/4},$$

*for some problem-independent universal constant $\widetilde{C}_2 > 0$.*

We are now ready to prove Corollary H.2.

*Proof of Corollary H.2.* We follow the proof of Theorem 4.4 and write

$$J(\pi^\star) - J(\widehat{\pi})$$
$$= \underbrace{\left(J(\pi^\star, \mathbf{b}^{\pi^\star}) - \widehat{J}(\pi^\star, \mathbf{b}^{\pi^\star})\right)}_{\text{(i)}} + \underbrace{\left(\widehat{J}(\pi^\star, \mathbf{b}^{\pi^\star}) - \widehat{J}(\widehat{\pi}, \mathbf{b}^{\widehat{\pi}})\right)}_{\text{(ii)}} + \underbrace{\left(\widehat{J}(\widehat{\pi}, \mathbf{b}^{\widehat{\pi}}) - J(\widehat{\pi}, \mathbf{b}^{\widehat{\pi}})\right)}_{\text{(iii)}}.$$

$$\text{(H.1)}$$

We deal with term (ii) first. By Lemma H.5, with probability at least $1 - \delta/2$, $\mathbf{b}^{\pi^\star} \in \mathrm{CR}^{\pi^\star}(\xi)$ and $\mathbf{b}^{\widehat{\pi}} \in \mathrm{CR}^{\widehat{\pi}}(\xi)$, which indicates that

$$\text{(ii)} = \widehat{J}(\pi^\star, \mathbf{b}^{\pi^\star}) - \widehat{J}(\widehat{\pi}, \mathbf{b}^{\widehat{\pi}}) \leq \max_{\mathbf{b} \in \mathrm{CR}^{\pi^\star}(\xi)} \widehat{J}(\pi^\star, \mathbf{b}) - \min_{\mathbf{b} \in \mathrm{CR}^{\widehat{\pi}}(\xi)} \widehat{J}(\widehat{\pi}, \mathbf{b}).$$

Then following (G.2), we can upper bound term (ii) by

$$\text{(ii)} \leq 2 \max_{\mathbf{b} \in \mathrm{CR}^{\pi^\star}(\xi)} \left|\widehat{J}(\pi^\star, \mathbf{b}) - \widehat{J}(\pi^\star, \mathbf{b}^{\pi^\star})\right|$$
$$\leq 2 \underbrace{\max_{\mathbf{b} \in \mathrm{CR}^{\pi^\star}(\xi)} \left|\widehat{J}(\pi^\star, \mathbf{b}) - J(\pi^\star, \mathbf{b})\right|}_{\text{(iv)}} + 2 \underbrace{\max_{\mathbf{b} \in \mathrm{CR}^{\pi^\star}(\xi)} \left|J(\pi^\star, \mathbf{b}) - J(\pi^\star, \mathbf{b}^{\pi^\star})\right|}_{\text{(v)}}$$
$$+ 2 \underbrace{\left|J(\pi^\star, \mathbf{b}^{\pi^\star}) - \widehat{J}(\pi^\star, \mathbf{b}^{\pi^\star})\right|}_{\text{(vi)}}. \qquad \text{(H.2)}$$

To bound term (v), we invoke Lemma D.1 which holds regardless of the underlying function classes and obtain that

$$\text{(v)} = 2 \max_{\mathbf{b} \in \mathrm{CR}^{\pi^\star}(\xi)} \sum_{h=1}^H \gamma^{h-1} \sqrt{C^\pi} \cdot \sqrt{\mathcal{L}_h^\pi(b_h, b_{h+1})} \leq 2\sqrt{C^{\pi^\star}} \sum_{h=1}^H \gamma^{h-1} \max_{\mathbf{b} \in \mathrm{CR}^{\pi^\star}(\xi)} \sqrt{\mathcal{L}_h^\pi(b_h, b_{h+1})}.$$

Now by Lemma H.6, with probability at least $1 - \delta$, $\max_{\mathbf{b} \in \mathrm{CR}^{\pi^\star}(\xi)} \sqrt{\mathcal{L}_h^\pi(b_h, b_{h+1})}$ is bounded by

$$\max_{\mathbf{b} \in \mathrm{CR}^{\pi^\star}(\xi)} \sqrt{\mathcal{L}_h^\pi(b_h, b_{h+1})}$$
$$\leq \widetilde{C}_2 \cdot (1 + \lambda) M_\mathbb{B} M_\mathbb{G} \cdot \sqrt{\frac{dH \log\left(1 + L_b L_\pi H n/\delta\right)}{n}} + \widetilde{C}_2 \cdot M_\mathbb{G}^{1/2} \epsilon_\mathbb{B}^{1/4}, \quad \forall h \in [H]. \quad \text{(H.3)}$$

Now we deal with the term (i), (iii), (iv), and (vi), respectively. To this end, we apply uniform concentration inequalities to bound $J(\pi, \mathbf{b})$ and $\widehat{J}(\pi, \mathbf{b})$ uniformly over the $\epsilon$-net of $\pi$ and $\mathbf{b}$ as described in the proof of Lemma H.5. By Hoeffding's inequality, we have that, with probability at least $1 - \delta$, for all $\pi$ and $\mathbf{b}$ in their $\epsilon$-nets,

$$\left|J(\pi, \mathbf{b}) - \widehat{J}(\pi, \mathbf{b})\right| \leq \sqrt{\frac{2M_\mathbb{B}^2 \log(\mathcal{N}_{\epsilon, \mathbf{b}} \mathcal{N}_{\epsilon, \pi}/\delta)}{n}},$$

where $\mathcal{N}_{\epsilon, \pi}$ and $\mathcal{N}_{\epsilon, \mathbf{b}}$ are the covering numbers defined in Appendix H.2. Here we use the regularity assumption that $|\sum_{a \in \mathcal{A}} b_1^\pi(a, w)| \leq M_\mathbb{B}$ for all $w \in \mathcal{W}$ and the definition of $J(\pi, \mathbf{b})$ from (D.1). Consequently, for all $\pi \in \Pi(\mathcal{H})$ and $\mathbf{b} \in \mathbb{B}^{\otimes H}$, we have

$$\left|J(\pi, \mathbf{b}) - \widehat{J}(\pi, \mathbf{b})\right| \leq \sqrt{\frac{2M_\mathbb{B}^2 \log(\mathcal{N}_{\epsilon, \mathbf{b}} \mathcal{N}_{\epsilon, \pi}/\delta)}{n}} + 2M_\mathbb{B}\epsilon. \qquad \text{(H.4)}$$

Next, all of (i), (iii), (iv), and (vi) are bounded by the R.H.S. of (H.4). Finally, by (H.1), (H.2), (H.3) and (H.4), we have that

$$J(\pi^\star) - J(\widehat{\pi})$$
$$\leq (i) + (iii) + (iv) + (vi) + (v)$$
$$\leq 2\sqrt{C^{\pi^\star}} \sum_{h=1}^{H} \gamma^{h-1} \left[ \widetilde{C}_2 \cdot (1+\lambda) M_{\mathbb{B}} M_{\mathbb{G}} \cdot \sqrt{\frac{dH \log\left(1 + L_b L_\pi H n/\delta\right)}{n}} + \widetilde{C}_2 \cdot M_{\mathbb{G}}^{1/2} \epsilon_{\mathbb{B}}^{1/4} \right]$$
$$+ 4 \left[ \sqrt{\frac{2 M_{\mathbb{B}}^2 \log(\mathcal{N}_{\epsilon,\mathbf{b}} \mathcal{N}_{\epsilon,\pi}/\delta)}{n}} + 2 M_{\mathbb{B}} \epsilon \right].$$

Finally, by taking $\epsilon = 1/n^2$, and plugging in the values of $\mathcal{N}_{\epsilon,\mathbf{b}}$ and $\mathcal{N}_{\epsilon,\pi}$ from Lemma H.3, we get

$$J(\pi^\star) - J(\widehat{\pi})$$
$$\leq 2\sqrt{C^{\pi^\star}} \sum_{h=1}^{H} \gamma^{h-1} \left[ \widetilde{C}_2 \cdot (1+\lambda) M_{\mathbb{B}} M_{\mathbb{G}} \cdot \sqrt{\frac{dH \log\left(1 + L_b L_\pi H n/\delta\right)}{n}} + \widetilde{C}_2 \cdot M_{\mathbb{G}}^{1/2} \epsilon_{\mathbb{B}}^{1/4} \right]$$
$$+ C_3 M_{\mathbb{B}} \sqrt{\frac{dH \log(1 + L_b L_\pi n/\delta)}{n}},$$

where $C_3$ is some problem-independent universal constant. We then simplify the expression and use the fact that

$$M_{\mathbb{G}} = \sup_{a,z} |g(a,z)| = \sup_{a,z} |\langle \nu(a,z), \omega \rangle| \leq \sup_{a,z} \|\nu(a,z)\|_2 \cdot \|\omega\|_2 \leq L_g.$$

This gives the result of Corollary H.2. $\qquad\square$

## H.4    PROOF OF LEMMAS IN APPENDIX H

### H.4.1    PROOF OF LEMMA H.5

*Proof of Lemma H.5.* First, for any $\epsilon \in (0,1)$, consider arbitrary $\pi = \{\pi_h\}_{h=1}^{H}$ and $\pi' = \{\pi'_h\}_{h=1}^{H}$ in $\Pi_{\mathrm{lin}}$ such that $\|\pi_h - \pi'_h\|_{\infty,1} \leq \epsilon$ for all $h \in [H]$. And consider arbitrary $\mathbf{b} = \{b_h\}_{h=1}^{H}$ and $\mathbf{b}' = \{b'_h\}_{h=1}^{H}$ in $\mathbb{B}^{\otimes H}$ such that $\|b_h - b'_h\|_{\infty} \leq \epsilon$ for all $h \in [H]$. Then by definition of $\Phi_{\pi,h}^{\lambda}(b_h, b_{h+1}; g)$ in (3.9) and $\widehat{\Phi}_{\pi,h}^{\lambda}(b_h, b_{h+1}; g)$ in (3.10), and that $\Phi_{\pi,h}^{\lambda} = \Phi_{\pi,h}^0$ and $\widehat{\Phi}_{\pi,h}^{\lambda} = \widehat{\Phi}_{\pi,h}^0$, one can easily get that

$$\left| \Phi_{\pi,h}(b_h, b_{h+1}; g) - \Phi_{\pi',h}(b'_h, b'_{h+1}; g) \right| \leq [2\epsilon + \gamma \cdot (\epsilon + \epsilon M_{\mathbb{B}})] \cdot M_{\mathbb{G}} \leq 4 M_{\mathbb{B}} M_{\mathbb{G}} \epsilon,$$
$$\left| \widehat{\Phi}_{\pi,h}(b_h, b_{h+1}; g) - \widehat{\Phi}_{\pi',h}(b'_h, b'_{h+1}; g) \right| \leq [2\epsilon + \gamma \cdot (\epsilon + \epsilon M_{\mathbb{B}})] \cdot M_{\mathbb{G}} \leq 4 M_{\mathbb{B}} M_{\mathbb{G}} \epsilon, \qquad \text{(H.5)}$$

for all $g \in \mathbb{G}$.

Now, same as in the proof of Lemma D.2, we want to show: for any $\pi \in \Pi(\mathcal{H})$,

$$\max_{g \in \mathbb{G}} \widehat{\Phi}_{\pi,h}^{\lambda}(b_h^{\pi}, b_{h+1}^{\pi}; g) \leq \xi.$$

The rest of the proof would be very similar to that of Lemma D.2 with an additional covering argument. To begin with, we again write $\widehat{\Phi}_{\pi,h}^{\lambda}(b_h^{\pi}, b_{h+1}^{\pi}; g) = \widehat{\Phi}_{\pi,h}(b_h^{\pi}, b_{h+1}^{\pi}; g) - \lambda \|g\|_{2,n}^2$.

Same as (F.10), we have that with probability at least $1 - \delta/2$,

$$\left| \|g\|_{2,n}^2 - \|g\|_2^2 \right| \leq \frac{1}{2} \|g\|_2^2 + \frac{M_{\mathbb{G}}^2 \log(2c_1/\zeta)}{2c_2 n}, \quad \forall g \in \mathbb{G}, \qquad \text{(H.6)}$$

where $\zeta = \min\{\delta, 2c_1 \exp(-c_2 n \alpha_{\mathbb{G},n}^2 / M_{\mathbb{G}}^2)\}$ and $c_1, c_2$ are some universal constants.

Next, we upper bound $|\widehat{\Phi}_{\pi',h}(b_h, b_{h+1}; g) - \Phi_{\pi',h}(b_h, b_{h+1}; g)|$ for any $\pi \in \Pi(\mathcal{H})$, and $\mathbf{b} \in \mathbb{B}^{\otimes H}$. We first prove this for a fixed $\epsilon$-net of $\Pi(\mathcal{H})$ and $\mathbb{B}^{\otimes H}$. Specifically, choose an $\epsilon$-net of $\Pi(\mathcal{H})$ such that for any $\pi = \{\pi_h\}_{h=1}^{H}$ and $\pi' = \{\pi'_h\}_{h=1}^{H}$ in this $\epsilon$-net, it holds that $\|\pi_h - \pi'_h\|_{\infty,1} \leq \epsilon$ for all

$h$. Also choose an $\epsilon$-net of $\mathbb{B}^{\otimes H}$ such that for any $\mathbf{b} = \{b_h\}_{h=1}^H$ and $\mathbf{b}' = \{b_h'\}_{h=1}^H$ in the $\epsilon$-net, it holds that $\|b_h - b_h'\|_\infty \leq \epsilon$ for all $h$. Denote the cardinality of these two $\epsilon$-net by $\mathcal{N}_{\epsilon,\pi}$ and $\mathcal{N}_{\epsilon,\mathbf{b}}$, respectively. Then by the same argument behind (F.11), we get that, with probability at least $1 - \delta/2$, for any $\pi$ and $\mathbf{b}$ in their $\epsilon$-nets, and for any $g \in \mathbb{G}$,

$$
\left| \widehat{\Phi}_{\pi,h}(b_h, b_{h+1}; g) - \Phi_{\pi,h}(b_h, b_{h+1}; g) \right|
$$
$$
\leq 18L\|g\|_2 \sqrt{\frac{M_{\mathbb{G}}^2 \log\left(2c_1 \mathcal{N}_{\epsilon,\mathbf{b}}^2 \mathcal{N}_{\epsilon,\pi} H/\zeta\right)}{c_2 n}} + \frac{18L M_{\mathbb{G}}^2 \log\left(2c_1 \mathcal{N}_{\epsilon,\mathbf{b}}^2 \mathcal{N}_{\epsilon,\pi} H/\zeta\right)}{c_2 n}, \quad \text{(H.7)}
$$

where $\zeta' = \min\{\delta, 2c_1 \mathcal{N}_{\epsilon,\mathbf{b}}^2 \mathcal{N}_{\epsilon,\pi} H \exp(-c_2 n \alpha_{\mathbb{G},n}^2/M_{\mathbb{G}}^2)\}$.

Now for any $\pi \in \Pi(\mathcal{H})$ and $\mathbf{b} \in \mathbb{B}^{\otimes H}$, by our construction of the $\epsilon$-nets, we can find a $\pi'$ and $\mathbf{b}'$ in the $\epsilon$-nets such that $\|\pi_h - \pi_h'\|_{\infty,1} \leq \epsilon$ and $\|b_h - b_h'\|_\infty \leq \epsilon$ for all $h$. Then we have that with probability at least $1 - \delta/2$, for any $\pi \in \Pi(\mathcal{H})$ and $\mathbf{b} \in \mathbb{B}^{\otimes H}$, and for any $g \in \mathbb{G}$,

$$
\left| \widehat{\Phi}_{\pi,h}(b_h, b_{h+1}; g) - \Phi_{\pi,h}(b_h, b_{h+1}; g) \right|
$$
$$
\leq |\widehat{\Phi}_{\pi,h}(b_h, b_{h+1}; g) - \widehat{\Phi}_{\pi',h}(b_h', b_{h+1}'; g)| + |\widehat{\Phi}_{\pi',h}(b_h', b_{h+1}'; g) - \Phi_{\pi',h}(b_h', b_{h+1}'; g)|
$$
$$
+ |\Phi_{\pi',h}(b_h', b_{h+1}'; g) - \Phi_{\pi,h}(b_h, b_{h+1}; g)|
$$
$$
\leq 8 M_{\mathbb{B}} M_{\mathbb{G}} \cdot \epsilon + 18L\|g\|_2 \sqrt{\frac{M_{\mathbb{G}}^2 \log\left(2c_1 \mathcal{N}_{\epsilon,\mathbf{b}}^2 \mathcal{N}_{\epsilon,\pi} H/\zeta\right)}{c_2 n}} + \frac{18L M_{\mathbb{G}}^2 \log\left(2c_1 \mathcal{N}_{\epsilon,\mathbf{b}}^2 \mathcal{N}_{\epsilon,\pi} H/\zeta\right)}{c_2 n},
$$
$$
\text{(H.8)}
$$

where the first step is by the triangle inequality and the second steps is by (H.5) and (H.7).

Now combine (H.6) and (H.8) with a union bound, we conclude that, with probability at least $1 - \delta$, for any $\pi \in \Pi(\mathcal{H})$,

$$
\max_{g \in \mathbb{G}} \widehat{\Phi}_{\pi,h}^\lambda(b_h^\pi, b_{h+1}^\pi; g)
$$
$$
= \max_{g \in \mathbb{G}} \left\{ \widehat{\Phi}_{\pi,h}(b_h^\pi, b_{h+1}^\pi; g) - \lambda\|g\|_{2,n}^2 \right\}
$$
$$
\leq \max_{g \in \mathbb{G}} \left\{ \Phi_{\pi,h}(b_h^\pi, b_{h+1}^\pi; g) - \lambda\|g\|_2^2 + \frac{\lambda}{2}\|g\|_2^2 + \frac{\lambda M_{\mathbb{G}}^2 \log(2c_1/\zeta)}{2c_2 n}, \right.
$$
$$
\left. + 18L\|g\|_2 \sqrt{\frac{M_{\mathbb{G}}^2 \log\left(2c_1 \mathcal{N}_{\epsilon,\mathbf{b}}^2 \mathcal{N}_{\epsilon,\pi} H/\zeta\right)}{c_2 n}} + \frac{18L M_{\mathbb{G}}^2 \log\left(2c_1 \mathcal{N}_{\epsilon,\mathbf{b}}^2 \mathcal{N}_{\epsilon,\pi} H/\zeta\right)}{c_2 n} \right\} + 8 M_{\mathbb{B}} M_{\mathbb{G}} \epsilon
$$
$$
\leq \max_{g \in \mathbb{G}} \Phi_{\pi,h}(b_h^\pi, b_{h+1}^\pi; g) + \max_{g \in \mathbb{G}} \left\{ -\frac{\lambda}{2}\|g\|_2^2 + 18L\|g\|_2 \sqrt{\frac{M_{\mathbb{G}}^2 \log\left(2c_1 \mathcal{N}_{\epsilon,\mathbf{b}}^2 \mathcal{N}_{\epsilon,\pi} H/\zeta\right)}{c_2 n}} \right\}
$$
$$
+ \frac{\lambda M_{\mathbb{G}}^2 \log(2c_1/\zeta)}{2c_2 n} + \frac{18L M_{\mathbb{G}}^2 \log\left(2c_1 \mathcal{N}_{\epsilon,\mathbf{b}}^2 \mathcal{N}_{\epsilon,\pi} H/\zeta\right)}{c_2 n} + 8 M_{\mathbb{B}} M_{\mathbb{G}} \epsilon
$$
$$
\leq \frac{728 L^2 M_{\mathbb{G}}^2 \log(2c_1 \mathcal{N}_{\epsilon,\mathbf{b}}^2 \mathcal{N}_{\epsilon,\pi} H/\zeta')}{\lambda n} + \frac{\lambda M_{\mathbb{G}}^2 \log(2c_1/\zeta)}{2c_2 n}
$$
$$
+ \frac{18L M_{\mathbb{G}}^2 \log(2c_1 \mathcal{N}_{\epsilon,\mathbf{b}}^2 \mathcal{N}_{\epsilon,\pi} H/\zeta')}{c_2 n} + 8 M_{\mathbb{B}} M_{\mathbb{G}} \epsilon, \quad \text{(H.9)}
$$

with $\zeta = \min\{\delta, 2c_1 \exp(-c_2 n \alpha_{\mathbb{G},n}^2/M_{\mathbb{G}}^2)\}$ and $\zeta' = \min\{\delta, 2c_1 \mathcal{N}_{\epsilon,\mathbf{b}}^2 \mathcal{N}_{\epsilon,\pi} H \exp(-c_2 n \alpha_{\mathbb{G},n}^2/M_{\mathbb{G}}^2)\}$ for any policy $\pi \in \Pi(\mathcal{H})$ and step $h$. Here the first inequality is by (H.6) and (H.8), the second inequality is trivial, and the last inequality holds from the fact that $\Phi_{\pi,h}(b_h^\pi, b_{h+1}^\pi; g) = 0$ and the fact that $\sup_{\|g\|_2} \{a\|g\|_2 - b\|g\|_2^2\} \leq a^2/4b$.

Now by Definition H.1, we apply Lemma H.3 with $\|\theta_h\|_2 \leq L_b$ and $\|\beta_h\| \leq L_\pi$ and get that

$$\log \mathcal{N}_{\epsilon,\pi} \leq dH \log \left( 1 + \frac{16 L_\pi}{\epsilon} \right),$$

$$\log \mathcal{N}_{\epsilon,b} \leq dH \log \left( 1 + \frac{2 L_b}{\epsilon} \right). \tag{H.10}$$

Now we pick $\epsilon = 1/n^2$, and together with (H.9) and (H.10), we get that

$$\max_{g \in \mathbb{G}} \widehat{\Phi}_{\pi,h}^\lambda(b_h^\pi, b_{h+1}^\pi; g) \leq C \cdot \frac{(\lambda + 1/\lambda) \, M_{\mathbb{B}}^2 M_{\mathbb{G}}^2 \left[ dH \log \left( 1 + L_b L_\pi H n/\delta \right) + n \alpha_{\mathbb{G},n}^2 / M_{\mathbb{G}}^2 \right]}{n} + C \cdot \frac{M_{\mathbb{B}} M_{\mathbb{G}}}{n^2},$$

where $C$ is some universal constant. Here we have plugged in the value of $\zeta$, $\zeta'$ and $L = 2 M_{\mathbb{B}}$. Finally, by plugging in the value of $\alpha_{\mathbb{G},n}$ from Lemma H.4, we conclude that

$$\max_{g \in \mathbb{G}} \widehat{\Phi}_{\pi,h}^\lambda(b_h^\pi, b_{h+1}^\pi; g) \leq C_1 \cdot \left( \lambda + \frac{1}{\lambda} \right) \cdot \frac{M_{\mathbb{B}}^2 M_{\mathbb{G}}^2 dH \log \left( 1 + L_b L_\pi H n/\delta \right)}{n} + C_1 \cdot \frac{M_{\mathbb{B}} M_{\mathbb{G}}}{n^2},$$

where $C_1$ is some problem-independent constant. Note that second term on the right hand side is smaller than the first term. Then the result follows from our choice of $\xi$ in Lemma H.5.

$\square$

### H.4.2 Proof of Lemma H.6

*Proof of Lemma H.6.* Consider any $\pi \in \Pi(\mathcal{H})$ and $\mathbf{b} = \{b_h\}_{h=1}^H \in \mathrm{CR}^\pi(\xi)$. Same as (F.12), we have

$$\max_{g \in \mathbb{G}} \widehat{\Phi}_{\pi,h}^\lambda(b_h, b_{h+1}; g)$$

$$\geq \underbrace{\max_{g \in \mathbb{G}} \left\{ \widehat{\Phi}_{\pi,h}(b_h, b_{h+1}; g) - \widehat{\Phi}_{\pi,h}(b_h^\star(b_{h+1}), b_{h+1}; g) - 2\lambda \|g\|_{2,n}^2 \right\}}_{(\star)}$$

$$- \max_{g \in \mathbb{G}} \widehat{\Phi}_{\pi,h}^\lambda(b_h^\star(b_{h+1}), b_{h+1}; g).$$

We again upper and lower bound term $(\star)$ respectively.

**Upper bound of term $(\star)$.** By the same argument as in the proof of Lemma F.1, we have that: for any $\mathbf{b} \in \mathbb{B}^{\otimes H}$, $\pi \in \Pi(\mathcal{H})$, and $h \in [H]$, it holds with probability at least $1 - \delta/2$ that

$$\max_{g \in \mathbb{G}} \widehat{\Phi}_{\pi,h}(b_h^\star(b_{h+1}), b_{h+1}; g) \leq \xi + \epsilon_{\mathbb{B}}^{1/2} M_{\mathbb{G}},$$

where $b_h^\star(b_{h+1})$ is defined in (F.3) and $\xi$ is defined in Lemma H.5. We then get

$$\max_{g \in \mathbb{G}} \widehat{\Phi}_{\pi,h}^\lambda(\widehat{b}_h(b_{h+1}), b_{h+1}; g) \leq \max_{g \in \mathbb{G}} \widehat{\Phi}_{\pi,h}^\lambda(b_h^\star(b_{h+1}), b_{h+1}; g) \leq \xi + \epsilon_{\mathbb{B}}^{1/2} M_{\mathbb{G}}, \tag{H.11}$$

where the first inequality follows from the definition of $\widehat{b}_h(b_{h+1})$ in (3.11). Also note that, by the construction of the confidence region $\mathrm{CR}^\pi(\xi)$, we have

$$\max_{g \in \mathbb{G}} \widehat{\Phi}_{\pi,h}^\lambda(b_h, b_{h+1}; g) - \max_{g \in \mathbb{G}} \widehat{\Phi}_{\pi,h}^\lambda(\widehat{b}_h(b_{h+1}), b_{h+1}; g) \leq \xi. \tag{H.12}$$

Furthermore, we can write

$$(\star) \leq \max_{g \in \mathbb{G}} \widehat{\Phi}_{\pi,h}^\lambda(b_h^\star(b_{h+1}), b_{h+1}; g) + \max_{g \in \mathbb{G}} \widehat{\Phi}_{\pi,h}^\lambda(b_h, b_{h+1}; g)$$

$$\leq \max_{g \in \mathbb{G}} \widehat{\Phi}_{\pi,h}^\lambda(b_h^\star(b_{h+1}), b_{h+1}; g)$$

$$+ \max_{g \in \mathbb{G}} \widehat{\Phi}_{\pi,h}^\lambda(b_h, b_{h+1}; g) - \max_{g \in \mathbb{G}} \widehat{\Phi}_{\pi,h}^\lambda(\widehat{b}_h(b_{h+1}), b_{h+1}; g)$$

$$+ \max_{g \in \mathbb{G}} \widehat{\Phi}_{\pi,h}^\lambda(\widehat{b}_h(b_{h+1}), b_{h+1}; g).$$

Combining with (H.11) and (H.12), we get that, with probability at least $1 - \delta/2$,

$$(\star) \leq 3\xi + 2\epsilon_{\mathbb{B}}^{1/2} M_{\mathbb{G}}. \tag{H.13}$$

**Lower bound of term** $(\star)$. First of all, same as (F.17), it holds with probability at least $1 - \delta/4$ that,

$$\left| \|g\|_{2,n}^2 - \|g\|_2^2 \right| \leq \frac{1}{2} \|g\|_2^2 + \frac{M_{\mathbb{G}}^2 \log(4c_1/\zeta)}{2c_2 n}, \quad \forall g \in \mathbb{G}, \tag{H.14}$$

where $\zeta = \min\{\delta, 4c_1 \exp(-c_2 n \alpha_{\mathbb{G},n}^2 / M_{\mathbb{G}}^2)\}$ for some absolute constants $c_1$ and $c_2$, and $\alpha_{\mathbb{G},n}$ is the critical radius of $\mathbb{G}$ defined in Assumption 4.2.

Second, we fix an $\epsilon$-net of $\Pi(\mathcal{H})$ and an $\epsilon$-net of $\mathbb{B}^{\otimes H}$, as described in Appendix H.2. Denote by $\mathcal{N}_{\epsilon,\pi}$ and $\mathcal{N}_{\epsilon,\mathbf{b}}$ their respective covering numbers. Then by the same argument behind (F.18) and a union bound, we get that, with probability at least $1 - \delta/4$, for all $\pi = \{\pi_h\}_{h=1}^H$, $\mathbf{b} = \{b_h\}_{h=1}^H$ and $\mathbf{b}' = \{b_h'\}_{h=1}^H$ in their $\epsilon$-nets, and for all $g \in \mathbb{G}$,

$$\left| \left( \widehat{\Phi}_{\pi,h}(b_h, b_{h+1}; g) - \widehat{\Phi}_{\pi,h}(b_h', b_{h+1}; g) \right) - \left( \Phi_{\pi,h}(b_h, b_{h+1}; g) - \Phi_{\pi,h}(b_h', b_{h+1}; g) \right) \right|$$

$$\leq 18L\|g\|_2 \sqrt{\frac{M_{\mathbb{G}}^2 \log(4c_1 \mathcal{N}_{\epsilon,\mathbf{b}}^3 \mathcal{N}_{\epsilon,\pi} H/\zeta')}{c_2 n}} + \frac{18LM_{\mathbb{G}}^2 \log\left(4c_1 \mathcal{N}_{\epsilon,\mathbf{b}}^3 \mathcal{N}_{\epsilon,\pi} H/\zeta'\right)}{c_2 n}, \tag{H.15}$$

where $\zeta' = \min\{\delta, 4c_1 \mathcal{N}_{\epsilon,\mathbf{b}}^3 \mathcal{N}_{\epsilon,\pi} H \exp(-c_2 n \alpha_{\mathbb{G},n}^2 / M_{\mathbb{G}}^2)\}$.

We then use (H.5), and conclude that, with probability at least $1 - \delta/4$, for all $\pi \in \Pi(\mathcal{H})$, and $\mathbf{b}$, $\mathbf{b}' \in \mathbb{B}^{\otimes H}$, and $g \in \mathbb{G}$,

$$\left| \left( \widehat{\Phi}_{\pi,h}(b_h, b_{h+1}; g) - \widehat{\Phi}_{\pi,h}(b_h', b_{h+1}; g) \right) - \left( \Phi_{\pi,h}(b_h, b_{h+1}; g) - \Phi_{\pi,h}(b_h', b_{h+1}; g) \right) \right|$$

$$\leq 18L\|g\|_2 \sqrt{\frac{M_{\mathbb{G}}^2 \log(4c_1 \mathcal{N}_{\epsilon,\mathbf{b}}^3 \mathcal{N}_{\epsilon,\pi} H/\zeta')}{c_2 n}} + \frac{18LM_{\mathbb{G}}^2 \log\left(4c_1 \mathcal{N}_{\epsilon,\mathbf{b}}^3 \mathcal{N}_{\epsilon,\pi} H/\zeta'\right)}{c_2 n} + 8M_{\mathbb{B}} M_{\mathbb{G}} \epsilon. \tag{H.16}$$

In the sequel, for simplicity, we denote that

$$\iota_n := \sqrt{\frac{M_{\mathbb{G}}^2 \log(4c_1 \mathcal{N}_{\epsilon,\mathbf{b}}^3 \mathcal{N}_{\epsilon,\pi} H/\zeta')}{c_2 n}}, \quad \iota_n' := \sqrt{\frac{M_{\mathbb{G}}^2 \log(4c_1/\zeta)}{2c_2 n}}, \tag{H.17}$$

where $\zeta$ and $\zeta'$ are same as in (H.14) and (H.15). Furthermore, given fixed $b_h, b_{h+1} \in \mathbb{B}$, we denote

$$g_h^\pi := \frac{1}{2\lambda} \ell_h^\pi(b_h, b_{h+1}) \in \mathbb{G}, \tag{H.18}$$

where $\ell_h^\pi$ is defined by (F.1) and $g_h^\pi \in \mathbb{G}$ follows from Assumption 4.3. We then have

$$(\star) = \max_{g \in \mathbb{G}} \left\{ \widehat{\Phi}_{\pi,h}(b_h, b_{h+1}; g) - \widehat{\Phi}_{\pi,h}(b_h^\star(b_{h+1}), b_{h+1}; g) - 2\lambda \|g\|_{2,n}^2 \right\}$$

$$\geq \widehat{\Phi}_{\pi,h}(b_h, b_{h+1}; g_h^\pi/2) - \widehat{\Phi}_{\pi,h}(b_h^\star(b_{h+1}), b_{h+1}; g_h^\pi/2) - \frac{\lambda}{2} \|g_h^\pi\|_{2,n}^2,$$

where the inequality holds because $g_h^\pi/2 \in \mathbb{G}$.

Together with (H.14) and (H.16), we have

$$(\star) \geq \Phi_{\pi,h}(b_h, b_{h+1}; g_h^\pi/2) - \Phi_{\pi,h}(b_h^\star(b_{h+1}), b_{h+1}; g_h^\pi/2) - 18L\iota_n \|g_h^\pi\|_2 - 18L\iota_n^2$$

$$- 8M_{\mathbb{B}} M_{\mathbb{G}} \epsilon - \frac{\lambda}{2} \left( \frac{3}{2} \|g_h^\pi\|_2^2 + \iota_n'^2 \right)$$

$$\geq \lambda \|g_h^\pi\|_2^2 - 18L\iota_n \|g_h^\pi\|_2 - \epsilon_{\mathbb{B}}^{1/2} M_{\mathbb{G}} - 18L\iota_n^2 - 8M_{\mathbb{B}} M_{\mathbb{G}} \epsilon - \frac{\lambda}{2} \left( \frac{3}{2} \|g_h^\pi\|_2^2 + \iota_n'^2 \right)$$

$$= \frac{\lambda}{4} \|g_h^\pi\|_2^2 - 18L\iota_n \|g_h^\pi\|_2 - \epsilon_{\mathbb{B}}^{1/2} M_{\mathbb{G}} - 18L\iota_n^2 - 8M_{\mathbb{B}} M_{\mathbb{G}} \epsilon - \frac{\lambda}{2} \iota_n'^2, \tag{H.19}$$

where the second inequality follows from the same reason as in (F.20).

Finally, combine (H.19) and (H.13) and we get

$$\frac{\lambda}{4}\|g_h^\pi\|_2^2 - 18L\iota_n\|g_h^\pi\|_2 - \epsilon_{\mathbb{B}}^{1/2}M_{\mathbb{G}} - 18L\iota_n^2 - 8M_{\mathbb{B}}M_{\mathbb{G}}\epsilon - \frac{\lambda}{2}\iota_n'^2 \leq 3\xi + 2\epsilon_{\mathbb{B}}^{1/2}M_{\mathbb{G}}.$$

This gives the following quadratic inequality w.r.t. $\|g_h^\pi\|_2$

$$\lambda\|g_h^\pi\|_2^2 - \underbrace{72L\iota_n}_{\text{A}}\|g_h^\pi\|_2 - \underbrace{4\left[18L\iota_n^2 + \frac{\lambda}{2}{\iota_n'}^2 + 3\xi + 8M_{\mathbb{B}}M_{\mathbb{G}}\epsilon + 3\epsilon_{\mathbb{B}}^{1/2}M_{\mathbb{G}}\right]}_{\text{B}} \leq 0.$$

By the fact that $x^2 - \mathrm{A}x - \mathrm{B} \leq 0$ implies $x \leq (\mathrm{A} + \sqrt{\mathrm{A}^2 + 4\mathrm{B}})/2 \leq \mathrm{A} + \sqrt{\mathrm{B}}$, we have

$$\|g_h^\pi\|_2 \leq \frac{72L\iota_n}{\lambda} + \sqrt{\frac{4}{\lambda}\left[18L\iota_n^2 + \frac{\lambda}{2}{\iota_n'}^2 + 3\xi + 8M_{\mathbb{B}}M_{\mathbb{G}}\epsilon + 3\epsilon_{\mathbb{B}}^{1/2}M_{\mathbb{G}}\right]}.$$

We then plug in the values of $\iota_n$ and $\iota_n'$ from (H.17), $\xi$ from Lemma H.5, $\zeta$ and $\zeta'$ from below (H.14) and (H.15), $\mathcal{N}_{\epsilon,\mathbf{b}}$ and $\mathcal{N}_{\epsilon,\pi}$ from Lemma H.3, $\alpha_{\mathbb{G},n}$ from Lemma H.4, and set $\epsilon = 1/n^2$. Simplify the expression and we get

$$\|g_h^\pi\|_2 \leq C \cdot \left(1 + \frac{1}{\lambda}\right)M_{\mathbb{B}}M_{\mathbb{G}} \cdot \sqrt{\frac{dH\log\left(1 + L_bL_\pi Hn/\delta\right)}{n}} + C \cdot \frac{M_{\mathbb{G}}^{1/2}\epsilon_{\mathbb{B}}^{1/4}}{\lambda},$$

where $C$ is some problem-independent universal constant. By (H.18) and (3.6), we have $\mathcal{L}_h^\pi(b_h, b_{h+1}) = \|2\lambda g_h^\pi\|_2^2$. It follows that

$$\sqrt{\mathcal{L}_h^\pi(b_h, b_{h+1})} = 2\lambda\|g_h^\pi\|_2 \leq \widetilde{C}_2 \cdot (1 + \lambda)M_{\mathbb{B}}M_{\mathbb{G}} \cdot \sqrt{\frac{dH\log\left(1 + L_bL_\pi Hn/\delta\right)}{n}} + \widetilde{C}_2 \cdot M_{\mathbb{G}}^{1/2}\epsilon_{\mathbb{B}}^{1/4},$$

for some constant $\widetilde{C}_2$. This finishes the proof. $\qquad\square$

# I AUXILIARY LEMMAS

We introduce some useful lemmas for the uniform concentration over function classes. Before we present the lemmas, we first introduce several notations. For a function class $\mathcal{F}$ on a probability space $(\mathcal{X}, P)$, we denote by $\|f\|_2^2$ the expectation of $f(X)^2$, that is $\|f\|_2^2 = \mathbb{E}_{X\sim\mathcal{P}}[f(X)^2]$. Also, we denote by

$$\mathcal{R}_n(\mathcal{F}, \delta) := \mathbb{E}\left[\sup_{f\in\mathcal{F}:\|f\|_2\leq\delta}\left|\frac{1}{n}\sum_{i=1}^n \epsilon_i f(X_i)\right|\right] \tag{I.1}$$

the localized Rademacher complexity of $\mathcal{F}$ with scale $\delta > 0$ and size $n \in \mathbb{N}$. Here $\{\epsilon_i\}_{i=1}^b$ and $\{X_i\}_{i=1}^n$ are i.i.d. and independent. Each $\epsilon_i$ is uniformly distributed on $\{+1, -1\}$ and each $X_i$ is distributed according to $P$. Finally, we denote by $\text{star}(\mathcal{F})$ the star-shaped set induced by set $\mathcal{F}$ as

$$\text{star}(\mathcal{F}) = \{\alpha f : \alpha \in [0, 1], f \in \mathcal{F}\}. \tag{I.2}$$

Now we are ready to present the lemmas for uniform concentration inequalities.

**Lemma I.1** (Localized Uniform Concentration 1 (Wainwright, 2019)). *Given a star-shaped and $b$-uniformly bounded function class $\mathcal{F}$, let $\delta_n$ be any positive solution of the inequality*

$$\mathcal{R}_n(\mathcal{F}; \delta) \leq \frac{\delta^2}{b}.$$

*Then for any $t \geq \delta_n$, we have that*

$$\left|\|f\|_n^2 - \|f\|_2^2\right| \leq \frac{1}{2}\|f\|_2^2 + \frac{1}{2}t^2, \quad \forall f \in \mathcal{F}$$

*with probability at least $1 - c_1\exp(-c_2 nt^2/b^2)$. If in addition $n\delta_n^2 \geq 2\log\left(4\log\left(1/\delta_n\right)\right)/c_2$, then we have that*

$$\left|\|f\|_n - \|f\|_2\right| \leq c_0\delta_n, \quad \forall f \in \mathcal{F}$$

*with probability at least $1 - c_1'\exp(-c_2' n\delta_n^2/b^2)$.*

*Proof of Lemma I.1.* See Theorem 14.1 of Wainwright (2019) for a detailed proof. □

**Lemma I.2** (Localized Uniform Concentration 2 (Foster and Syrgkanis, 2019))**.** *Consider a star-shaped function class $\mathcal{F} : \mathcal{X} \mapsto \mathbb{R}$ with $\sup_{f \in \mathcal{F}} \|f\|_\infty \leq b$, and pick any $f^\star \in \mathcal{F}$. Also, consider a loss function $\ell : \mathbb{R} \times \mathcal{Y} \mapsto \mathbb{R}$ which is L-Lipschitz in its first argument with respect to the $\|\cdot\|_2$-norm. Now let $\delta_n^2 \geq 4 \log (41 \log (2c_2 n)) / (c_2 n)$ be any solution to the inequality:*

$$\mathcal{R}_n(\mathrm{star}(\mathcal{F} - f^\star); \delta) \leq \frac{\delta^2}{b}.$$

*Then for any $t \geq \delta_n$ and some absolute constants $c_1, c_2 > 0$, with probability $1 - c_1 \exp(-c_2 n t^2 / b^2)$ it holds that*

$$\left| \left( \widehat{\mathbb{E}}_n[\ell(f(x), y)] - \widehat{\mathbb{E}}_n \left[ \ell \left( f^\star(x), y \right) \right] \right) - \left( \mathbb{E}[\ell(f(x), y)] - \mathbb{E} \left[ \ell \left( f^\star(x), y \right) \right] \right) \right|$$
$$\leq 18Lt \left( \|f - f^\star\|_2 + t \right), \tag{I.3}$$

*for any $f \in \mathcal{F}$. If furthermore, the loss function $\ell$ is linear in $f$, i.e., $\ell((f + f')(x), y) = \ell(f(x), y) + \ell(f'(x), y)$ and $\ell(\alpha f(x), y) = \alpha \ell(f(x), z)$, then the lower bound on $\delta_n^2$ is not required.*

*Proof of Lemma I.2.* See Lemma 11 of Foster and Syrgkanis (2019) for a detailed proof. □

**Remark I.3.** *We remark that in the original Lemma 11 of Foster and Syrgkanis (2019), inequality (I.3) only holds for $\delta_n$, and we extend it to any $t \geq \delta_n$ since according to Lemma 13.6 of Wainwright (2019) we know that $\mathcal{R}_n(\mathcal{F}; \delta)/\delta$ is a non-increasing function of $\delta$ on $(0, +\infty)$, which indicates that $t \geq \delta_n$ also solves the inequality.*

