# OpenReview forum: "Pessimism in the Face of Confounders: Provably Efficient Offline Reinforcement Learning in Partially Observable Markov Decision Processes"
_ICLR.cc/2023/Conference — ICLR 2023 poster_

### Official Review · Reviewer_bBxr · 2022-10-17

**Confidence:** 3
**Clarity, Quality, Novelty And Reproducibility:** The paper writing is great. The proof…
**Correctness:** 4
**Technical Novelty And Significance:** 2
**Empirical Novelty And Significance:** Not applicable
**Recommendation:** 6

**Strength And Weaknesses:**

### Strength

The assumptions in this paper is reasonable or frequently considered in previous works. The authors introduced the ideas of their methods very clearly and make this paper easy to follow although mathematically heavy. Although using pessimism is not a surprising idea for offline learning in POMDP, there are still some novelty and interesting parts in this paper.

### Weakness

This paper is limited to the setting with latent states.

The learning framework (i.e. solving the bridge function for policy value estimation and using pessimism for policy learning so that sub-opt gap can be bounded via coverge w.r.t. optimal policies) is not something new and well-studied in previous literatures (although in different settings, the techniques and ideas are similar). So I'm feeling the novelty is kind of just above the marginal level.

Algorithm seems not computationally efficient.

**Summary Of The Paper:**

This paper studied offline policy evaluation and learning in POMDP setting with confounders. In this setting, the dataset is generated by a state depending policy, while the learner can only get access to observation. Based on some additional assumptions about negative control, bridge functions and concentrability coefficient and etc, the authors proposed a minimax approach to estimate policy value and a pessimism algorithm called P3O to identify near-optimal policy with bounded sub-optimality gap.

**Summary Of The Review:**

This is a solid paper with some interesting results, while because similar techniques/frameworks have been well studied in settings different from this paper, the contribution and novelty is at the marginal level to me.

---

> ### Author Response · Authors · 2022-11-11
> **Response to Reviewer bBxr (1)**
>
> Thank you very much for your valuable feedback! We appreciate all your constructive suggestions and we now address your concerns point-by-point in the following.
>
> **Q1:**
> This paper is limited to the setting with latent states.
>
> **A1:**
> The setting with latent states (i.e. confounded POMDP) is more general than the setting without latent states (i.e. MDP), and thus the former setting can be reduced to the latter setting. Our work can cover the setting without latent states. For example, under the tabular MDP setting, we have $O=S$, and we can set the negative control variables $Z$ and $W$ to be $S$.
>
> Furthermore, in terms of technical challenges, the case without latent states is less challenging than ours, as there are no latent confounders that affect both the responses and the actions. Consequently, for confounded POMDP, we require new techniques compared to pessimism in MDP papers (e.g. [1]). We further discuss the technical novelty in the answer to **Q2**.
>
> **Q2:**
> Novelty: The learning framework (i.e. solving the bridge function for policy value estimation and using pessimism for policy learning so that sub-opt gap can be bounded via coverage w.r.t. optimal policies) is not something new and well-studied in previous literatures (although in different settings, the techniques and ideas are similar). So I'm feeling the novelty is kind of just above the marginal level.
>
> **A2:**
> Our work is very different from the previous works on MDP with pessimism (e.g. [1]), since they do not involve the existence of unobserved confounders.
>
> Due to the existence of confounders, the whole estimation procedure is different. Specifically, to adjust for the confounders, we make use of the bridge functions from proximal causal inference. These bridge functions are defined to satisfy certain conditional moment equations. In order to solve the moment equations from offline data, minimax estimation is needed.
>
> Furthermore, to apply pessimism, we need to construct confidence regions for uncertainty quantification. However, the construction of the confidence region is also significantly different from that in MDPs. Here we construct confidence regions as the level sets of the minimax loss w.r.t. the bridge functions. Theoretically, to apply pessimism in a valid manner, we need to establish two properties of the confidence regions: (i) the true bridge functions are in the confidence regions; (ii) all bridge functions in the confidence regions have small errors (RMSE loss). To prove these two properties, novel analysis techniques are needed, which are discussed in Section 3.2, and Appendix D.
>
> Briefly speaking, in Lemma D.3, we establish (ii) by proving a fast rate (i.e., $\tilde{\mathcal{O}}(n^{-1/2})$) for the RMSE loss. Standard analysis can only achieve a slow rate (i.e. $\tilde{\mathcal{O}}(n^{-1/4})$), which is suboptimal. To achieve our fast rate, in Appendix F.3 we take a series of novel proof steps. The main trick there is to identify a key term (term ($\star$) in Eq. (F.12)) that is related to the RMSE, and then carefully upper bound and lower bound this term respectively. The resulting upper bounds and lower bounds are then related to the RMSE loss again, which gives a quadratic inequality satisfied by RMSE. By solving this inequality, we finally derive the fast rate of the RMSE loss for **all** the bridge functions in the confidence regions.
>
> To sum up, our work is novel compared to the existing work on MDP with pessimism.
>
> **References:**
>
> [1] Xie, T., Cheng, C. A., Jiang, N., Mineiro, P., & Agarwal, A. (2021). Bellman-consistent pessimism for offline reinforcement learning. Advances in neural information processing systems, 34, 6683-6694.

---

> > ### Author Response · Authors · 2022-11-11
> > **Response to Reviewer bBxr (2)**
> >
> > **Q3:**
> > Algorithm seems not computationally efficient.
> >
> > **A3:**
> > We clarify that our algorithm P3O is information-theoretic and is indeed not tractable in the case of general function approximations.
> >
> > However, we point out that when the primal and dual function classes are parametrized (e.g., using linear function classes) and the policy class is also parametrized (e.g., finite class or using soft-max parametrization), it can become trackable.
> >
> > Therefore, for implementation, we need to consider some detailed function classes that are parametrized. For example, under the linear function approximation case considered in Appendix **H.1**, the problem becomes tractable.
> >
> > P3O contains two main steps: policy evaluation (line 3 of Algorithm 1) and policy optimization (line 4 of Algorithm 1).
> >
> > For policy evaluation, it is a minimization problem which is linear in $b_1$ by Eq (3.12). To find the confidence region $CR^\pi(\xi)$, we need to solve the minimax optimization problem defined by Eq. (3.10). Under the linear case, this reduces to the minimax optimization in the parameters of $b$ and $g$ according to Eq. (3.9).
> >
> > For policy optimization, we need to optimize over the parameters of $\pi$ according to Eq (3.13). Again under the linear case where $\Pi$ is the soft-max function class, this is implementable. In practice, we can optimize over the parameters of $\pi$ by running NPG / TRPO / PPO. This is considered in [1].
> >
> > **References:**
> >
> > [1] Provable Benefits of Actor-Critic Methods for Offline Reinforcement Learning. Andrea Zanette, Martin Wainwright, Emma Brunskill. Neurips 2021.

---

> ### Author Response · Authors · 2022-11-18
> **Response to Reviewer bBxr**
>
> Dear Reviewer bBxr, thank you very much for handling our paper! Please let us know if you have further comments given our response, and we will address further concerns.

---

### Official Review · Reviewer_gsEb · 2022-10-23

**Confidence:** 4
**Correctness:** 3
**Technical Novelty And Significance:** 3
**Empirical Novelty And Significance:** Not applicable
**Recommendation:** 6

**Clarity, Quality, Novelty And Reproducibility:**

Clarity
- (f) There are unnecessary repeated (or almost identical) sentences in Section 1 and Section 3.
- (g) Compared to the explanation of removing confounding information in Section 3.1, the discussion on addressing the distribution shift in Section 3.2 is less clear.
- (h) Unclear technical contribution F.3.

**Strength And Weaknesses:**

Strengths
- (a) The submission leverages developments in causal inference to remove confounding variables in POMDP settings.
- (b) The construction of the confidence region combines the residual bridge functions seamlessly with the minimax estimator.
- (c) The submission carefully compares notations and assumptions with those in the literature.
Rigorous analysis.

Weaknesses
- (d) The submission only applies the proximal causal inference to three policy types (C.1.1 to C.1.3). It would be better to have a systematic treatment for various confounding settings.
- (e) Although the analysis of Theorem 4.4 is standard, the main context should briefly cover the main steps to achieve the claim result. The authors can justify the technical contribution in this part.

**Summary Of The Paper:**

The submission studies offline RL under POMDP settings. Besides the typical challenge of distribution shift, the submission also needs to resolve a confounding issue due to the latent states. The submission combines proximal causal inference and confidence region built by the minimax estimator to conduct a pessimistic value estimation. The proposed algorithm achieves a regret of order $\tilde{O}(H\sqrt{log(N_{fun})/n})$, where $n$ is the number of trajectories, $H$ is the length of each trajectory, and $N_{fun}$ is a complexity measure of the function class.

**Summary Of The Review:**

Given the novelties of introducing proximal causal inference, the construction of the confidence region, and combining them to solve confounded data issue, I recommend a weak acceptance of the submission.

---

> ### Author Response · Authors · 2022-11-11
> **Response to Reviewer gsEb (1)**
>
> Thank you very much for your valuable feedback! We appreciate all your constructive suggestions and we now address your concerns point-by-point in the following.
>
> **Q1:**
> The work only applies the proximal causal inference to three policy types (C.1.1 to C.1.3). It would be better to have a systematic treatment for various confounding settings.
>
> **A1:**
> Our work actually provides one possible way of systematic treatment for various confounding settings. Specifically, the negative control variables $Z$ and $W$ can be regarded as general mapping on the observable variables. Our paper is written under this general notation of negative control variables. These three policy types (C.1.1 to C.1.3) are just some specific examples of our general setting, and they are also very common in the literature. For instance, the literature includes proxy variables for reactive policy [1, 2, 3], finite-length history policy [4, 5], and full-length history policy [6]. Finally, it is also an exciting future work for us to discover more examples that can be included to our framework based on the notion of proxy variables.
>
> **Q2:**
> Although the analysis of Theorem 4.4 is standard, the main context should briefly cover the main steps to achieve the claim result. The authors can justify the technical contribution in this part.
>
> **A2:**
> We postpone the proof sketch of Theorem 4.4 to Appendix D due to the space limit.
>
> Also, please note that though the two-step analysis framework of Theorem 4.4 is standard in the literature, its detailed proof is technically novel. Specifically, the proof relies on two technical lemmas, Lemma D.2 and Lemma D.3. Specifically,
> - Lemma D.2 shows that with high probability the confidence region contains the true value bridge function.
> - Lemma D.3 shows that all the bridge functions in the confidence region have small residual mean square error (RMSE).
>
> The proof of these lemmas requires novel and careful treatment in order to get a fast convergence rate ($\tilde{\mathcal{O}}(n^{-1/2})$). Specifically, for both Lemma D.2 and Lemma D.3, we utilize localized uniform concentration inequalities. For Lemma D.3, we use an upper-and-lower-bound argument (Page 32-34 in Appendix F.3) to derive the fast rate of the RMSE loss for **all** the bridge functions in the confidence region. We further explain the proof of these two lemmas in the answer to **Q4**.
>
> **References:**
>
> [1] Azizzadenesheli, Kamyar, Alessandro Lazaric, and Animashree Anandkumar. "Reinforcement learning of POMDPs using spectral methods." Conference on Learning Theory. PMLR, 2016.
>
> [2] Li, Yanjie, Baoqun Yin, and Hongsheng Xi. "Finding optimal memoryless policies of POMDPs under the expected average reward criterion." European Journal of Operational Research211.3 (2011): 556-567.
>
> [3] Loch, John, and Satinder Singh. "Using Eligibility Traces to Find the Best Memoryless Policy in Partially Observable Markov Decision Processes." ICML. Vol. 98. 1998.
>
> [4] Efroni, Y., Jin, C., Krishnamurthy, A., & Miryoosefi, S. (2022). Provable reinforcement learning with a short-term memory. arXiv preprint arXiv:2202.03983.
>
> [5] Uehara, Masatoshi, et al. "Provably efficient reinforcement learning in partially observable dynamical systems." arXiv preprint arXiv:2206.12020 (2022).
>
> [6] Blümlein, Theresa, Joel Persson, and Stefan Feuerriegel. "Learning Optimal Dynamic Treatment Regimes Using Causal Tree Methods in Medicine." Machine Learning for Healthcare (2022).

---

> > ### Author Response · Authors · 2022-11-11
> > **Response to Reviewer gsEb (2)**
> >
> > **Q3:**
> > Compared to the explanation of removing confounding information in Section 3.1, the discussion on addressing the distribution shift in Section 3.2 is less clear.
> >
> > **A3:**
> > Policy learning involves two components -- policy evaluation and policy optimization. Policy evaluation refers to estimating the objective $J(\pi)$ from data, and policy optimization means we want to estimate the optimal policy by choosing the policy that maximizes the estimated objective $\hat J$.
> >
> > To approximately learn the optimal policy, a sufficient condition is that $J(\pi)$ and $\hat J(\pi)$ are close for all policies $\pi \in \Pi$. Since $\hat J$ is constructed from the data, this requires that the distribution of trajectory induced by $\pi$, denoted by $d(\pi)$, is covered by the data distribution, for all $\pi$. In other words, the distributional shift between $d(\pi)$ and  $d(\text{data})$ is controlled for all $\pi$. This is a very restrictive assumption as the data collection in real application is often subject to practical limitations and cannot be arbitrarily explorative.
> >
> > In contrast, with pessimism, we only need the distribution shift between the distribution induced by $\pi^*$ (i.e. d(\pi^*)) and the data distribution $d(\text{data})$ is controlled. The data distribution does not need to cover $d(\pi)$ for any suboptimal $\pi \in \Pi$. That is, with pessimism, we can reduce the distribution shift between $d(\pi)$ and $d(\text{data})$ for all $\pi\in\Pi$, to $d( \pi^* )$ and $d(\text{data})$.
> >
> > To see why this is the case, recall that for offline policy optimization, the plan is to estimate $J(\pi)$ for $\pi \in \Pi$, and find $\hat{\pi} = \text{argmax} \hat{J} (\pi)$. The suboptimality defined as $\text{suboptimality}(\pi) = J(\pi^*) - J(\hat{\pi})$.  The suboptimality can be decomposed into three terms: $J(\pi^*) - J(\hat{\pi}) = [J(\pi^*- \hat{J}(\pi^*)] + [\hat{J} (\pi^*) - \hat{J}(\hat{\pi})] + [\hat{J}(\hat{\pi}) - J(\hat{\pi})]$.
> >
> > Pessimism can guarantee that the third term satisfies $\hat{J} (\hat{\pi}) - J (\hat{\pi}) \leq 0$.
> > For the second term, we simply have $\hat{J} (\pi^*) - \hat{J}(\hat{\pi}) \leq 0$ by $\hat{\pi} = \text{argmax} \hat{J} (\pi)$.
> > Altogether, the suboptimality can be upper bounded as $J(\pi^*) - J(\hat{\pi}) \leq |J(\pi^*- \hat{J}(\pi^*)|$. Observe that the right-hand side only depends on the optimal policy $\pi^*$. Therefore, pessimism can reduce the difference to data distribution between the trajectory induced by the optimal policy only, thus addressing the distribution shift issue.

---

> > > ### Author Response · Authors · 2022-11-11
> > > **Response to Reviewer gsEb (3)**
> > >
> > > **Q4:**
> > > Unclear technical contribution F.3.
> > >
> > > **A4:**
> > > Lemma D.3 proven in **Appendix F.3** is a key technical lemma of our paper and its proof relies on novel constructions and analysis techniques, which we explain in the following.
> > >
> > > Lemma D.3 establishes the fast rate (i.e. $\tilde{\mathcal{O}}(n^{-1/2})$) of the residual mean squared error (RMSE) for **each** bridge function in our coupled confidence regions. Compared with existing works on minimax estimation, e.g., [1, 2], they only proved the fast rate of the minimax estimator itself. In contrast, here we firstly utilize the minimax loss to construct novel confidence regions which are coupled between each time step, and secondly we prove that **each** bridge function in the proposed confidence region enjoys a fast rate of RMSE.
> > >
> > > Theoretically, in order to achieve the fast rate for the whole confidence region, we took a series of novel proof steps in Appendix F.3. In the first place, we show that the true bridge function is contained in the confidence region and it enjoys a fast rate of an empirical RMSE (Lemma F.1). Then given any bridge function in the confidence region, we identify a key term (term ($\star$) in Eq. (F.12)) which is related to the RMSE of this bridge function. By carefully upper & lower bound this term, where Lemma F.1 is applied, we eventually obtain a quadratic inequality that the RMSE of this bridge function satisfies (Eq. (F.21) in the update version). By solving this inequality, we can derive an upper bound on the RMSE loss which is uniform over the bridge functions in the confidence region (Eq. (F.22) in the update version), which is exactly the fast rate of the whole confidence region.
> > >
> > > Thank you very much for pointing this out. We update the paper by adding the above explanation at the beginning of Appendix F.3 to clarify the analysis.
> > >
> > > **Q5:**
> > > There are unnecessary repeated (or almost identical) sentences in Section 1 and Section 3.
> > >
> > > **A5:**
> > > Thank you for the feedback! We were trying to emphasize some important messages. We update the paper in the revision given your suggestion by leaving out some repeated messages at the beginning of Section 3.
> > >
> > > **References:**
> > >
> > > [1] DIKKALA, N., LEWIS, G., MACKEY, L. and SYRGKANIS, V. (2020). Minimax estimation of conditional moment models. Advances in Neural Information Processing Systems 33 12248– 12262.
> > >
> > > [2] UEHARA, M., IMAIZUMI, M., JIANG, N., KALLUS, N., SUN, W. and XIE, T. (2021). Finite sample analysis of minimax offline reinforcement learning: Completeness, fast rates and first-order efficiency. arXiv preprint arXiv:2102.02981.

---

> ### Author Response · Authors · 2022-11-18
> **Response to Reviewer gsEb**
>
> Dear Reviewer gsEb, thank you very much for handling our paper! Please let us know if you have further comments given our response, and we will address further concerns.

---

### Official Review · Reviewer_MuiH · 2022-10-24

**Confidence:** 4
**Clarity, Quality, Novelty And Reproducibility:** I think the paper is original, but th…
**Correctness:** 3
**Technical Novelty And Significance:** 3
**Empirical Novelty And Significance:** Not applicable
**Recommendation:** 6

**Strength And Weaknesses:**

*Strength*

- The paper tackles an important and challenging problem. Up to my best knowledge, this is the earliest kind of results that gives provable guarantees for offline RL in POMDPs.

- The construction of confidence intervals around bridge functions, and analysis involving it, seems new.



*Weakness*

- Additional observations (Z and W) are somewhat non-standard and unjustified. Why couldn’t them be just a part of observations from the beginning?

- This paper mostly uses the notions and assumptions defined in previous works. While related works are properly cited, they are hard to parse in general and would have been nicer if the paper has its own interpretation and intuition. For instance, the dependency of \epsilon_B^{1/4} in the final result seems quite arbitrary. It’s not clear what it even means.

- Related, coverage coefficient is defined in terms of $q_h^\pi$. How this can be interpreted in tabular settings, or in fully observable setting (a.k.a. MDP)? Such explanations could be helpful for readers.

- From a technical writing perspective, it is quite hard to follow. e.g., I cannot understand - how the construction of a confidence set is more challenging than standard problems? what is the intuition behind the construction and how do we expect it to give us polynomial bound? etc.

**Summary Of The Paper:**

This paper considers the offline RL problem in episodic-POMDPs. Specifically, data generation policy (or behavioral policy) is assumed to access the latent states while the target policy cannot, and the dataset does not contain trajectories of latent states. Basic idea is to combine the notion of bridge functions first proposed for the off-policy evaluation in POMDPs (e.g., Uehara et al., 2021), and the pessimism principle for offline RL which has gained a lot of attention recently.

In general, offline RL is intractable in POMDPs. To make the problem tractable, the paper assumes that additional observations (Z and W) are supplied such that short snapshots of trajectories satisfy a certain rank sufficiency assumption (e.g., weak-observability, Liu et al., 2022), under which the existence of bridge functions is known to be guaranteed. Under such conditions, the authors develop a simple offline RL algorithm and provide a polynomial upper bound under some additional assumptions.


**Summary Of The Review:**

This paper initiates an important study on new problems, and could be a good addition to literature.

---

> ### Author Response · Authors · 2022-11-11
> **Response to Reviewer MuiH (1)**
>
> Thank you very much for your valuable feedback! We appreciate all your constructive suggestions and we now address your concerns point-by-point in the following.
>
> **Q1:**
> Additional observations ($Z$ and $W$) are somewhat non-standard and unjustified. Why couldn’t they be just a part of observations from the beginning?
>
> **A1:**
> Actually, the variables $Z$ and $W$ are the same as the negative control action and negative control outcome of proxy variables. Such variables are widely studied and thus justified both theoretically and empirically in causal inference literature [1, 2, 3, 4].
>
> More importantly, as in the special setting of a POMDP, we require in Assumption 3.1 that both $Z$ and $W$ are measurable with respect to. the observed trajectories, which basically means that they are functions of the observed trajectories and therefore are not “additional”. We then specify the general conditions for $Z$ and $W$ to be valid proxy variables (Assumption 3.1). By using the notion of proxy variables in general rather than directly considering specific parts of observations, we aim to develop a general framework of algorithm and theory to handle confoundedness in POMDP via proxy variables.
>
> Meanwhile, we identify three concrete examples of proxy variables depending on the policy that we use. The examples, including proxy variables for a reactive policy [5, 6, 7], finite-length history policy [8, 9], and full-length history policy [10] are all broadly studied in the related literature. Finally, it is also an exciting future work for us to discover more examples that can be included to our framework based on the notion of proxy variables.
>
> **References:**
>
> [1] Lipsitch M, Tchetgen E T, Cohen T. Negative controls: a tool for detecting confounding and bias in observational studies[J]. Epidemiology (Cambridge, Mass.), 2010, 21(3): 383.
>
> [2] Miao W, Geng Z, Tchetgen Tchetgen E J. Identifying causal effects with proxy variables of an unmeasured confounder[J]. Biometrika, 2018, 105(4): 987-993.
>
> [3] Miao W, Shi X, Tchetgen E T. A confounding bridge approach for double negative control inference on causal effects[J]. arXiv preprint arXiv:1808.04945, 2018.
>
> [4] Tchetgen E J T, Ying A, Cui Y, et al. An introduction to proximal causal learning[J]. arXiv preprint arXiv:2009.10982, 2020.
>
> [5] Azizzadenesheli, Kamyar, Alessandro Lazaric, and Animashree Anandkumar. "Reinforcement learning of POMDPs using spectral methods." Conference on Learning Theory. PMLR, 2016.
>
> [6] Li, Yanjie, Baoqun Yin, and Hongsheng Xi. "Finding optimal memoryless policies of POMDPs under the expected average reward criterion." European Journal of Operational Research211.3 (2011): 556-567.
>
> [7] Loch, John, and Satinder Singh. "Using Eligibility Traces to Find the Best Memoryless Policy in Partially Observable Markov Decision Processes." ICML. Vol. 98. 1998.
>
> [8] Efroni, Y., Jin, C., Krishnamurthy, A., & Miryoosefi, S. (2022). Provable reinforcement learning with a short-term memory. arXiv preprint arXiv:2202.03983.
>
> [9] Uehara, Masatoshi, et al. "Provably efficient reinforcement learning in partially observable dynamical systems." arXiv preprint arXiv:2206.12020 (2022).
>
> [10] Blümlein, Theresa, Joel Persson, and Stefan Feuerriegel. "Learning Optimal Dynamic Treatment Regimes Using Causal Tree Methods in Medicine." Machine Learning for Healthcare (2022).

---

> > ### Author Response · Authors · 2022-11-11
> > **Response to Reviewer MuiH (2)**
> >
> > **Q2:**
> > Some notions and assumptions which appear in previous works are hard to parse in general and would have been nicer if the paper had its own interpretation and intuition. For instance, the dependency of $\epsilon_{\mathbb{B}}^{1/4}$ in the final result seems arbitrary. It’s not clear what it even means.
> >
> > **A2:**
> > We appreciate your suggestions on how the technical notions and assumptions can be better presented for our specific setting. In fact, the notions and assumptions you are probably mentioning are necessary to derive the sample efficiency of our proposed algorithm. These notions are common in the literature on offline policy evaluation (OPE) and offline RL in standard MDPs. Here we make a brief interpretation of the notions and assumptions used for theoretical analysis in our paper.
> >
> > *Assumption 4.1* is a partial coverage assumption which requires that the offline dataset generated by $π^b$ has a good coverage of the trajectories induced by $\pi^{\star}$. In view of the usage of bridge functions for identifying policy value to handle confoundedness, we customize the concentrability coefficient via the notion of weight bridge function, which can be understood as the density ratio in standard offline RL theory. In your question **Q3**, we give a more intuitive interpretation of this via the tabular MDP case.
> >
> > *Assumption 4.2* is a standard regularity assumption on the function classes $\mathbb{B}$ and $\mathbb{G}$, requiring the properties like uniform boundedness, finite localized Rademacher complexity, etc, which are commonly used in the theoretical analysis of RL algorithms with general function approximations. We also give a specific example of linear function class that satisfies all these regularity assumptions in Appendix H.
> >
> > *Assumption 4.3* is also common in the literature of model-free offline RL and we customize the assumption to adapt to the proximal causal inference tools that we use. Basically, it requires two things. The first is a completeness assumption, which requires that the dual function class $\mathbb{G}$ is rich enough to represent the residual mean squared error (RMSE, Eq. (3.6)) via a dual formulation, i.e., Eq. (3.8). The second is a realizability assumption, which requires that the primal function class $\mathbb{B}$ is rich enough to admit solutions to the value bridge function equations, i.e., Eq. (3.2).
> >
> > Finally, regarding the coefficient $\epsilon_{\mathbb{B}}^{1/4}$ in the suboptimality of P3O that you mentioned, it is just the error term that characterizes how powerful the primal function class $\mathbb{B}$ is in terms of admitting solutions to the equations that value bridge functions satisfy. Similar error terms also appear in the suboptimality bounds for standard offline RL algorithms, e.g., [1], where in that case such an error term actually characterizes how powerful the function class is in terms of admitting solutions to the equations that standard value functions satisfy, i.e., the celebrated Bellman equation. When $\mathbb{B}$ is large enough such that solutions to such equations exist in $\mathbb{B}$, then $\epsilon_{\mathbb{B}} = 0$ and P3O enjoys a $\tilde{\mathcal{O}}(n^{-1/2})$-suboptimality.
> >
> > **Q3:**
> > Related to **Q2**, the coverage coefficient is defined in terms of $q^{\pi}_h$. How this can be interpreted in tabular settings, or in fully observable settings (a.k.a. MDP)?
> >
> > **A3:**
> > For the tabular MDP case, the state is fully observable and there is no confounding. We can simply choose the negative control variables as $Z_h = W_h = S_h$, which satisfies Assumption 3.1. With this, according to Eq. (3.3), the weight bridge function $q_h^{\pi}$ is now given by
> > $$
> > q_h^{\pi}(S_h,A_h) = \frac{\mathcal{P}_h^{\pi}(S_h)}{\mathcal{P}_h^b(S_h)\pi_h^b(A_h|S_h)},
> > $$
> > which coincides with the concentrability coefficient in standard offline RL literature. Meanwhile, the partial coverage coefficient is then defined in terms of this concentrability coefficient. Similar definitions of partial coverage coefficients are used by many works, e.g., [2, 3]. Also, by the same argument, we can see that the value bridge function $b_h^{\pi}$ becomes
> > $$
> >     b_h^{\pi} (S_h,A_h) = Q_h^{\pi}(S_h,A_h)\cdot\pi_h(A_h|S_h),
> > $$
> > where $Q_h^{\pi}$ is just the action value function (Q-function) of $\pi$ in the tabular MDP.
> >
> > **References:**
> >
> > [1] Xie, T., Cheng, C. A., Jiang, N., Mineiro, P., & Agarwal, A. (2021). Bellman-consistent pessimism for offline reinforcement learning. Advances in neural information processing systems, 34, 6683-6694.
> >
> > [2] Uehara M, Sun W. Pessimistic model-based offline reinforcement learning under partial coverage[J]. arXiv preprint arXiv:2107.06226, 2021.
> >
> > [3] Liu Z, Lu M, Wang Z, et al. Welfare maximization in competitive equilibrium: Reinforcement learning for markov exchange economy[C], International Conference on Machine Learning. PMLR, 2022: 13870-13911.

---

> > > ### Author Response · Authors · 2022-11-11
> > > **Response to Reviewer MuiH (3)**
> > >
> > > **Q4:**
> > > From a technical writing perspective, it is quite hard to follow. E.g., I cannot understand how the construction of a confidence set is more challenging than standard problems? What is the intuition behind the construction and how do we expect it to give us a polynomial bound? Etc.
> > >
> > > **A4:**
> > > Actually, the construction of the confidence set is key to our algorithm and theory, and our theory features a new analysis technique for the confidence set, which serves as one of our main contributions. We now explain the challenges we face and the intuition behind the new confidence set in the following.
> > >
> > > To elaborate, we first note that the reason why we construct confidence sets for the estimation of  bridge functions is that we need it to perform pessimistic estimation in order to combat the distributional shift problem in offline policy learning. Without pessimism, we cannot derive suboptimality bounds under only a partial coverage assumption. In this case, there exists some policy in the policy class that induces a trajectory distribution that is far from the data distribution. One way to handle this issue is to impose a much stronger full coverage that the distributions induced by **all policies** are covered by the data distribution, which is impractical. Instead, we leverage pessimism to handle this issue, which necessitates the construction of confidence sets for the bridge functions.
> > >
> > > Then to construct the confidence set, we take the conditional moment equations that the bridge functions satisfy into consideration. The conditional moment equations necessitate minimax estimation since directly minimizing the MSE loss involves estimating an expectation over the square of another conditional expectation, i.e., Eq. (3.6), which is generally intractable and may induce suboptimal statistical rates. Therefore, we need to construct a confidence set based on the level set of the minimax loss, which involves different concentration and geometric properties compared with standard confidence sets based on e.g., MSE loss. Meanwhile, the new confidence set also necessitates new analysis tools to achieve a $\tilde{\mathcal{O}}(n^{-1/2})$ statistical rate.
> > >
> > > Regarding the intuition behind the construction, we hope the confidence set satisfies that i) it contains the true bridge function; ii) all the functions in the set have small minimax loss. In fact, due to our construction, the confidence set is a level set of the empirical minimax loss. Since the true bridge function satisfies the primal problem, i.e., zero minimax loss, its empirical minimax loss is also small due to a uniform concentration, which guarantees i). Based on that, by applying another uniform concentration of the empirical minimax loss over the whole confidence set, we can then show that the functions in the set also enjoy small minimax loss, which ensures ii). This is the intuition for how we construct the confidence set.

---

> ### Author Response · Authors · 2022-11-18
> **Response to Reviewer MuiH**
>
> Dear Reviewer MuiH, thank you very much for handling our paper! Please let us know if you have further comments given our response, and we will address further concerns.

---

### Official Review · Reviewer_ttMi · 2022-10-25

**Confidence:** 2
**Clarity, Quality, Novelty And Reproducibility:** The paper is well-written.
**Correctness:** 4
**Technical Novelty And Significance:** 3
**Empirical Novelty And Significance:** Not applicable
**Recommendation:** 6

**Strength And Weaknesses:**

Strength: The results are solid and the analysis is sound. From my perspective, two main technical contributions are  1) The author formulates a causal inference-based setting to tackle the cofounder issues. 2) Based on this framework and given bridge function class, the author designed a confidence region estimator.

Weakness: The paper is very abstract and considers the high-level framework. But such causal inference and the bridge function seems not very realistic to me. It would be if authors can give some real-world examples of those proxy variables or maybe test that on some causal datasets.

**Summary Of The Paper:**

This paper proposed the first provably efficient offline RL algorithm for POMDPs with a confounded dataset, which achieves $n^{-1/2}$ under partial coverage assumption. Specifically, the author solves three key difficulties: 1) To tacke the confounder issues, the paper assumes the existence of the proxy variables from causal inference and the bridge functions, but only the candidate is given, both are unobserved in the offline datasets. Then the author proposes the novel confidence regions 2) To tack the distributional shift, the author proposes pessimistic policy optimization 3) Finally, to tackle the large action space, the author allows the function approximation in their designed algorithm.

**Summary Of The Review:**

This is a typical solid theoretical paper that is beyond the acceptance threshold. But it is unclear to me how realistic or practical this framework is.

---

> ### Author Response · Authors · 2022-11-11
> **Response to Reviewer ttMi**
>
> Thank you very much for your valuable feedback! We appreciate all your comments and suggestions. We now address your concerns in the following.
>
> **Q1:**
> The causal inference and the bridge function seems not very realistic to me. It would be if authors can give some real-world examples of those proxy variables or maybe test that on some causal datasets.
>
> **A1:**
> First, let us consider the real-world example of applying the POMDP model to sepsis treatment studied by [7]. In such an example, the state, action, observation, and reward of the POMDP are given by the following:
> - State variable $S_h$ refers to the clinical state of the patient, e.g., sepsis, SIRS, Bacteremia, etc.
> - Observable variable $O_h$ refers to all the information one can read from a medical device, such as the heart rate, the respiratory rate, blood pressure, blood test result of infection, etc.
> - Action $A_h$ refers to certain treatment given to the patient. For example, each antibiotic combination can be considered as an action. As mentioned in [7], a total of 48 antibiotics have been included in the patient’s remedy.
> - Reward/cost values need to be provided empirically by physicians, based on the severity of each state. In the example of [7], the states and their corresponding rewards/costs include: Healthy (100,000), No SIRS (50,000), Probable Sepsis (PS, 5000), SIRS (-50), Bacteremia (-10,000), etc.
> - Finally, a history trajectory is the record of antibiotic treatment received by the patient. The behavior policy is some treatment plans that have been applied to some patients to generate the dataset.
>
> When using reactive policies (Example 2.1), the negative control action variable ($Z_h$) is just the observation variable $O_{h-1}$ which reflects the patient’s clinical state at the last treatment time step, and the negative control outcome variable ($W_h$) is just the observation variable $O_h$ at the current time step. Furthermore, when the observation $O$ contains enough information to reflect the underlying state $S$, which basically implies a certain full rank assumption, we can then use Example C.1 to guarantee the existence of the bridge functions. In the sepsis treatment example, it makes sense to assume that the observables (i.e. heart rate, blood pressure, blood test result, etc) can reflect whether the patient is healthy or having a blood  infection. Otherwise, why would the doctors measure these vital signals of the patient? In summary, this application in sepsis treatment gives a real-world example of the proximal causal inference setup and the bridge function.
>
> Furthermore, we need to point out again that the proximal causal inference setup is **not** unrealistic since it has been widely studied and applied in causal inference literatures, e.g., [1, 2, 3, 4]. For example, in [4], authors applied proximal causal inference to empirical applications including the right heart catheterization study, and Methotrexate study. Besides, there are several recent works on offline policy evaluation also applying the technique of bridge function when dealing with confounding issues of the data, e.g., [5, 6].
> Overall, proximal causal inference and bridge functions are useful techniques with broad real-world applications. Here in our work, we adapt proximal causal inference to design a provably sample efficient RL algorithm for offline policy learning in POMDP.
>
>
> **References:**
>
> [1] Lipsitch M, Tchetgen E T, Cohen T. Negative controls: a tool for detecting confounding and bias in observational studies[J]. Epidemiology (Cambridge, Mass.), 2010, 21(3): 383.
>
> [2] Miao W, Geng Z, Tchetgen Tchetgen E J. Identifying causal effects with proxy variables of an unmeasured confounder[J]. Biometrika, 2018, 105(4): 987-993.
>
> [3] Miao W, Shi X, Tchetgen E T. A confounding bridge approach for double negative control inference on causal effects[J]. arXiv preprint arXiv:1808.04945, 2018.
>
> [4] Tchetgen E J T, Ying A, Cui Y, et al. An introduction to proximal causal learning. arXiv preprint arXiv:2009.10982, 2020.
>
> [5] Shi, C., Uehara, M., & Jiang, N. (2021). A minimax learning approach to off-policy evaluation in partially observable markov decision processes. arXiv preprint arXiv:2111.06784.
>
> [6] Bennett, A., & Kallus, N. (2021). Proximal reinforcement learning: Efficient off-policy evaluation in partially observed markov decision processes. arXiv preprint arXiv:2110.15332.
>
> [7] Tsoukalas, Athanasios, Timothy Albertson, and Ilias Tagkopoulos. "From data to optimal decision making: a data-driven, probabilistic machine learning approach to decision support for patients with sepsis." JMIR medical informatics 3.1 (2015): e3445.

---

> ### Author Response · Authors · 2022-11-18
> **Response to Reviewer ttMi**
>
> Dear Reviewer ttMi, thank you very much for handling our paper! Please let us know if you have further comments given our response, and we will address further concerns.

---

### Decision · Program_Chairs · 2023-01-20

**Decision:**

Accept: poster

**Justification For Why Not Higher Score:**

The paper is mostly of theoretical importance and interest could be limited to the broader community.

**Justification For Why Not Lower Score:**

The paper makes solid theoretical advances in off policy optimization in POMDPs.

**Metareview: Summary, Strengths And Weaknesses:**

All the reviewers are in agreement that the paper makes a solid theoretical contribution in the are of offline policy optimization in POMDPs, based on proximal causal inference identification assumptions. Given the weak positive support by all reviewers I also read the main body of the paper and I believe this is indeed a good theoretical contribution in the literature. It combines ideas from proximal causal inference and the idea of pessimism for policy optimization, but offers new theoretical results to make the connection work. In particular, the theoretical statistical fast rate guarantee for all functions in the confidence region based on minimax estimation is interesting and novel. For this reason I recommend acceptance of this paper.

**Note From Pc:**

if the above contains the word "oral" or "spotlight" please see: "oral" presentation means -> notable-top-5% and "spotlight" means -> notable-top-25%. As stated in our emails, we are disassociating presentation type from AC recommendations